 Select

# Gauging non-invertible symmetries on the lattice

Sahand Seifnashri[1], Shu-Heng Shao[2,3] and Xinping Yang[4]

**1** School of Natural Sciences, Institute for Advanced Study, Princeton, NJ
**2** Center for Theoretical Physics, Massachusetts Institute of Technology, Cambridge, MA
**3** Yang Institute for Theoretical Physics, Stony Brook University, Stony Brook, NY
**4** Department of Physics, Yale University, New Haven, CT

## Abstract

We provide a general prescription for gauging finite non-invertible symmetries in 1+1d lattice Hamiltonian systems. Our primary example is the $\text{Rep}(D_8)$ fusion category generated by the Kennedy-Tasaki transformation, which is the simplest anomaly-free non-invertible symmetry on a spin chain of qubits. We explicitly compute its lattice F-symbols and illustrate our prescription for a particular (non-maximal) gauging of this symmetry. In our gauging procedure, we introduce two qubits around each link, playing the role of "gauge fields" for the non-invertible symmetry, and impose novel Gauss's laws. Similar to the Kramers-Wannier transformation for gauging an ordinary $\mathbb{Z}_2$, our gauging can be summarized by a gauging map, which is part of a larger, continuous non-invertible cosine symmetry.

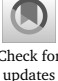

# 1 Introduction

Symmetry is a fundamental organizing principle in nature, playing a crucial role in the study of phase transitions and topological phases of matter. In recent years, the concept of global symmetry has been generalized [1], leading to new constraints on phase diagrams and a unified characterization of gapped phases based on distinct symmetry-breaking patterns of these generalized symmetries. This extension has also uncovered new topological phases protected by these symmetries.

One such generalization is the notion of non-invertible symmetry, which is implemented by conserved operators without an inverse. See, for example, [2–5] for reviews. The simplest example is the non-invertible operator in the 1+1d critical transverse-field Ising lattice model [6–8], which acts invertibly only on the $\mathbb{Z}_2$-even sector of the Hilbert space and is associated with the Kramers-Wannier duality map [9–13].

Can these non-invertible global symmetries be gauged? In continuum quantum field theory, gauging an ordinary internal discrete global symmetry $G$ involves two steps. First, we couple the theory to flat background gauge fields. Second, we promote these gauge fields to dynamical variables. However, for non-invertible symmetries, the precise notion of gauge fields remains unclear. Consequently, gauging is usually formulated without directly using gauge fields.

The key observation is that, for ordinary symmetries, coupling to flat background gauge fields is equivalent to inserting topological defects in the spacetime manifold. Thus, gauging corresponds to summing over all possible insertions of such defects. Since there are infinitely many defect configurations, one must provide a well-defined prescription for the summation. Mathematically, these defects correspond to the transition functions of the $G$-bundle, and gauging corresponds to summing over all possible $G$-bundles. This procedure of gauging a discrete global symmetry has been fully understood in the context of 1+1d conformal field theory (CFT), where it is known as orbifolding.

In 1+1d continuum quantum field theory, the gauging of finite, internal non-invertible global symmetries follows a similar prescription: one sums over the corresponding non-invertible topological defects in spacetime [14–18]. As reviewed in [17, 19], the basic idea is to triangulate the spacetime manifold and insert a network of topological defects in a way that the final outcome is independent of the chosen triangulation. See [19–27] for more recent discussions.

In this work, we extend this gauging procedure in terms of topological defects to finite, internal, non-invertible symmetries in 1+1d lattice Hamiltonian systems. Our primary example is the simplest gaugeable non-invertible symmetry on a spin chain of qubits, which is the $\mathrm{Rep}(D_8)$ fusion category [28].[1] On a finite periodic chain, the symmetry is generated by a non-invertible operator D that satisfies the algebra

$$\mathsf{D}^2 = 1 + \eta + \eta^{\mathrm{e}} + \eta^{\mathrm{o}}, \tag{1}$$

where

$$\eta^{\mathrm{e}} = \prod_{j:\mathrm{even}} X_j, \qquad \eta^{\mathrm{o}} = \prod_{j:\mathrm{odd}} X_j, \qquad \eta = \prod_j X_j, \tag{2}$$

generate a $\mathbb{Z}_2 \times \mathbb{Z}_2$ subgroup of $\mathrm{Rep}(D_8)$. The action of the non-invertible symmetry on $\mathbb{Z}_2 \times \mathbb{Z}_2$-invariant local operators is given by the Kennedy-Tasaki transformation [33, 34] (see also [12, 35, 36]):

$$\mathsf{D}\, Z_{j-1} Z_{j+1} = Z_{j-1} X_j Z_{j+1}\, \mathsf{D}, \qquad \mathsf{D}\, X_j = X_j\, \mathsf{D}. \tag{3}$$

We further construct the topological defects associated with these operators and compute explicitly their lattice F-symbols.

This symmetry is free of 't Hooft anomaly in the sense that it admits a symmetry-protected topological (SPT) phase. In fact, it admits three SPT phases [30] (see also [37–40]) whose lattice realizations are given in [28]. Correspondingly, there are three different ways to gauge the entire $\mathrm{Rep}(D_8)$ fusion category in continuum quantum field theories, which we refer to as maximal gaugings. These choices are generalizations of discrete torsions for non-invertible symmetries. See [41–52] for more discussions on non-invertible SPT phases on the lattice.

Here, we focus on one particular gauging that, roughly speaking, only gauges the "sum" of the following symmetry elements:

$$1 + \eta + \mathsf{D}. \tag{4}$$

---

[1]The Kramers-Wannier symmetry in the critical Ising model mixes with lattice translations [6, 7]. Thus, it is not clear if there exists a well-defined procedure to gauge it. Moreover, it implies an LSM-type constraint [7] that signals an 't Hooft anomaly that obstructs gauging it [29–32].

Table 1: Comparison between the gauging of an ordinary $\mathbb{Z}_2$ symmetry with that of a non-invertible symmetry. (Because of space limitation, we did not record the Hamiltonian coupled to gauge fields for the $h_2$ term.)

| | gauging $\mathbb{Z}_2$ | gauging $\mathcal{A} = 1 \oplus \eta \oplus \mathcal{D}$ of Rep($D_8$) |
|---|---|---|
| symmetric Hamiltonians | $h_0 \sum_j X_j$ $+h_1 \sum_j Z_j Z_{j+1} + \dots$ | $h_0 \sum_j X_j + h_1 \sum_j Z_{j-1} Z_{j+1}(1+X_j)$ $+h_2 \sum_j Z_{j-2} Z_{j+2}(1+X_{j-1}X_{j+1}) + \dots$ |
| gauge fields | $\sigma^{x,y,z}_{j-\frac{1}{2}}$ | $\sigma^{x,y,z}_{j-\frac{1}{2}} \quad , \quad \tau^{x,y,z}_j$ |
| Gauss's law | $\dfrac{1 + \sigma^x_{j-\frac{1}{2}} X_j \sigma^x_{j+\frac{1}{2}}}{2} = 1$ | $\dfrac{1 + \left(\tau^z_{j-1}\sigma^x_{j-\frac{1}{2}}\tau^z_j\right)}{2} \dfrac{1 + \left(\sigma^z_{j-\frac{1}{2}}\tau^x_j X_j \sigma^z_{j+\frac{1}{2}}\right)}{2} = 1$ |
| coupling to gauge fields | $h_0 \sum_j X_j +$ $h_1 \sum_j Z_j \sigma^z_{j+\frac{1}{2}} Z_{j+1} + \dots$ | $h_0 \sum_j X_j + \dfrac{h_1}{2} \sum_j Z_{j-1}Z_{j+1}(1+X_j)$ $\times \left(\sigma^x_{j-\frac{1}{2}} - \sigma^y_{j-\frac{1}{2}}\right)\left(\sigma^x_{j+\frac{1}{2}} + \tau^x_j \sigma^y_{j+\frac{1}{2}}\right) + \dots$ |
| gauging maps | $X_j \rightsquigarrow Z_{j-1}Z_j$, $Z_{j-1}Z_j \rightsquigarrow X_{j-1}$ | $X_j \rightsquigarrow X_j$, $Z_{j-1}Z_{j+1}(1+X_j) \rightsquigarrow Z_{j-1}Z_{j+1}(1+X_j)$, $Z_{j-2}Z_{j+2}(1+X_{j-1}X_{j+1})$ $\rightsquigarrow Z_{j-2}Z_{j+2}(1+X_{j-1}X_{j+1})(X_j)^{\frac{1-X_{j-1}}{2}}$ |
| enhanced symmetries | Kramers-Wannier $(KW)^\dagger KW = 1 + \eta$ | Rep($D_{16}$) fusion category $(L_{\frac{\pi}{4}})^\dagger(L_{\frac{\pi}{4}}) = (L_{\frac{3\pi}{4}})^\dagger(L_{\frac{3\pi}{4}}) = 1 + \eta + D$ |

This non-maximal gauging generalizes the notion of gauging an anomaly-free subgroup in the case of ordinary finite group symmetries. Mathematically, the sum, along with certain extra data, constitute an *algebra* in the Rep($D_8$) fusion category.

Our lattice gauging procedure follows the continuum one closely by summing over the non-invertible topological defects and operators. However, since space and time are treated on different footings in Hamiltonian lattice systems, our approach also uncovers new insights. Interestingly, our prescription leads to a precise notion of "gauge fields" for non-invertible symmetries, realized as qubits on each link coupled to the original matter degrees of freedom. More specifically, our gauging procedure includes two steps. First, we introduce these "gauge field" qubits on every link. Second, we impose Gauss's law constraints. This prescription mirrors the gauging of an on-site, ordinary $\mathbb{Z}_2$ symmetry, which we review in Appendix A. See Table 1 for a comparison between gauging a $\mathbb{Z}_2$ symmetry and that of a non-invertible symmetry.

Along the way, we also encounter an interesting continuous, non-invertible symmetry that is sometimes referred to as the *cosine* non-invertible symmetry. This symmetry was first dis-

cussed in the context of the orbifold branch of $c = 1$ compact boson CFT in [53, 54]. The cosine non-invertible symmetry includes the Rep($D_8$) symmetry as its subcategory and admits a very simple expression:[2]

$$L_\theta = \frac{1+\eta}{2} \left( e^{i\theta Q} + e^{-i\theta Q} \right),$$ (5)

where $Q = \frac{1}{2}\sum_{j=1}^{L}(-X_1)(-X_2)\cdots(-X_j)$ is a non-local "charge" for this symmetry.

Let us compare our gauging procedure on the lattice with other approaches in the literature. Our prescription generalizes the gauging of (non-on-site) invertible symmetries described in [56]. See [57–59] for gauging Matrix Product Unitary (MPU) symmetries. The authors of [46] gauge the entire non-invertible Rep($G$) fusion category on the group-based Pauli Hilbert space. In contrast, we focus on a non-maximal gauging of Rep($D_8$) on a qubit chain. Moreover, our prescription can be applied to arbitrary gaugings in terms of algebra objects. Finally, [60] constructs a Matrix Product Operator (MPO) presentation of the gauging map and shows that it coincides with the duality MPOs of [61, 62].

In Section 2, we describe the lattice realization of Rep($D_8$) non-invertible symmetry operators and defects in detail. In particular, we discuss fusion operators and use them to compute all the lattice F-symbols explicitly. In Section 3, we describe a non-maximal gauging of this symmetry. Finally, Section 4 discusses the generalization to gauging arbitrary finite non-invertible symmetries.

In Appendix A, we review the gauging of spin-flip $\mathbb{Z}_2$ symmetry in a language that is generalizable for non-invertible symmetries. Appendix B includes the derivation of the cosine symmetry and the construction of its topological defects. Appendix C, D, and E contain detailed computations involving gauging and F-symbols.

## 2 Rep($D_8$) symmetry operators and defects

In this section, we discuss the non-invertible symmetry operators and their associated defects in Hamiltonian lattice systems. In Section 2.1, we motivate the discussion by starting with a simple Hamiltonian model and discussing its symmetries. Section 2.2 gives a more detailed discussion of the non-invertible operators as matrix product operators. In Section 2.3, we discuss the defects. We explicitly compute their lattice F-symbols and show that they match with the Rep($D_8$) fusion category.

### 2.1 Hamiltonian and its phase diagram

We focus on the Hamiltonian

$$H = h_0 \sum_{j=1}^{L} X_j + h_1 \sum_{j=1}^{L} Z_{j-1}Z_{j+1}(1+X_j).$$ (6)

We assume the number of sites $L$ is even and periodic boundary conditions. This Hamiltonian has been studied in [55, 63–65] and is referred to as the Ising zigzag model or the dual XXZ model. Later on, we will discuss infinitely many deformations of this Hamiltonian while preserving the symmetry of our interest.

As discussed in Appendix B, this Hamiltonian is related to XXZ Hamiltonian (7)

$$H_{XXZ} = h_1 \sum_{j=1}^{L}(X_j X_{j+1} + Y_j Y_{j+1}) - h_0 \sum_{j=1}^{L} Z_j Z_{j+1},$$ (7)

---

[2]The MPO expression for this operator was recently discussed in [55].

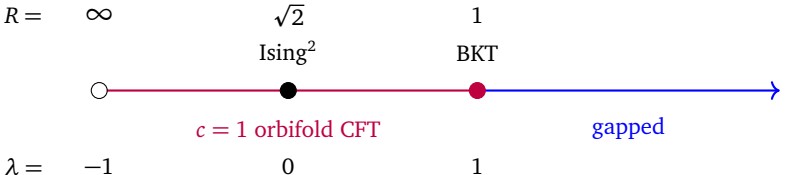

Figure 1: The phase diagram of the Hamiltonian (6). For $\lambda = -h_0/h_1 \in (-1, 1]$, it flows to the $c = 1$ orbifold CFT with radius $R$ given in (8). For $\lambda > 1$ it is gapped with one ground state.

by gauging a $\mathbb{Z}_2$ symmetry (which is implemented by the Kramers-Wannier transformation) followed by a unitary transformation. The XXZ model is gapless for $\lambda \equiv -h_0/h_1 \in (-1, 1]$, and there is a Berezinskii–Kosterlitz–Thouless (BKT) transition at $\lambda = 1$, beyond which point the model is gapped with two ground states. For even $L$, the XXZ model is known to flow to the $c = 1$ compact boson CFT with radius $R$ given by [66, 67]:[3]

$$R^2 = \frac{1}{1 - \frac{1}{\pi} \cos^{-1}(\lambda)}, \quad \lambda \in (-1, 1]. \tag{8}$$

(The XXZ model with odd $L$ corresponds to the same CFT but with a topological defect insertion [69].)

We can, therefore, deduce the phase diagram of (6) from the above well-known phase diagram of the XXZ model. For $\lambda \in (-1, 1]$, our Hamiltonian (6) is described by the $c = 1$ $S^1/\mathbb{Z}_2$ orbifold CFT with radius given by (8). In particular, the $\lambda = 0$ point corresponds to the $R = \sqrt{2}$ orbifold CFT, which is known to be two decoupled Ising CFT. There is a BKT transition at $\lambda = 1$, corresponding to the $R = 1$ orbifold CFTs (also known as $SU(2)_1/\mathbb{Z}_2$), which sits at the intersection point between the compact boson and orbifold branches [68]. The model is gapped with a unique ground state when $\lambda > 1$. See Figure 1 for a summary of the phase diagram.

## 2.2 Symmetry operators

The Hamiltonian (6) is invariant under the non-invertible symmetry transformation

$$Z_{j-1}Z_{j+1} \rightsquigarrow Z_{j-1}X_jZ_{j+1}, \qquad X_j \rightsquigarrow X_j. \tag{9}$$

This transformation is implemented by the non-invertible operator

$$\mathrm{D} = \mathrm{Tr}_\ell(\mathbb{U}_\mathrm{D}^L \mathbb{U}_\mathrm{D}^{L-1} \cdots \mathbb{U}_\mathrm{D}^1) = \boxed{\mathbb{U}_\mathrm{D}^1}\!-\!\boxed{\mathbb{U}_\mathrm{D}^2}\!-\cdots-\!\boxed{\mathbb{U}_\mathrm{D}^L}\,, \tag{10}$$

where

$$\mathbb{U}_\mathrm{D}^j = \mathrm{CNOT}_{\ell,j}\,\mathrm{H}_\ell = \frac{1}{\sqrt{2}}\begin{pmatrix} 1 & 1 \\ X_j & -X_j \end{pmatrix}. \tag{11}$$

Here $\mathrm{CNOT}_{\ell,j} = X_j^{\frac{1-Z_\ell}{2}}$ is the CNOT gate, and $\mathrm{H}_\ell = (X_\ell + Z_\ell)/\sqrt{2}$ is the Hadamard gate. This expression makes it manifest that D is a Matrix Product Operator (MPO) with the local tensor

---

[3]Our convention for the radius of the compact boson CFT is such that T-duality acts as $R \to 1/R$ and $R = 1$ is the self-dual point with $SU(2)_1$ Kac-Moody algebra. Our convention differs from [68] whose radius $R_{\mathrm{Ginsparg}}$ is related to ours by $R = \sqrt{2}R_{\mathrm{Ginsparg}}$.

at site $j$ given by $\mathbb{U}_D^j$. $X_\ell$ and $Z_\ell$ act on the auxiliary qubit labeled by $\ell$, and the trace $\mathrm{Tr}_\ell$ is taken only over the auxiliary qubit. The precise meaning of the non-invertible transformation (9) is

$$DZ_{j-1}Z_{j+1} = Z_{j-1}X_jZ_{j+1}D, \quad DX_j = X_jD. \tag{12}$$

One important property here is that the MPO tensor is unitary, i.e., $(\mathbb{U}_D^j)^\dagger = (\mathbb{U}_D^j)^{-1}$, where $\dagger$ is the complex transpose on both the physical qubits (labeled by $j$) and the auxiliary qubit (labeled by $\ell$).[4] It follows that this MPO tensor $\mathbb{U}_D^j$ is identical to the movement operator of the defect, as we will discuss later.

The non-invertible operator D satisfies the fusion algebra

$$
\begin{aligned}
D^2 = 1 + \eta + \eta^e + \eta^o\,, \qquad & \eta^e D = \eta^o D = D\eta^e = D\eta^o = D\,, \\
\eta = \eta^e\eta^o = \eta^o\eta^e\,, \qquad & (\eta^e)^2 = (\eta^o)^2 = 1\,.
\end{aligned}
\tag{14}
$$

Here,

$$\eta^e = \prod_{j:\mathrm{even}} X_j\,, \qquad \eta^o = \prod_{j:\mathrm{odd}} X_j\,, \qquad \eta = \prod_j X_j \tag{15}$$

form the $\mathbb{Z}_2^e \times \mathbb{Z}_2^o$ symmetry of the Hamiltonian (6). This algebra is the fusion ring of the irreducible representations of the dihedral group $D_8$ of order 8, with D corresponding to the 2-dimensional irrep and $1, \eta^e, \eta^o, \eta$ corresponding to the trivial irrep and the three non-trivial 1-dimensional irreps. These operators together form a $\mathrm{Rep}(D_8)$ fusion category on the lattice.[5]

We conclude that there is a $\mathrm{Rep}(D_8)$ fusion category for every Hamiltonian in the one-parameter family in (6). This is consistent with the fact that the entire branch of the $c = 1$ orbifold CFT (which is the continuum limit of (6) for $\lambda \in (-1, 1]$) also has the same fusion category symmetry [54].

### 2.2.1 Continuous non-invertible symmetry

The non-invertible $\mathrm{Rep}(D_8)$ symmetry of the Hamiltonian (6) is part of a larger and continuous non-invertible symmetry, sometimes referred to as the "cosine" symmetry [53, 54]. As we explain in Appendix B, the cosine symmetry is related to the O(2) symmetry of the XXZ chain by gauging the $\mathbb{Z}_2$ spin-flip/charge-conjugation symmetry. The lattice realization of the cosine symmetry has recently been studied in [55] for the same Hamiltonian (6).

The cosine symmetry is implemented by non-invertible operators $L_\theta$ given by

$$
\begin{aligned}
L_\theta &= \frac{1+\eta}{2}\left(e^{i\theta Q} + e^{-i\theta Q}\right), \\
Q &= \frac{1}{2}(1 - X_1 + X_1X_2 - X_1X_2X_3\ldots - X_1X_2\cdots X_{L-1}),
\end{aligned}
\tag{16}
$$

which commute with Hamiltonian (6). They satisfy the cosine-like fusion algebra relations

$$L_\theta L_{\theta'} = L_{\theta+\theta'} + L_{\theta-\theta'}\,, \qquad (L_\theta)^\dagger = L_{-\theta} = L_\theta\,, \qquad L_{\theta+2\pi} = L_\theta\,. \tag{17}$$

---

[4]On the other hand, the MPO tensor for the non-invertible Kramers-Wannier operator in the critical Ising model, whose MPO tensor is [7, 10]

$$\mathbb{U}_{KW}^j = \begin{pmatrix} |+\rangle\langle0|_j & |-\rangle\langle1|_j \\ |-\rangle\langle0|_j & |+\rangle\langle1|_j \end{pmatrix} = \frac{1}{2\sqrt{2}}\begin{pmatrix} 1+X_j-iY_j+Z_j & -1+X_j+iY_j+Z_j \\ 1-X_j+iY_j+Z_j & 1+X_j+iY_j-Z_j \end{pmatrix}, \tag{13}$$

which is not unitary. See [57, 61, 62, 70–72] for more general MPO symmetries.

[5]The operator D is related to the $\mathrm{Rep}(D_8)$ operator introduced in [28] (see also [73]) by conjugation of $\prod_{j=1}^L CZ_{j,j+1}$. The operator D leaves the product state $|++\cdots\rangle$ invariant, whereas the one in [28] leaves the $\mathbb{Z}_2 \times \mathbb{Z}_2$ cluster state invariant. The MPO (11) for D has bond dimension two, which equals the quantum dimension of the non-invertible object in the $\mathrm{Rep}(D_8)$ fusion category. (This was referred to as the "on-site" condition for an MPO in [51].)

The cosine symmetry has the following MPO presentation

$$\mathsf{L}_\theta = \mathrm{Tr}_\ell\left(\mathbb{U}_\theta^L \cdots \mathbb{U}_\theta^2 \mathbb{U}_\theta^1\right), \quad \text{where} \quad \mathbb{U}_\theta^j = \mathrm{CNOT}_{\ell,j} Z_\ell e^{i\theta\frac{Y_\ell}{2}} = \begin{pmatrix} \cos\frac{\theta}{2} & \sin\frac{\theta}{2} \\ \sin\frac{\theta}{2} X_j & -\cos\frac{\theta}{2} X_j \end{pmatrix}. \quad (18)$$

Note that even though the charge $Q$ is not translation invariant, the cosine symmetry $\mathsf{L}_\theta$ is. Indeed, the MPO presentation (18) manifests the translation invariance.

We notice that the $\mathrm{Rep}(D_8)$ symmetry is included in the cosine symmetry as

$$\mathsf{L}_0 = 1 + \eta^\mathrm{e}\eta^\mathrm{o}, \qquad \mathsf{L}_\pi = \eta^\mathrm{e} + \eta^\mathrm{o}, \qquad \mathsf{L}_{\pi/2} = \mathsf{D}. \quad (19)$$

More generally, the cosine operator $\mathsf{L}_{2\pi/n}$ (together with some other invertible operators) generate a $\mathrm{Rep}(D_{2n})$ fusion category.

The one-parameter family of Hamiltonians (6) respects the continuous non-invertible cosine symmetry. This is consistent with the fact that the entire branch of the $c = 1$ orbifold CFT preserves the cosine symmetry [53, 54]. However, more general $\mathrm{Rep}(D_8)$ symmetric deformations of (6) would break the cosine symmetry (such as the $h_2$ term discussed in Section 3.2.3).

## 2.3 Symmetry defects

Having discussed the non-invertible operator $\mathsf{D}$, we now discuss the associated defect, which is a local modification of the Hamiltonian and the Hilbert space. To distinguish the operator from the defect, we denote the latter by $\mathcal{D}$ with a different font.

### 2.3.1 Movement operators

The only input data we need to derive the defect is the *movement operator* $\lambda_\mathcal{D}^j$. (This terminology will become clear in a moment.) For the $\mathrm{Rep}(D_8)$ symmetry, the movement operator coincides with the MPO tensor $\mathbb{U}_\mathsf{D}^j$:

$$\lambda_\mathcal{D}^j = \boxed{\mathbb{U}_\mathsf{D}^j} = \mathrm{CNOT}_{\ell,j}\, \mathsf{H}_\ell. \quad (20)$$

For the following discussion, we start with an infinite chain and derive a local expression for the defect. By the locality of the defect, the final result can be defined on a finite chain. Consider a general $\mathrm{Rep}(D_8)$-symmetric Hamiltonian, which is invariant under (9) and therefore commutes with the non-invertible operator $\mathsf{D}$. The Hamiltonian with defect $\mathcal{D}$ inserted at link $(J, J+1)$ is given by

$$H_\mathcal{D}^{(J,J+1)} = U_\mathcal{D}^{\leq J} H (U_\mathcal{D}^{\leq J})^\dagger. \quad (21)$$

Here, $U_\mathcal{D}^{\leq J}$ is a transformation of the operator algebra defined by a formal sequence of unitary transformations:

$$U_\mathcal{D}^{\leq J} = \lambda_\mathcal{D}^J \lambda_\mathcal{D}^{J-1} \lambda_\mathcal{D}^{J-2} \cdots = \cdots \boxed{\mathbb{U}_\mathsf{D}^{J-2}}\boxed{\mathbb{U}_\mathsf{D}^{J-1}}\boxed{\mathbb{U}_\mathsf{D}^{J}}. \quad (22)$$

We refer to $U_\mathcal{D}^{\leq J}$ as the *defect creation homomorphism*. The defect Hamiltonian is defined on a Hilbert space with an extra qubit $|0\rangle_\ell, |1\rangle_\ell$ that is localized at link $(J, J+1)$ and is acted by the Pauli operators $X_\ell, Z_\ell$. Mathematically, $U_\mathcal{D}^{\leq J} : \mathscr{A} \to \mathscr{A}_\mathcal{D}^{(J,J+1)}$ is a homomorphism from the operator algebra $\mathscr{A}$ of local operators to the defect operator algebra $\mathscr{A}_\mathcal{D}^{(J,J+1)} = \mathscr{A} \otimes \mathrm{End}(\mathbb{C}_\ell^2)$,

where $\mathbb{C}^2_\ell$ stands for the extra qubit. The defect creation homomorphism maps local operators to operators with finite, though potentially arbitrarily large, support. The non-invertibility of the defect corresponds to the fact that this homomorphism is not surjective and, consequently, does not have a right-inverse.

The movement operator acts on local operators as[6]

$$
\lambda^j_{\mathcal{D}} : \quad
\begin{matrix}
X_j \mapsto X_j\,, & X_\ell \mapsto Z_\ell\,, \\
Z_j \mapsto Z_j Z_\ell\,, & Z_\ell \mapsto X_\ell X_j\,.
\end{matrix}
\tag{23}
$$

It follows that $U^{\leq J}_{\mathcal{D}}$ acts on local operators as

$$
U^{\leq J}_{\mathcal{D}} : \quad
\begin{matrix}
X_j \mapsto X_j\,, & \text{for all } j\,, \\
Z_j \mapsto Z_j\,, & \text{for } j > J\,, \\
Z_j \mapsto Z_j X_{j+1} X_{j+3} \cdots X_{J-3} X_{J-1} Z_\ell\,, & \text{for } j \leq J \text{ and } J - j = 0 \pmod 2\,, \\
Z_j \mapsto Z_j X_{j+1} X_{j+3} \cdots X_{J-2} X_J X_\ell\,, & \text{for } j \leq J \text{ and } J - j = 1 \pmod 2\,.
\end{matrix}
\tag{24}
$$

From this we find the Hamiltonian for (6) with a $\mathcal{D}$ defect inserted at link $(J, J+1)$:

$$
\begin{aligned}
H^{(J,J+1)}_{\mathcal{D}} = h_0 \sum_j X_j + h_1 \sum_{j \neq J, J+1} Z_{j-1} Z_{j+1} (1 + X_j) \\
+ h_1 X_\ell Z_{J-1} Z_{J+1} (1 + X_J) + h_1 Z_\ell Z_J Z_{J+2} (1 + X_{J+1})\,.
\end{aligned}
\tag{25}
$$

This defect is topological since it can be moved via the unitary movement operator $\lambda^J_{\mathcal{D}}$ in the sense that

$$
\lambda^J_{\mathcal{D}} H^{(J-1,J)}_{\mathcal{D}} (\lambda^J_{\mathcal{D}})^{-1} = H^{(J,J+1)}_{\mathcal{D}}\,.
\tag{26}
$$

In Appendix B, we derive the defect Hamiltonian for the more general cosine symmetry for the Hamiltonian (6), with the $\mathrm{Rep}(D_8)$ defect being a special case. In Appendix C, we discuss the $\mathrm{Rep}(D_8)$ defect Hamiltonians of more general $\mathrm{Rep}(D_8)$ symmetric Hamiltonians.

The defect Hamiltonians for the invertible symmetries $\eta^{\mathrm{e}}, \eta^{\mathrm{o}}, \eta$ are given by

$$
H^{(1,2)}_\eta = h_0 \sum_j X_j + h_1 \left[ \sum_{j \neq 1,2} Z_{j-1} Z_{j+1} (1 + X_j) - Z_0 Z_2 (1 + X_1) - Z_1 Z_3 (1 + X_2) \right]\,,
$$

$$
H^{(1,2)}_{\eta^{\mathrm{e}}} = h_0 \sum_j X_j + h_1 \left[ \sum_{j \neq 1,2} Z_{j-1} Z_{j+1} (1 + X_j) - Z_0 Z_2 (1 + X_1) + Z_1 Z_3 (1 + X_2) \right]\,,
\tag{27}
$$

$$
H^{(1,2)}_{\eta^{\mathrm{o}}} = h_0 \sum_j X_j + h_1 \left[ \sum_{j \neq 1,2} Z_{j-1} Z_{j+1} (1 + X_j) + Z_0 Z_2 (1 + X_1) - Z_1 Z_3 (1 + X_2) \right]\,,
$$

where $H^{(2,3)}_{\eta^{\mathrm{o}}} = H^{(1,2)}_{\eta^{\mathrm{o}}}$, and $H^{(0,1)}_{\eta^{\mathrm{e}}} = H^{(1,2)}_{\eta^{\mathrm{e}}}$. The corresponding movement operators are

$$
\lambda^j_g =
\begin{cases}
(-1)^{g_{\mathrm{e}} g_{\mathrm{o}}} X^{g_{\mathrm{o}}}_j\,, & \text{for odd } j\,, \\
(-1)^{g_{\mathrm{e}} g_{\mathrm{o}}} X^{g_{\mathrm{e}}}_j\,, & \text{for even } j\,.
\end{cases}
\tag{28}
$$

Explicitly, they are $\lambda^j_\eta = -X_j$, $\lambda^{2n}_{\eta^{\mathrm{e}}} = +X_{2n}$, $\lambda^{2n+1}_{\eta^{\mathrm{o}}} = +X_{2n+1}$, $\lambda^{2n+1}_{\eta^{\mathrm{e}}} = \lambda^{2n}_{\eta^{\mathrm{o}}} = 1$. These movement operators obey equations similar to (26).

The defining equations of the movement operator, such as (26), only determine them up to phases. Here, we have chosen a specific phase for each $\lambda^j_\bullet$ in (20) and (28) to match the lattice F-symbols with those in the continuum. See Section 2.3.3 and Appendix D.

---

[6]For a unitary operator $U$, we define the arrow $\mapsto$ as $U : \mathcal{O} \mapsto U \mathcal{O} U^\dagger$. For a non-invertible operator D, we write $\mathcal{O}_1 \rightsquigarrow \mathcal{O}_2$ if $D \mathcal{O}_1 = \mathcal{O}_2 D$.

### 2.3.2 Fusion operators

Having discussed the topological defects of the $\text{Rep}(D_8)$ non-invertible symmetry, here we discuss the fusion between them. In general, the fusion between two defects, $\mathcal{L}$ and $\mathcal{M}$, is implemented by a local operator[7]

$$(\lambda^j)^{\mathcal{N}}_{\mathcal{L}\otimes\mathcal{M}} \;=\; \text{[diagram]}\,, \tag{29}$$

that we call the *fusion operator* and satisfies

$$(\lambda^j)^{\mathcal{N}}_{\mathcal{L}\otimes\mathcal{M}} H^{(j-1,j);(j,j+1)}_{\mathcal{L};\mathcal{M}} = H^{(j,j+1)}_{\mathcal{N}} (\lambda^j)^{\mathcal{N}}_{\mathcal{L}\otimes\mathcal{M}}\,. \tag{30}$$

Here, $H^{(j-1,j);(j,j+1)}_{\mathcal{L};\mathcal{M}}$ is the defect Hamiltonian of $\mathcal{L}$ and $\mathcal{M}$ defects inserted on links $(j-1,j)$ and $(j,j+1)$.

The topological defects of the $\text{Rep}(D_8)$ symmetry satisfy an important property that is the corresponding movement operators commute with each other. For instance, $\lambda^j_{\mathcal{D}}$ commutes with $\lambda^{j'}_{\eta}$ even for $j = j'$. This property implies that we can unambiguously insert multiple defects on a single link. To do that, we start by creating two defects that are well-separated and then act with the movement operators to bring both of them on a particular link. The above property guarantees that the final result is independent of the order in which we bring these two defects to that particular link.

Since inserting multiple defects on a single link is unambiguous, the fusion operation of defects on adjacent links can be broken into two steps: first, bringing them onto a single link and then performing the fusion. The latter is implemented by the *on-site* fusion operator that we denote by

$$m^{\mathcal{N}}_{\mathcal{L}\otimes\mathcal{M}} \;=\; \text{[diagram]}\,, \tag{31}$$

which satisfies

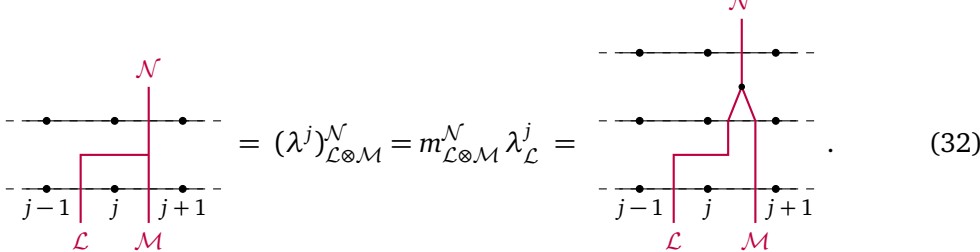

$$= (\lambda^j)^{\mathcal{N}}_{\mathcal{L}\otimes\mathcal{M}} = m^{\mathcal{N}}_{\mathcal{L}\otimes\mathcal{M}} \lambda^j_{\mathcal{L}} = \tag{32}$$

For instance, starting with an $\eta$ defect and a $\mathcal{D}$ defect on links $(0,1)$ and $(1,2)$

$$H^{(0,1);(1,2)}_{\eta;\mathcal{D}} = h_0 \sum_j X_j + h_1 \sum_{j\neq 0,1,2} Z_{j-1}Z_{j+1}(1+X_j)$$
$$- h_1 Z_{-1}Z_1(1+X_0) - h_1 X_\ell Z_0 Z_2(1+X_1) + h_1 Z_\ell Z_1 Z_3(1+X_2)\,, \tag{33}$$

---

[7]More precisely, $(\lambda^j)^{\mathcal{N}}_{\mathcal{L}\otimes\mathcal{M}} : \mathcal{H}^{(j-1,j);(j,j+1)}_{\mathcal{L};\mathcal{M}} \to \mathcal{H}^{(j,j+1)}_{\mathcal{N}}$ is a locally acting transformation/homomorphism since its domain and codomain might be different. Here $\mathcal{H}^{(j-1,j);(j,j+1)}_{\mathcal{L};\mathcal{M}}$ and $\mathcal{H}^{(j,j+1)}_{\mathcal{N}}$ are the corresponding defect Hilbert spaces.

we first move the $\eta$ defect, by conjugation with $\lambda^1_\eta = X_1$, to link $(1,2)$ and find

$$
\begin{aligned}
H^{(1,2)}_{\eta \otimes \mathcal{D}} = \lambda^1_\eta H^{(0,1);(1,2)}_{\eta;\mathcal{D}} (\lambda^1_\eta)^{-1} = h_0 \sum_j X_j + h_1 \sum_{j \neq 1,2} Z_{j-1} Z_{j+1}(1 + X_j) \\
- h_1 X_\ell Z_0 Z_2 (1 + X_1) - h_1 Z_\ell Z_1 Z_3 (1 + X_2).
\end{aligned}
\tag{34}
$$

Now, we fuse the two defects using the on-site fusion operator

$$
m^{\mathcal{D}}_{\eta \otimes \mathcal{D}} = Z_\ell X_\ell =
\tag{35}
$$

that satisfies

$$
m^{\mathcal{D}}_{\eta \otimes \mathcal{D}} H^{(1,2)}_{\eta \otimes \mathcal{D}} = H^{(1,2)}_{\mathcal{D}} m^{\mathcal{D}}_{\eta \otimes \mathcal{D}}.
\tag{36}
$$

Note that the on-site fusion operator $m^{\mathcal{D}}_{\eta \otimes \mathcal{D}}$ does not act on the physical degrees of freedom and only acts on the defect degrees of freedom $|0\rangle_\ell$ and $|1\rangle_\ell$.[8] Using (32), we find the fusion operator $(\lambda^1)^{\mathcal{D}}_{\eta \otimes \mathcal{D}} = Z_\ell X_\ell X_1$, which obeys

$$
(\lambda^1)^{\mathcal{D}}_{\eta \otimes \mathcal{D}} H^{(0,1);(1,2)}_{\eta;\mathcal{D}} = H^{(1,2)}_{\mathcal{D}} (\lambda^1)^{\mathcal{D}}_{\eta \otimes \mathcal{D}}.
\tag{37}
$$

Up to phases, the non-trivial on-site fusion operators involving $\eta$ and $\mathcal{D}$ defects are

$$
\begin{aligned}
m^{\mathcal{D}}_{\eta \otimes \mathcal{D}} = Z_\ell X_\ell, \qquad & \qquad m^{\mathcal{D}}_{\mathcal{D} \otimes \eta} = X_\ell Z_\ell, \\
m^1_{\mathcal{D}_1 \otimes \mathcal{D}_2} = \langle 00|_{\ell_1,\ell_2} + \langle 11|_{\ell_1,\ell_2}, \qquad & \qquad m^\eta_{\mathcal{D}_1 \otimes \mathcal{D}_2} = \langle 01|_{\ell_1,\ell_2} - \langle 10|_{\ell_1,\ell_2}.
\end{aligned}
\tag{38}
$$

To make it clear which qubit is associated with which defect, we will add an extra subscript $i = 1,2,3,\dots$ to the defect label, such as $\mathcal{D}_i$, to specify this piece of information whenever this is an ambiguity. Here, the qubits $\ell_1$ and $\ell_2$, respectively, corresponds to the left and right $\mathcal{D}$ defects in the fusion $\mathcal{D}_1 \otimes \mathcal{D}_2$. However, we emphasize that $\mathcal{D}_1$ and $\mathcal{D}_2$ are the *same* kind of defect, and $i = 1,2$ is just a label to distinguish these two copies of the same defect. The on-site fusion operators $m^1_{\mathcal{D}_1 \otimes \mathcal{D}_2}$ and $m^\eta_{\mathcal{D}_1 \otimes \mathcal{D}_2}$ satisfy

$$
m^1_{\mathcal{D}_1 \otimes \mathcal{D}_2} H^{(1,2)}_{\mathcal{D}_1 \otimes \mathcal{D}_2} = H\, m^1_{\mathcal{D}_1 \otimes \mathcal{D}_2}, \quad \text{and} \quad m^\eta_{\mathcal{D}_1 \otimes \mathcal{D}_2} H^{(1,2)}_{\mathcal{D}_1 \otimes \mathcal{D}_2} = H^{(1,2)}_\eta\, m^\eta_{\mathcal{D}_1 \otimes \mathcal{D}_2},
\tag{39}
$$

where

$$
\begin{aligned}
H^{(1,2)}_{\mathcal{D}_1 \otimes \mathcal{D}_2} = h_0 \sum_j X_j + h_1 \sum_{j \neq 1,2} Z_{j-1} Z_{j+1}(1 + X_j) \\
+ h_1 X_{\ell_1} X_{\ell_2} Z_0 Z_2 (1 + X_1) + h_1 Z_{\ell_1} Z_{\ell_2} Z_1 Z_3 (1 + X_2).
\end{aligned}
\tag{40}
$$

The other on-site fusion operators involving $1$, $\eta$, and $\mathcal{D}$ are:

$$
m^a_{1 \otimes a} = m^a_{a \otimes 1} = -m^1_{\eta \otimes \eta} = 1, \qquad \text{for } a = 1, \eta, \mathcal{D}.
\tag{41}
$$

The remaining on-site fusion operators involve the defects $\eta^{\mathrm{e}}$ and $\eta^{\mathrm{o}}$. To shorten the expressions, we will parametrize $\mathbb{Z}^{\mathrm{e}}_2 \times \mathbb{Z}^{\mathrm{o}}_2$ defects by $g = (g_{\mathrm{e}}, g_{\mathrm{o}})$ such that

---

[8]Even though $m^{\mathcal{D}}_{\eta \otimes \mathcal{D}}$ happens to be independent of $j$, on-site fusion operators generally can depend on $j$. For instance, those involving $\eta^{\mathrm{e}}$ or $\eta^{\mathrm{o}}$, which we will discuss later, do. Nonetheless, we suppress the $j$ dependence to avoid cluttering.

$\eta^e = (1,0), \eta^o = (0,1), \eta = (1,1), 1 = (0,0)$. Using this convention, the on-site fusion operators at site $j$ are compactly given by

$$m_{g \otimes h}^{g+h} = \begin{cases} (-1)^{g_o h_e}, & \text{for odd } j, \\ (-1)^{g_e h_o}, & \text{for even } j, \end{cases}$$

$$m_{g \otimes \mathcal{D}_1}^{\mathcal{D}_2} = \left(m_{\mathcal{D}_2 \otimes g}^{\mathcal{D}_1}\right)^\dagger = \begin{cases} \left(|0\rangle_{\ell_2}\langle 0|_{\ell_1} + |1\rangle_{\ell_2}\langle 1|_{\ell_1}\right)(Z_{\ell_1})^{g_e}(X_{\ell_1})^{g_o}, & \text{for odd } j, \\ \left(|0\rangle_{\ell_2}\langle 0|_{\ell_1} + |1\rangle_{\ell_2}\langle 1|_{\ell_1}\right)(Z_{\ell_1})^{g_o}(X_{\ell_1})^{g_e}, & \text{for even } j, \end{cases} \tag{42}$$

$$m_{\mathcal{D}_1 \otimes \mathcal{D}_2}^{g} = \begin{cases} \left(\langle 00|_{\ell_1,\ell_2} + \langle 11|_{\ell_1,\ell_2}\right)(X_{\ell_1})^{g_o}(Z_{\ell_1})^{g_e}, & \text{for odd } j, \\ \left(\langle 00|_{\ell_1,\ell_2} + \langle 11|_{\ell_1,\ell_2}\right)(X_{\ell_1})^{g_e}(Z_{\ell_1})^{g_o}, & \text{for even } j. \end{cases}$$

Moreover, using the movement operators (20) and (28), we find the full-fledged fusion operators $(\lambda^j)_{\mathcal{L} \otimes \mathcal{M}}^{\mathcal{N}}$ from (32). The phases of the fusion and movement operators are chosen so that the lattice F-symbols we find below match the usual convention used in the literature.

### 2.3.3  F-symbols

Having defined the fusion operators on the lattice, we are now ready to discuss the lattice F-symbols, which are defined by

$$\sum_f (F_{abc}^d)_e^f \quad : \tag{43}$$

$$(\lambda^j)_{a \otimes e}^d \, \lambda_a^{j-1} \, (\lambda^j)_{b \otimes c}^e = \sum_f (F_{abc}^d)_e^f \, (\lambda^j)_{f \otimes c}^d \, (\lambda^{j-1})_{a \otimes b}^f,$$

for $a, b, c, d, e, f \in \{1, \eta, \eta^e, \eta^o, \mathcal{D}\}$. In general, the lattice F-symbols may depend on $j$. However, as we will see, this is not the case in our situation, and our F-symbols are independent of $j$.

The non-trivial F-symbols, as computed in Appendix D, are

$$= \frac{1}{2} \sum_h \chi(g,h) \qquad , \tag{44}$$

and

$$= \chi(g,h) \qquad , \qquad = \chi(g,h) \qquad , \tag{45}$$

and all the other F-symbols are equal to 1:

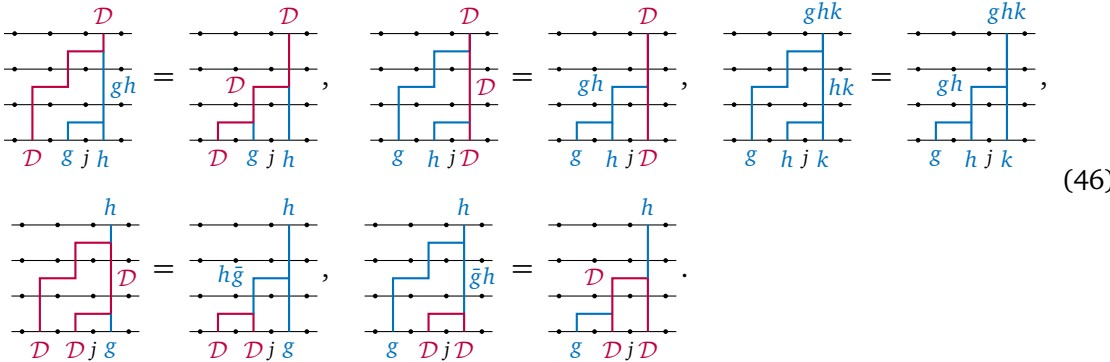

(46)

Above, $g = (g_e, g_o)$, $h = (h_e, h_o) \in \mathbb{Z}_2^e \times \mathbb{Z}_2^o$ where $gh = (g_e + h_e, g_o + h_o)$, and $\chi : \mathbb{Z}_2^e \times \mathbb{Z}_2^o \to \mathbb{Z}_2^e \times \mathbb{Z}_2^o$ is a symmetric bi-character of $\mathbb{Z}_2 \times \mathbb{Z}_2$ given by

$$\chi(g, h) = (-1)^{g_e h_o + g_o h_e} . \tag{47}$$

The lattice F-symbols agrees with those for the Rep($D_8$) fusion category, which is one of the $\mathbb{Z}_2 \times \mathbb{Z}_2$ Tambara-Yamagami categories [74], with the choice of the bicharacter given in (47) and positive Frobenius-Schur indicator. This is to be contrasted with the situation in the critical Ising lattice model, where the algebra for the Kramers-Wannier non-invertible lattice operator mixes with lattice translation [6]. Because of this mixing, the Kramers-Wannier symmetry on the lattice is not described by a fusion category [7]. See [26, 36, 56, 72, 75–79] for more discussions on the mixing between non-invertible symmetries and lattice translations.

Finally, as shown in Appendix D, the on-site fusion operators obey similar relations as (43) with the same F-symbols:

$$\sum_f (F_{abc}^d)_e^f \qquad : \qquad m_{a \otimes e}^d m_{b \otimes c}^e = \sum_f (F_{abc}^d)_e^f m_{f \otimes c}^d m_{a \otimes b}^f . \tag{48}$$

A similar computation of the F-symbols was performed in [80, 81] for the case of modular tensor categories in 2+1d and anomalous invertible symmetries in 1+1d. They derived the F-symbols through the fusion of finite-mass domain-wall excitations rather than the fusion of topological defects. Their approach involved choosing a Hamiltonian that spontaneously breaks the symmetry and studying domain-wall states.

# 3  Gauging a non-invertible symmetry

In this section, we extend the procedure outlined in [56] to the case of gauging non-invertible symmetries on a spin chain of qubits.

In continuum field theory, when gauging an ordinary finite group symmetry $G$, there are multiple inequivalent options, specified by a choice of the subgroup $H \subseteq G$ and a discrete torsion class in $H^2(H, U(1))$. For gauging a finite non-invertible symmetry described by a fusion category, these choices are generalized to the choice of an algebra object $\mathcal{A}$ with a multiplication morphism $m$ [19].

Rep($D_8$) symmetry is the simplest gaugeable fusion category symmetry that can be realized on a spin chain of qubits. There are 11 ways to gauge Rep($D_8$), as summarized in Table 2

of [24] (see also [23]). Out of these 11 ways to gauge $\mathrm{Rep}(D_8)$, 5 of them correspond to gauging ordinary invertible subgroups, and the other 6 correspond to gauging non-group-like objects. Here, we focus on gauging the algebra object

$$\mathcal{A} = 1 \oplus \eta \oplus \mathcal{D}, \tag{49}$$

where $\eta$ is one of the $\mathbb{Z}_2$ defects and $\mathcal{D}$ is the non-invertible defect. This algebra object is not maximal in the sense that it does not contain every object of $\mathrm{Rep}(D_8)$. In contrast to a subgroup, the topological defects in algebra objects do not necessarily form a subcategory, i.e., they need not be closed under fusion. For instance, $\mathcal{D} \otimes \mathcal{D}$ involves defects $\eta^{\mathrm{e}}, \eta^{\mathrm{o}}$ which are not included in $\mathcal{A}$. This specific choice of $\mathcal{A}$ serves as the simplest nontrivial example of gauging a non-invertible lattice symmetry.

In Section 3.1, we summarize our final gauging procedure for gauging $\mathcal{A} = 1 \oplus \eta \oplus \mathcal{D}$. In Section 3.2, we provide details of the gauging.

## 3.1  Summary of the gauging procedure

Gauging is a procedure that starts with a Hamiltonian $H$ that commutes with $\mathrm{Rep}(D_8)$ symmetry operators and results in the Hamiltonian $H_{\mathrm{gauged}}$ of the gauged theory. The Hilbert space of the gauged theory is defined by the original degrees of freedom plus the gauge fields subject to Gauss's law constraints. Below is a short summary and the final results.

1. **Gauss's laws**: We introduce dynamical gauge fields for $\mathcal{A} = 1 \oplus \eta \oplus \mathcal{D}$ subject to Gauss's law constraints. Namely, we insert qubits $\sigma^{x,y,z}_{j-\frac{1}{2}}$ and $\tau^{x,y,z}_j$ per each link and site as the gauge degrees of freedom and impose Gauss's law constraints:

$$P_j = \left( \frac{1 + \tau^z_{j-1} \sigma^x_{j-\frac{1}{2}} \tau^z_j}{2} \right) \left( \frac{1 + \sigma^z_{j-\frac{1}{2}} \tau^x_j X_j \sigma^z_{j+\frac{1}{2}}}{2} \right) = 1, \tag{50}$$

where $X_j$ is the Pauli-$X$ operator acting on the original qubit of site $j$.

2. **Symmetric local operators**: Write the Hamiltonian as a sum of local terms invariant under the $\mathrm{Rep}(D_8)$ non-invertible symmetry. For instance, the simplest such terms are:

$$H = h_0 \sum_j X_j + h_1 \sum_j Z_{j-1} Z_{j+1}(1 + X_j) + h_2 \sum_j Z_{j-2} Z_{j+2}(1 + X_{j-1} X_{j+1}) + \dots \tag{51}$$

3. **Coupling to gauge fields**: We couple each symmetric term in the Hamiltonian to dynamical gauge fields $\sigma^{x,y,z}_{j-\frac{1}{2}}$ and $\tau^{x,y,z}_j$:

$$\tilde{H} = h_0 \sum_j X_j + h_1 \sum_j Z_{j-1} Z_{j+1}(1 + X_j) \left( \sigma^x_{j-\frac{1}{2}} \frac{1 - i\sigma^z_{j-\frac{1}{2}}}{\sqrt{2}} \sigma^x_{j+\frac{1}{2}} \frac{1 + i\tau^x_j \sigma^z_{j+\frac{1}{2}}}{\sqrt{2}} \right)$$

$$+ h_2 \sum_j Z_{j-2} Z_{j+2}(1 + X_{j-1} X_{j+1}) \left( \sigma^x_{j-\frac{3}{2}} \frac{1 - i\sigma^z_{j-\frac{3}{2}}}{\sqrt{2}} \sigma^x_{j-\frac{1}{2}} \frac{1 + i\tau^x_{j-1} \sigma^z_{j-\frac{1}{2}}}{\sqrt{2}} \right. \tag{52}$$

$$\left. \times \sigma^x_{j+\frac{1}{2}} \frac{1 - i\sigma^z_{j+\frac{1}{2}}}{\sqrt{2}} \sigma^x_{j+\frac{3}{2}} \frac{1 + i\tau^x_{j+1} \sigma^z_{j+\frac{3}{2}}}{\sqrt{2}} (\tau^x_{j-1} \tau^x_j)^{\frac{1 - X_{j-1}}{2}} \right) + \dots$$

4. **"Gauge fixing"**: Rewrite $\tilde{H}$ in terms of gauge invariant operators $\tilde{X}_j \equiv X_j$ and $\tilde{Z}_j \equiv Z_j \tau^z_j$, which commute with $P_j$. The final Hamiltonian of the gauged theory is

$$
\begin{aligned}
H_{\text{gauged}} = {} & h_0 \sum_j \tilde{X}_j + h_1 \sum_j \tilde{Z}_{j-1} \tilde{Z}_{j+1} (1 + \tilde{X}_j) \\
& + h_2 \sum_j \tilde{Z}_{j-2} \tilde{Z}_{j+2} (1 + \tilde{X}_{j-1} \tilde{X}_{j+1}) (\tilde{X}_j)^{\frac{1-\tilde{X}_{j-1}}{2}} + \dots
\end{aligned}
\tag{53}
$$

The gauging procedure can be summarized by a gauging map. If we relabel $\tilde{X}_j, \tilde{Z}_j$ as $X_j, Z_j$ in the final Hamiltonian, this map is implemented by one of the non-invertible cosine operators $\mathsf{L}_{\frac{\pi}{4}}$ in the following sense:

$$
\begin{aligned}
& \mathsf{L}_{\frac{\pi}{4}} X_j = X_j \mathsf{L}_{\frac{\pi}{4}}\,, \\
& \mathsf{L}_{\frac{\pi}{4}} Z_{j-1} Z_{j+1} (1 + X_j) = Z_{j-1} Z_{j+1} (1 + X_j) \mathsf{L}_{\frac{\pi}{4}}\,, \\
& \mathsf{L}_{\frac{\pi}{4}} Z_{j-2} Z_{j+2} (1 + X_{j-1} X_{j+1}) = Z_{j-2} Z_{j+2} (1 + X_{j-1} X_{j+1}) (X_j)^{\frac{1-X_{j-1}}{2}} \mathsf{L}_{\frac{\pi}{4}}\,.
\end{aligned}
\tag{54}
$$

(See Appendix B.3 for a derivation.) Similarly, $\mathsf{L}_{\frac{3\pi}{4}} = \eta^e \mathsf{L}_{\frac{\pi}{4}} = \eta^o \mathsf{L}_{\frac{\pi}{4}}$ implements the same gauging map on the $\text{Rep}(D_8)$ symmetric operators. Note that the first two terms, $h_0$ and $h_1$, are invariant under gauging $\mathcal{A}$, but the $h_2$ term changes. The invariance under gauging $\mathcal{A}$ implies that the model enjoys an even larger fusion category symmetry, which was discussed in [27]. For our Hamiltonian with $h_2 = 0$, the enhanced symmetry associated with the invariance under gauging $\mathcal{A}$ is the $\text{Rep}(D_{16})$ fusion category, which is generated by the $\text{Rep}(D_8)$ operators together with the gauging maps $\mathsf{L}_{\frac{\pi}{4}}, \mathsf{L}_{\frac{3\pi}{4}}$:

$$
\begin{aligned}
& (\mathsf{L}_{\frac{\pi}{4}})^2 = (\mathsf{L}_{\frac{3\pi}{4}})^2 = 1 + \eta + \mathsf{D}\,, \\
& \mathsf{D}^2 = 1 + \eta^e + \eta^o + \eta\,, \\
& \mathsf{D}\mathsf{L}_{\frac{\pi}{4}} = \mathsf{L}_{\frac{\pi}{4}} \mathsf{D} = \mathsf{D}\mathsf{L}_{\frac{3\pi}{4}} = \mathsf{L}_{\frac{3\pi}{4}} \mathsf{D} = \mathsf{L}_{\frac{\pi}{4}} + \mathsf{L}_{\frac{3\pi}{4}}\,, \\
& \mathsf{L}_{\frac{\pi}{4}} \mathsf{L}_{\frac{3\pi}{4}} = \mathsf{L}_{\frac{3\pi}{4}} \mathsf{L}_{\frac{\pi}{4}} = \eta^e + \eta^o + \mathsf{D}\,.
\end{aligned}
\tag{55}
$$

Recall that $\mathsf{L}_{\frac{\pi}{2}} = \mathsf{D}$, $(\mathsf{L}_\theta)^\dagger = \mathsf{L}_\theta$, $\mathsf{L}_\pi = \eta^e \eta^o$, and $\eta = \eta^e \eta^o$. See Section 4.1 for the discussion of more general gauging maps. This matches with the continuum result, where it is known that the entire branch of the $c = 1$ orbifold CFT is invariant under gauging $\mathcal{A}$ [24] and has an enhanced $\text{Rep}(D_{16})$ fusion category symmetry (which is part of the continuous cosine symmetry). This $\text{Rep}(D_{16})$ fusion category is analogous to the Kramers-Wannier symmetry of any model that is invariant under gauging an ordinary $\mathbb{Z}_2$ symmetry (see Appendix A.1).

## 3.2 Gauging $\mathcal{A} = 1 \oplus \eta \oplus \mathcal{D}$

Here, we derive the gauging procedure outlined in Section 3.1. In Appendix A, we present a similar gauging procedure for the case of gauging an ordinary, invertible $\mathbb{Z}_2$ global symmetry, using the language of algebra objects and topological defects. The readers are encouraged to read that appendix as a warm-up.

### 3.2.1 Algebra object on the lattice

First, we find the topological defect associated with the algebra object $\mathcal{A}$ to be inserted on all links for gauging. For simplicity, we consider the Hamiltonian in (6), but the following discussion on the movement and fusion operators apply more generally to any $\text{Rep}(D_8)$ symmetric Hamiltonian (see Section 3.2.3).

The defect Hamiltonian for $\mathcal{A} = 1 \oplus \eta \oplus \mathcal{D}$ is given by

$$
\begin{aligned}
H_{\mathcal{A}}^{(1,2)} = h_0 \sum_j X_j + h_1 \sum_{j \neq 1,2} Z_{j-1} Z_{j+1} (1 + X_j) \\
+ h_1 (-i Y_\ell)^{\frac{1-Z_a}{2}} Z_\ell Z_0 Z_2 (1 + X_1) + h_1 Z_\ell Z_1 Z_3 (1 + X_2).
\end{aligned}
\tag{56}
$$

This expression is derived from first decomposing $\mathcal{A}$ into $1 \oplus \eta$ and $\mathcal{D}$, which leads to

$$
H_{\mathcal{A}}^{(1,2)} = |0\rangle\langle 0|_a \otimes H_{1\oplus\eta}^{(1,2)} + |1\rangle\langle 1|_a \otimes H_{\mathcal{D}}^{(1,2)} = \frac{1+Z_a}{2} \otimes H_{1\oplus\eta}^{(1,2)} + \frac{1-Z_a}{2} \otimes H_{\mathcal{D}}^{(1,2)},
\tag{57}
$$

where, $H_{\mathcal{D}}^{(1,2)}$ is given in (25), and

$$
\begin{aligned}
H_{1\oplus\eta}^{(1,2)} &= |0\rangle\langle 0|_\ell \otimes H + |1\rangle\langle 1|_\ell \otimes H_\eta^{(1,2)} \\
&= h_0 \sum_j X_j + h_1 \sum_{j \neq 1,2} Z_{j-1} Z_{j+1} (1 + X_j) \\
&+ h_1 Z_\ell Z_0 Z_2 (1 + X_1) + h_1 Z_\ell Z_1 Z_3 (1 + X_2).
\end{aligned}
\tag{58}
$$

Note that there is a four-dimensional Hilbert space localized on the defect $\mathcal{A}$ described by the qubits $|0\rangle_a, |1\rangle_a$ and $|0\rangle_\ell, |1\rangle_\ell$. These qubits are acted upon by the Pauli operators $X_a, Z_a$ and $X_\ell, Z_\ell$, respectively. Projecting to the $Z_a = +1$ and $Z_a = -1$ subspaces, respectively, corresponds to the sub-defects $1 \oplus \eta$ and $\mathcal{D}$ of $\mathcal{A}$. Moreover, in the subspace $Z_a = +1$, $Z_\ell = +1$ and $Z_\ell = -1$ corresponds to 1 and $\eta$, respectively.

The defect $\mathcal{A}$ is topological and can be moved with the unitary movement operator

$$
\lambda_{\mathcal{A}}^j = \frac{1+Z_a}{2} \left( \frac{1+Z_\ell}{2} + \frac{1-Z_\ell}{2} \lambda_\eta^j \right) + \frac{1-Z_a}{2} \lambda_{\mathcal{D}}^j = (-X_j)^{\frac{1-Z_\ell}{2}} (Z_\ell H_\ell)^{\frac{1-Z_a}{2}}.
\tag{59}
$$

Above, we have used (20) and (28).

The next step is to find the algebra multiplication of $\mathcal{A}$ on the lattice, which is a particular fusion operator that fuses two adjacent $\mathcal{A}$ defects into a single $\mathcal{A}$ defect. Like other fusion operators, we implement the fusion in two steps by first moving the defects together and fusing them afterwards. Namely, we define

$$
(\lambda^j)_{\mathcal{A}_1 \otimes \mathcal{A}_2}^{\mathcal{A}_2} = \quad \cdots \quad = \quad \cdots \quad = m_{\mathcal{A}_1 \otimes \mathcal{A}_2}^{\mathcal{A}_2} \lambda_{\mathcal{A}_1}^j,
\tag{60}
$$

where the on-site fusion operator $m_{\mathcal{A}_1 \otimes \mathcal{A}_2}^{\mathcal{A}_2}$ does not act on the original degrees of freedom $(X_j, Z_j)$ and only acts on the defect degrees of freedom described by qubits $a_1, a_2, \ell_1, \ell_2$. (Recall that we use the subscripts of $\mathcal{A}_i$ to distinguish their associated qubits labeled by $a_i, \ell_i$. However, we emphasize that $\mathcal{A}_i$ with $i = 1, 2$ are two copies of the same algebra object.)

The fusion $\mathcal{A}_1 \otimes \mathcal{A}_2 \to \mathcal{A}_2$ consists of 10 fusion channels, associated with the on-site fusion operators $m_{1\otimes 1}^1, m_{1\otimes\eta}^\eta, m_{\eta\otimes 1}^\eta, m_{\eta\otimes\eta}^1, m_{1\otimes\mathcal{D}_2}^{\mathcal{D}_2}, m_{\eta\otimes\mathcal{D}_2}^{\mathcal{D}_2}, m_{\mathcal{D}_1\otimes 1}^{\mathcal{D}_2}, m_{\mathcal{D}_1\otimes\eta}^{\mathcal{D}_2}, m_{\mathcal{D}_1\otimes\mathcal{D}_2}^1, m_{\mathcal{D}_1\otimes\mathcal{D}_2}^\eta$. According to [24] (which uses the same convention of F-symbols as we do), $m_{\mathcal{A}_1\otimes\mathcal{A}_2}^{\mathcal{A}_2}$ is a sum of these 10

on-site fusion operators with the same coefficient 1/2. This led us to the following proposal for the on-site fusion operator for $\mathcal{A}$ on the lattice:

$$2m^{\mathcal{A}_2}_{\mathcal{A}_1 \otimes \mathcal{A}_2} =$$

$$\left[ \left( m^1_{1\otimes 1}|0\rangle\langle 0|_{\ell_2} + m^\eta_{1\otimes\eta}|1\rangle\langle 1|_{\ell_2} \right)\langle 0|_{\ell_1} + \left( m^\eta_{\eta\otimes 1}|1\rangle\langle 0|_{\ell_2} + m^1_{\eta\otimes\eta}|0\rangle\langle 1|_{\ell_2} \right)\langle 1|_{\ell_1} \right]|0\rangle_{a_2}\langle 00|_{a_1 a_2}$$

$$+ \left( m^{\mathcal{D}_2}_{1\otimes\mathcal{D}_2}\langle 0|_{\ell_1} + m^{\mathcal{D}_2}_{\eta\otimes\mathcal{D}_2}\langle 1|_{\ell_1} \right)|1\rangle_{a_2}\langle 01|_{a_1 a_2} + \left( m^{\mathcal{D}_2}_{\mathcal{D}_1\otimes 1}\langle 0|_{\ell_2} + m^{\mathcal{D}_2}_{\mathcal{D}_1\otimes\eta}\langle 1|_{\ell_2} \right)|1\rangle_{a_2}\langle 10|_{a_1 a_2}$$

$$+ \left( |0\rangle_{\ell_2} m^1_{\mathcal{D}_1\otimes\mathcal{D}_2} + |1\rangle_{\ell_2} m^\eta_{\mathcal{D}_1\otimes\mathcal{D}_2} \right)|0\rangle_{a_2}\langle 11|_{a_1 a_2}. \qquad (61)$$

Using the explicit expression for the on-site fusion operators (42), this can be dramatically simplified as (see Appendix E)

$$m^{\mathcal{A}_2}_{\mathcal{A}_1 \otimes \mathcal{A}_2} = \quad\cdots\cdots\quad = \frac{1}{2}\left( \langle 0|_{a_1} + X_{a_2}\langle 1|_{a_1} \right)\left( \langle 0|_{\ell_1} + Z_{a_2}X_{\ell_2}Z_{\ell_2}\langle 1|_{\ell_1} \right). \qquad (62)$$

In Appendix E, we show that it satisfies associativity

$$\cdots \quad = \quad \cdots \quad : \quad m^{\mathcal{A}_3}_{\mathcal{A}_1\otimes\mathcal{A}_3} m^{\mathcal{A}_3}_{\mathcal{A}_2\otimes\mathcal{A}_3} = m^{\mathcal{A}_3}_{\mathcal{A}_2\otimes\mathcal{A}_3} m^{\mathcal{A}_2}_{\mathcal{A}_1\otimes\mathcal{A}_2}, \qquad (63)$$

The Hermitian conjugation of $m^{\mathcal{A}_2}_{\mathcal{A}_1 \otimes \mathcal{A}_2}$ is

$$(m^{\mathcal{A}_2}_{\mathcal{A}_1 \otimes \mathcal{A}_2})^\dagger = \quad\cdots\cdots\quad = \frac{1}{2}\left( |0\rangle_{\ell_1} + Z_{\ell_2}X_{\ell_2}Z_{a_2}|1\rangle_{\ell_1} \right)\left( |0\rangle_{a_1} + X_{a_2}|1\rangle_{a_1} \right), \qquad (64)$$

which is a co-multiplication that satisfies the co-associativity condition given by the Hermitian conjugate of (63). Together, $m^{\mathcal{A}_2}_{\mathcal{A}_1 \otimes \mathcal{A}_2}$ and its Hermitian conjugate give the explicit lattice realization of the multiplication and co-multiplication for the algebra object. They satisfy all the axioms of separable symmetric Frobenius algebras [15, 19, 82–84], which include the Frobenius condition

$$\cdots \quad = \quad \cdots \quad = \quad \cdots \quad : \qquad (65)$$

$$\left( m^{\mathcal{A}}_{\mathcal{A}_1\otimes\mathcal{A}_2} \right)^\dagger m^{\mathcal{A}}_{\mathcal{A}_1\otimes\mathcal{A}_2} = m^{\mathcal{A}_2}_{\mathcal{A}\otimes\mathcal{A}_2}\left( m^{\mathcal{A}_1}_{\mathcal{A}_1\otimes\mathcal{A}} \right)^\dagger = m^{\mathcal{A}_1}_{\mathcal{A}_1\otimes\mathcal{A}}\left( m^{\mathcal{A}_2}_{\mathcal{A}\otimes\mathcal{A}_2} \right)^\dagger,$$

and the separability condition

$$m^{\mathcal{A}}_{\mathcal{A}_1 \otimes \mathcal{A}_2} \left( m^{\mathcal{A}}_{\mathcal{A}_1 \otimes \mathcal{A}_2} \right)^{\dagger} = \mathbb{1}_{\mathcal{A}} . \tag{66}$$

The separability condition implies that $\left( m^{\mathcal{A}}_{\mathcal{A}_1 \otimes \mathcal{A}_2} \right)^{\dagger} m^{\mathcal{A}}_{\mathcal{A}_1 \otimes \mathcal{A}_2}$ is a projection operator. This fact will be crucial in constructing a generalized Gauss law for the non-invertible gauging associated with algebra $\mathcal{A}$.

Furthermore, there is a unit and co-unit of the algebra given by

$$u_{\mathcal{A}} = 2|0\rangle_{\mathrm{a}}|0\rangle_{\ell} , \quad \text{and} \quad (u_{\mathcal{A}})^{\dagger} = 2\langle 0|_{\mathrm{a}}\langle 0|_{\ell} . \tag{67}$$

The unit and co-unit satisfy

$$m^{\mathcal{A}_2}_{\mathcal{A}_1 \otimes \mathcal{A}_2} u_{\mathcal{A}_1} = m^{\mathcal{A}_2}_{\mathcal{A}_2 \otimes \mathcal{A}_1} u_{\mathcal{A}_1} = \mathbb{1}_{\mathcal{A}_2} , \tag{68}$$

and the Hermitian conjugate of this relation, where $\mathbb{1}_{\mathcal{A}_2} = \mathbb{1}_{\mathrm{a}_2} \otimes \mathbb{1}_{\ell_2}$.

There is one last axiom known as the symmetric condition

$$= \quad , \tag{69}$$

which involves the fusion of $\mathcal{A}$ with its dual $\mathcal{A}^*$:

$$e_{\mathcal{A}} = \langle 00|_{\mathrm{a}_1,\ell_1} \langle 00|_{\mathrm{a}_2,\ell_2} \, m^1_{1 \otimes 1} + \langle 01|_{\mathrm{a}_1,\ell_1} \langle 01|_{\mathrm{a}_2,\ell_2} \, m^1_{\eta \otimes \eta^*} + \langle 1|_{\mathrm{a}_1} \langle 1|_{\mathrm{a}_2} \otimes m^1_{\mathcal{D} \otimes \mathcal{D}^*} , \tag{70}$$

where $e_{\mathcal{A}}$ is known as the evaluation morphism in the context of fusion categories. We have used $\mathrm{a}_1, \ell_1$ and $\mathrm{a}_2, \ell_2$ to label the qubits for $\mathcal{A}$ and $\mathcal{A}^*$ defects, respectively. The dual $\mathcal{L}^*$ of a defect $\mathcal{L}$ is such that $\mathcal{L} \otimes \mathcal{L}^* = 1 \oplus \cdots$ includes the trivial defect. In our case, each individual defect inside $\mathcal{A}$ (that is, $\eta$ and $\mathcal{D}$) is self-dual and we will identify $\mathcal{A}$ with $\mathcal{A}^*$. Moreover, the evaluation morphism is equal to

$$e_{\mathcal{A}} = \quad m = (u_{\mathcal{A}_2})^{\dagger} m^{\mathcal{A}_2}_{\mathcal{A}_1 \otimes \mathcal{A}_2} = \left( \langle 00|_{\mathrm{a}_1,\mathrm{a}_2} + \langle 11|_{\mathrm{a}_1,\mathrm{a}_2} \right) \left( \langle 00|_{\ell_1,\ell_2} - Z_{\mathrm{a}_2} \langle 11|_{\ell_1,\ell_2} \right) , \tag{71}$$

which can be verified using (42). The symmetric condition then follows from the above relation and the previous conditions. Alternatively, the symmetric condition follows from the

haploid condition [84], which states that the identity defect appears with multiplicity one inside $\mathcal{A} = 1 \oplus \eta \oplus \mathcal{D}$.

Finally, as we show in Appendix E.3, the fusion operator $(\lambda^j)^{\mathcal{A}_2}_{\mathcal{A}_1 \otimes \mathcal{A}_2}$ obeys a similar set of axioms for Frobenius algebras.

### 3.2.2  Gauss's law of $\mathcal{A}$

The fusion operator $(\lambda^j)^{\mathcal{A}_2}_{\mathcal{A}_1 \otimes \mathcal{A}_2}$ satisfies a separability condition similar to its onsite counterpart in (66):

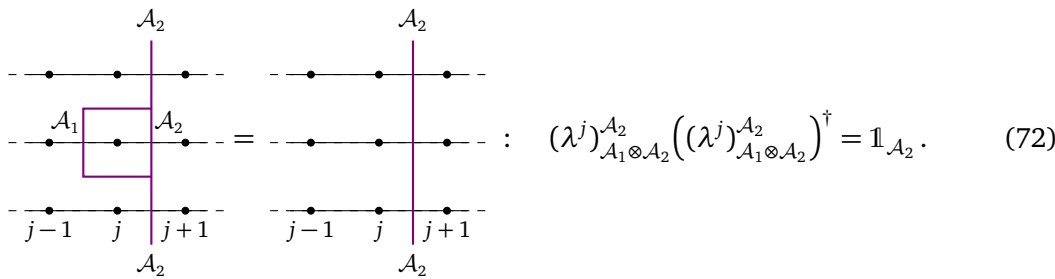

$$(\lambda^j)^{\mathcal{A}_2}_{\mathcal{A}_1 \otimes \mathcal{A}_2} \left( (\lambda^j)^{\mathcal{A}_2}_{\mathcal{A}_1 \otimes \mathcal{A}_2} \right)^\dagger = \mathbb{1}_{\mathcal{A}_2}. \tag{72}$$

Gauss's law operator at site $j$ is defined in terms of the fusion operator as

$$P_j = \cdots = \left( (\lambda^j)^{\mathcal{A}_2}_{\mathcal{A}_1 \otimes \mathcal{A}_2} \right)^\dagger \left( (\lambda^j)^{\mathcal{A}_2}_{\mathcal{A}_1 \otimes \mathcal{A}_2} \right) = (\lambda^j_{\mathcal{A}})^\dagger \left( m^{\mathcal{A}}_{\mathcal{A}_1 \otimes \mathcal{A}_2} \right)^\dagger m^{\mathcal{A}}_{\mathcal{A}_1 \otimes \mathcal{A}_2} \lambda^j_{\mathcal{A}}. \tag{73}$$

Separability (72) guarantees that $P_j$ is a projection operator $P_j^2 = P_j$. Note that Gauss's law operators at different sites commute with each other, i.e., $P_j P_{j'} = P_{j'} P_j$. Imposing Gauss's law amounts to setting $P_j = 1$ at every site.

Using (59) and (62), the Gauss law operator can be simplified as

$$P_j = \frac{1 + H_{\ell_{j-\frac{1}{2}}} X_{a_{j-\frac{1}{2}}} X_{a_{j+\frac{1}{2}}}}{2} \frac{1 - X_j Y_{\ell_{j-\frac{1}{2}}} Y_{\ell_{j+\frac{1}{2}}} Z_{a_{j+\frac{1}{2}}}}{2}. \tag{74}$$

To emphasize the difference between the original spin and gauge degrees of freedom, we rename the operators acting on the $\ell$-qubits and a-qubits by $\sigma$ and $\tau$ Pauli operators as:

$$\sigma^x_{j-\frac{1}{2}} = H_{\ell_{j-\frac{1}{2}}}, \qquad \sigma^z_{j-\frac{1}{2}} = Y_{\ell_{j-\frac{1}{2}}}, \qquad \tau^x_j = -Z_{a_{j+\frac{1}{2}}}, \qquad \tau^z_j = X_{a_{j+\frac{1}{2}}}. \tag{75}$$

The Gauss law operator, in terms of these new variables, is

$$P_j = \frac{1 + \mathcal{G}_{j-\frac{1}{2}}}{2} \frac{1 + \mathcal{G}_j}{2}, \qquad \mathcal{G}_{j-\frac{1}{2}} = \tau^z_{j-1} \sigma^x_{j-\frac{1}{2}} \tau^z_j, \qquad \mathcal{G}_j = \sigma^z_{j-\frac{1}{2}} \left( \tau^x_j X_j \right) \sigma^z_{j+\frac{1}{2}}. \tag{76}$$

Note that all $\mathcal{G}_j$ and $\mathcal{G}_{j'-\frac{1}{2}}$ commute with each another. Setting $P_j = 1$ enforces $\mathcal{G}_{j-\frac{1}{2}} = \mathcal{G}_j = 1$ at every $j$. The two qubits, acted by the $\sigma$ and $\tau$ operators can be viewed as the "gauge fields" for gauging $\mathcal{A}$.

### 3.2.3  Symmetric operators of Rep($D_8$)

Thus far, our discussion of the movement operators, fusion operators, and Gauss's law for the non-invertible symmetry is independent of the choice of the Hamiltonian. Here we identify all the Rep($D_8$)-symmetric local operators and write symmetric Hamiltonians in terms of such operators. Following the terminology of [85, 86], we refer to the algebra of all local symmetric operators as the bond algebra of Rep($D_8$).

We first note that the bond algebra of the $\mathbb{Z}_2^e \times \mathbb{Z}_2^o$ symmetry is generated by $X_j$ and $Z_j Z_{j+2}$ for all $j$. From (9), we then find the following list of Rep($D_8$)-symmetric local operators:[9]

$$X_j, \quad Z_{j-1}Z_{j+1}(1+X_j), \quad Z_{j-2}Z_{j+2}(1+X_{j-1}X_{j+1}), \quad Z_{j-3}Z_{j+3}(1+X_{j-2}X_jX_{j+2}), \ldots \quad (77)$$

It is straightforward to see that these operators form a complete basis of symmetric local operators, in the sense that the bond algebra of Rep($D_8$) is generated by

$$X_j \quad \text{and} \quad \left(\prod_{l=0}^{k} Z_{j-(k-2l)}\right)\left(1 + \prod_{l=0}^{k-1} X_{j-(k-1-2l)}\right), \quad (78)$$

for integer $k \geq 1$.

Therefore, any Rep($D_8$)-symmetric Hamiltonian with range-four interactions has the form

$$H = h_0 \sum_j X_j + h_1 \sum_j Z_{j-1}Z_{j+1}(1+X_j) + h_2 \sum_j Z_{j-2}Z_{j+2}(1+X_{j-1}X_{j+1}) + \ldots, \quad (79)$$

where the terms in '$\cdots$' are more complicated functions of $X_j$, $Z_{j-1}Z_{j+1}(1+X_j)$, and $Z_{j-2}Z_{j+2}(1+X_{j-1}X_{j+1})$. Moreover, the coupling constant $h_0, h_1, h_2$ can, in general, depend on $j$ and break the lattice translation symmetry. In the following, we only study the effect of gauging on the first three terms in the Hamiltonian above.

### 3.2.4  Coupling to dynamical "gauge fields" for $\mathcal{A}$

Next, we couple a symmetric Hamiltonian to the $\mathcal{A}$ gauge degrees of freedom introduced in Section 3.2.2. As explained earlier, this process is equivalent to introducing $\mathcal{A}$ defects on all links. The Hamiltonian for a single $\mathcal{A}$ defect was derived in (56), which involves two extra qubits labeled by $\ell$ and a. Next, we repeat the same construction for every link and find the Hamiltonian with an $\mathcal{A}$ defect on every link:

$$\tilde{H} = h_0 \sum_j X_j + h_1 \sum_j \mathsf{h}_{1,j} + h_2 \sum_j \mathsf{h}_{2,j},$$

$$\mathsf{h}_{1,j} = Z_{j-1}Z_{j+1}(1+X_j)\left(\sigma^x_{j-\frac{1}{2}}\frac{1-i\sigma^z_{j-\frac{1}{2}}}{\sqrt{2}}\sigma^x_{j+\frac{1}{2}}\frac{1+i\tau^x_j\sigma^z_{j+\frac{1}{2}}}{\sqrt{2}}\right),$$

$$\mathsf{h}_{2,j} = Z_{j-2}Z_{j+2}(1+X_{j-1}X_{j+1})\left(\sigma^x_{j-\frac{3}{2}}\frac{1-i\sigma^z_{j-\frac{3}{2}}}{\sqrt{2}}\sigma^x_{j-\frac{1}{2}}\frac{1+i\tau^x_{j-1}\sigma^z_{j-\frac{1}{2}}}{\sqrt{2}}\right.$$

$$\left. \times \sigma^x_{j+\frac{1}{2}}\frac{1-i\sigma^z_{j+\frac{1}{2}}}{\sqrt{2}}\sigma^x_{j+\frac{3}{2}}\frac{1+i\tau^x_{j+1}\sigma^z_{j+\frac{3}{2}}}{\sqrt{2}}(\tau^x_{j-1}\tau^x_j)^{\frac{1-X_{j-1}}{2}}\right). \quad (80)$$

Here, we have expressed the Hamiltonian in terms of the new variables introduced in (75). See Appendix C for more details.

---

[9]Note that $Z_{j-1}Z_{j+1}(1-X_j)$ transforms by a sign under D. Therefore, a bilinear formed by such terms is a linear combination of the generators in (77). For instance, $Z_{j-2}Z_{j+2}(1-X_{j-1})(1-X_{j+1})$ is equal to $2Z_{j-2}Z_{j+2}(1+X_{j-1}X_{j+1}) - Z_{j-2}Z_{j+2}(1+X_{j-1})(1+X_{j+1})$.

To summarize, we have introduced a new qubit on every link (acted by $\sigma_{j+\frac{1}{2}}^{x,y,z}$) and a new qubit on every site (acted by $\tau_j^{x,y,z}$) and couple them to the original Hamiltonian. These new degrees of freedom can be interpreted as the "gauge fields" for gauging the non-invertible symmetry $\mathcal{A}$. The introduction of these gauge field degrees of freedom gives an enlarged, $2^{3L}$-dimensional Hilbert space $\widetilde{\mathcal{H}}$, on which the Hamiltonian $\tilde{H}$ is defined. We further impose Gauss's law $P_j = 1$, which enforces two conditions $\mathcal{G}_{j-\frac{1}{2}} = \mathcal{G}_j = 1$ for every $j$. This reduces the final Hilbert space to $\mathcal{H}_{\text{gauge}}$, which is $2^L$-dimensional.

By construction, every local term in the Hamiltonian $\tilde{H}$ is gauge-invariant, i.e., it commutes with the projection operators $P_j = \frac{1+\mathcal{G}_{j-1/2}}{2}\frac{1+\mathcal{G}_j}{2}$ from Gauss' law in (76):

$$\mathsf{h}_{I,j'} P_j = P_j \mathsf{h}_{I,j'}\,. \tag{81}$$

However, while $\mathcal{G}_j$ commutes with all the terms in the Hamiltonian $\tilde{H}$, the other operator $\mathcal{G}_{j-\frac{1}{2}}$ does not commute with $\mathsf{h}_{1,j'}$ and $\mathsf{h}_{2,j'}$ for all $j'$. Instead, we have

$$
\begin{aligned}
\mathcal{G}_{j-\frac{1}{2}}\,\mathsf{h}_{1,j'} &= \begin{cases} \mathsf{h}_{1,j'}\,\mathcal{G}_{j-\frac{1}{2}}\mathcal{G}_j\,, & \text{if } j = j'\,, \\ \mathsf{h}_{1,j'}\,\mathcal{G}_{j-\frac{1}{2}}\,, & \text{if } j \neq j'\,, \end{cases} \\[2mm]
\mathcal{G}_{j-\frac{1}{2}}\,\mathsf{h}_{2,j'} &= \begin{cases} \mathsf{h}_{2,j'}\,\mathcal{G}_{j-\frac{1}{2}}\mathcal{G}_j\,, & \text{if } j = j' \pm 1\,, \\ \mathsf{h}_{2,j'}\,\mathcal{G}_{j-\frac{1}{2}}\,, & \text{if } j \neq j' \pm 1\,. \end{cases}
\end{aligned}
\tag{82}
$$

This is in sharp contrast with the gauging of an invertible symmetry, where the corresponding unitary operators from Gauss's law commute with every local term in the Hamiltonian (see, for example, Appendix A.1). Because of (82), it is not possible to define a "gauge transformation" implemented by $\mathcal{G}_{j-\frac{1}{2}}$ that leaves the Hamiltonian invariant.

### 3.2.5  "Gauge fixing"

Here, we "solve" for the Gauss law constraints and rewrite the gauged theory as a model with tensor product Hilbert space:

I.  First, we identify a set of gauge-invariant operators

$$\tilde{X}_j \equiv X_j\,, \qquad \tilde{Z}_j \equiv Z_j\,\tau_j^z\,, \tag{83}$$

which commute with Gauss's law operators in (76):

$$P_j = \frac{1 + \left(\tau_{j-1}^z \sigma_{j-\frac{1}{2}}^x \tau_j^z\right)}{2}\frac{1 + \left(\sigma_{j-\frac{1}{2}}^z \tau_j^x \tilde{X}_j \sigma_{j+\frac{1}{2}}^z\right)}{2}\,. \tag{84}$$

Note that $\tilde{X}_j$ and $\tilde{Z}_j$ form a complete basis of all gauge-invariant observables of the gauged theory. They have the standard commutation relation of decoupled qubits and, therefore, the gauged theory has a tensor product Hilbert space.

II.  Next, we rewrite the Hamiltonian $\tilde{H}$ of (80) in terms of these gauge invariant operators using Gauss's law constraints. In other words, we find the Hamiltonian $H_{\text{gauged}}$ such that it satisfies

$$\tilde{H}\left(\prod_j P_j\right) = H_{\text{gauged}}\left(\prod_j P_j\right)\,, \tag{85}$$

and is written in terms of the gauge invariant operators (83). Therefore, $\tilde{H}$ and $H_{\text{gauged}}$ have the same action on gauge-invariant states $|\psi\rangle$ satisfying $P_j|\psi\rangle = |\psi\rangle$.

**$h_1$-term:** We start with the $h_1$-term in (80), which in terms of $\tilde{X}_j$ and $\tilde{Z}_j$ is

$$\tilde{Z}_{j-1}\tilde{Z}_{j+1}(1+\tilde{X}_j)\tau^z_{j-1}\tau^z_{j+1}\left(\sigma^x_{j-\frac{1}{2}}\frac{1-i\sigma^z_{j-\frac{1}{2}}}{\sqrt{2}}\sigma^x_{j+\frac{1}{2}}\frac{1+i\tau^x_j\sigma^z_{j+\frac{1}{2}}}{\sqrt{2}}\right). \tag{86}$$

Multiplying this term from the right by $P_j P_{j+1}$ and using $\left(\sigma^z_{j-\frac{1}{2}}\tau^x_j\tilde{X}_j\sigma^z_{j+\frac{1}{2}}\right)P_j = P_j$, we obtain:

$$\tilde{Z}_{j-1}\tilde{Z}_{j+1}(1+\tilde{X}_j)\tau^z_{j-1}\tau^z_{j+1}\left(\frac{1+i\sigma^z_{j-\frac{1}{2}}}{\sqrt{2}}\frac{1-i\sigma^z_{j-\frac{1}{2}}\tilde{X}_j}{\sqrt{2}}\tau^z_{j-1}\tau^z_{j+1}\right)P_j P_{j+1} \tag{87}$$
$$= \tilde{Z}_{j-1}\tilde{Z}_{j+1}(1+\tilde{X}_j)P_j P_{j+1}.$$

In the last equality, we have used the projector $(1+\tilde{X}_j)$ to set $\tilde{X}_j = 1$ in the parentheses. The $h_1$-term becomes $\tilde{Z}_{j-1}\tilde{Z}_{j+1}(1+\tilde{X}_j)$, and therefore does not change under gauging!

**$h_2$-term:** The simplest term that does change under gauging is the $h_2$-term in (80). In terms of the gauge invariant operators, it is

$$\tilde{Z}_{j-2}\tilde{Z}_{j+2}\left(1+\tilde{X}_{j-1}\tilde{X}_{j+1}\right)\tau^z_{j-2}\tau^z_{j+2}\left(\tau^x_{j-1}\tau^x_j\right)^{\frac{1-\tilde{X}_{j-1}}{2}}$$
$$\times\left(\frac{1+i\sigma^z_{j-\frac{3}{2}}}{\sqrt{2}}\frac{1-i\tau^x_{j-1}\sigma^z_{j-\frac{1}{2}}}{\sqrt{2}}\frac{1+i\sigma^z_{j+\frac{1}{2}}}{\sqrt{2}}\frac{1-i\tau^x_{j+1}\sigma^z_{j+\frac{3}{2}}}{\sqrt{2}}\right)\sigma^x_{j-\frac{3}{2}}\sigma^x_{j-\frac{1}{2}}\sigma^x_{j+\frac{1}{2}}\sigma^x_{j+\frac{3}{2}}. \tag{88}$$

We again multiply this term on the right by $P_{j-1}P_j P_{j+1}P_{j+2}$ to simplify it. We find

$$\tilde{Z}_{j-2}\tilde{Z}_{j+2}\left(1+\tilde{X}_{j-1}\tilde{X}_{j+1}\right)\left(\sigma^z_{j-\frac{3}{2}}\tilde{X}_{j-1}\tilde{X}_j\sigma^z_{j+\frac{1}{2}}\right)^{\frac{1-\tilde{X}_{j-1}}{2}}$$
$$\times\left(\frac{1+i\sigma^z_{j-\frac{3}{2}}}{\sqrt{2}}\frac{1-i\sigma^z_{j-\frac{3}{2}}\tilde{X}_{j-1}}{\sqrt{2}}\frac{1+i\sigma^z_{j+\frac{1}{2}}}{\sqrt{2}}\frac{1-i\sigma^z_{j+\frac{1}{2}}\tilde{X}_{j+1}}{\sqrt{2}}\right)P_{j-1}P_j P_{j+1}P_{j+2} \tag{89}$$
$$= \tilde{Z}_{j-2}\tilde{Z}_{j+2}\left(1+\tilde{X}_{j-1}\tilde{X}_{j+1}\right)\left(\frac{1+\tilde{X}_{j-1}}{2}+\frac{1-\tilde{X}_{j-1}}{2}\tilde{X}_j\right)P_{j-1}P_j P_{j+1}P_{j+2},$$

where in the last line we have used the projector $\left(1+\tilde{X}_{j-1}\tilde{X}_{j+1}\right)$ and the $P_j$'s. Therefore, the $h_2$-term after gauging becomes $\tilde{Z}_{j-2}\tilde{Z}_{j+2}(1+\tilde{X}_{j-1}\tilde{X}_{j+1})(\tilde{X}_j)^{\frac{1-\tilde{X}_{j-1}}{2}}$.

Putting everything together, the Hamiltonian of the gauged theory is

$$H_{\text{gauged}} = h_0\sum_j\tilde{X}_j + h_1\sum_j\tilde{Z}_{j-1}\tilde{Z}_{j+1}(1+\tilde{X}_j)+h_2\sum_j\tilde{Z}_{j-2}\tilde{Z}_{j+2}(1+\tilde{X}_{j-1}\tilde{X}_{j+1})(\tilde{X}_j)^{\frac{1-\tilde{X}_{j-1}}{2}}. \tag{90}$$

By relabeling $\tilde{X}_j, \tilde{Z}_j$ as $X_j, Z_j$, we have arrived at the final expression of the gauged Hamiltonian on a $2^L$-dimensional tensor product Hilbert space $\mathcal{H}_{\text{gauge}}$. The relabeling provides an isomorphism between the original Hilbert space $\mathcal{H}$ and the final one $\mathcal{H}_{\text{gauge}}$.

## 4 Gauging maps and generalizations

In the previous section, we described gauging $\mathcal{A} = 1\oplus\eta\oplus\mathcal{D}$ of $\text{Rep}(D_8)$ non-invertible symmetry. Our gauging procedure is general and can be applied to any finite non-invertible symmetry. Here, we describe the general gauging procedure using the gauging map.

## 4.1 Gauging maps

Gauging a finite symmetry $\mathcal{C}$, in general, can be described by a *gauging map*

$$\mathsf{G} : \mathcal{H} \to \mathcal{H}_{\text{gauge}} \tag{91}$$

between the Hilbert spaces before and after gauging. $\mathsf{G}$ maps local and $\mathcal{C}$-symmetric Hamiltonians to local Hamiltonians in the sense that

$$\mathsf{G}H = H_{\text{gauged}}\,\mathsf{G}\,, \tag{92}$$

where $H$ and $H_{\text{gauged}}$ denote the Hamiltonians before and after gauging, respectively. In certain special cases (e.g., gauging $\mathbb{Z}_2$ and $\mathcal{A} = 1 \oplus \eta \oplus \mathcal{D}$), the Hilbert spaces $\mathcal{H}_{\text{gauge}}$ and $\mathcal{H}$ are isomorphic and can be identified. However, this is not necessarily true in general.

The defining property of the gauging map is that it satisfies

$$\mathsf{G}^{\dagger}\mathsf{G} = \mathsf{A}\,, \tag{93}$$

where $\mathsf{A}$ is the non-invertible operator associated with the algebra object $\mathcal{A}$ that is being gauged. For the case of gauging $\mathbb{Z}_2$ and $\mathcal{A} = 1 \oplus \eta \oplus \mathcal{D}$ we have $\mathsf{A} = 1 + \eta$ and $\mathsf{A} = 1 + \eta + \mathsf{D}$, respectively. Conversely, a non-invertible operator $\mathsf{G}$ that satisfies (93), and maps symmetric local Hamiltonians to local Hamiltonians, implements the gauging of the algebra object $\mathcal{A}$. We note that the gauging map is not unique and has an ambiguity by composing it with a local unitary transformation from the left. This is the same ambiguity as the one in identifying $\mathcal{H}_{\text{gauge}}$ with $\mathcal{H}$ when they are isomorphic.

For the case of gauging $\mathbb{Z}_2$, the gauging map is described by the Kramers-Wannier operator KW that indeed satisfies $(\text{KW})^{\dagger}\text{KW} = 1 + \eta$ (see Appendix A.1). For the case of Rep($\mathsf{D}_8$), the gauging map can be taken to be the cosine non-invertible operator $\mathsf{L}_{\pi/4}$ (or $\mathsf{L}_{3\pi/4}$) since it satisfies

$$\mathsf{L}_{\pi/4}^{\dagger}\mathsf{L}_{\pi/4} = 1 + \eta + \mathsf{D}\,. \tag{94}$$

(See (17) and (19).) The action of this cosine symmetry on local operators is given in (54). Indeed, it maps the Hamiltonian (51) into (53) (up to relabeling $X_j, Z_j$ with $\tilde{X}_j, \tilde{Z}_j$), thereby, verifying our gauging procedure.

## 4.2 The general gauging procedure

Now, we summarize the general procedure applicable for gauging any algebra object $\mathcal{A}$ of a finite non-invertible symmetry $\mathcal{C}$.

1. Find an algebra object $\mathcal{A}$, its fusion operator $(\lambda^j)_{\mathcal{A}_1 \otimes \mathcal{A}_2}^{\mathcal{A}_2}$, and unit $u_{\mathcal{A}}$ satisfying the lattice version of Frobenius algebra axioms:[10]

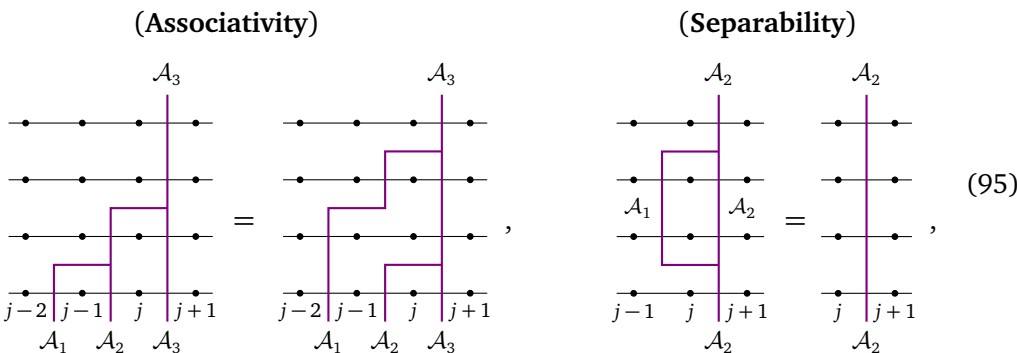

$$\tag{95}$$

---

[10]We assume that the co-multiplication and co-unit are equal to the Hermitian conjugate of the multiplication and unit, respectively.

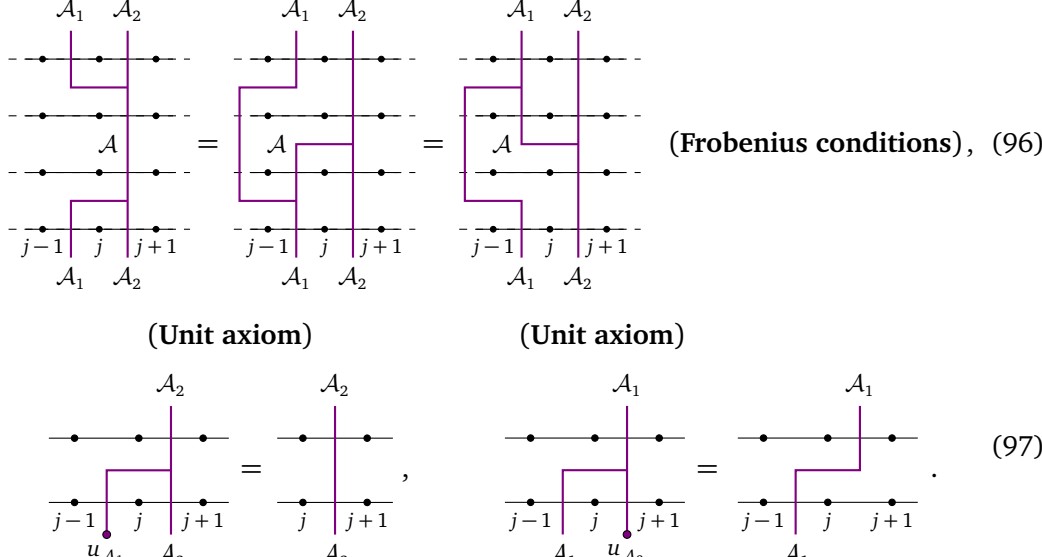

$$\ldots = \ldots = \ldots \qquad \textbf{(Frobenius conditions)}, \quad (96)$$

$$\textbf{(Unit axiom)} \qquad\qquad \textbf{(Unit axiom)}$$

$$\ldots = \ldots , \quad \text{and} \quad \ldots = \ldots . \qquad (97)$$

The axioms for the co-unit are the Hermitian conjugate of the axioms for the unit.

$$\ldots = \ldots \qquad \textbf{(Symmetric condition)}, \qquad (98)$$

where

$$(\lambda^j)^1_{\mathcal{A}\otimes\mathcal{A}^*} = \ldots , \quad \text{and} \quad (\lambda^j)^1_{\mathcal{A}^*\otimes\mathcal{A}} = \ldots \qquad (99)$$

are the 'evaluation' fusion operators of the algebra object $\mathcal{A}$ given by the fusion operators of the underlying non-invertible symmetry defects.[11]

2. Construct (commuting) Gauss's law operators

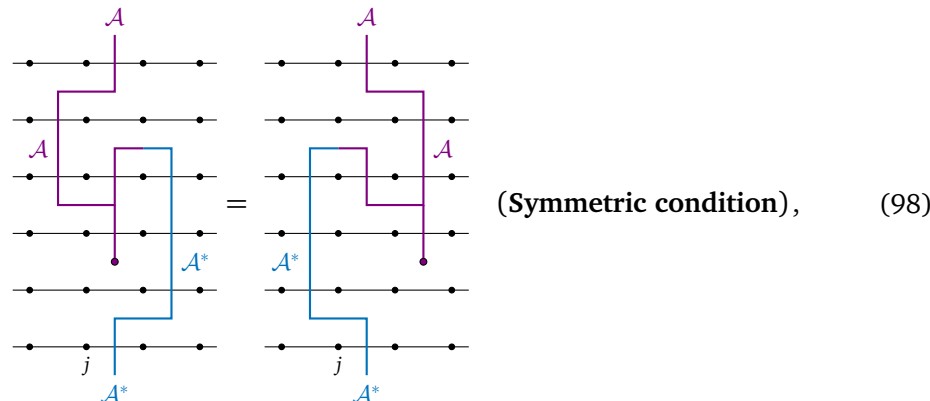

$$P_j \equiv \ldots = \left((\lambda^j)^{\mathcal{A}_{j+\frac{1}{2}}}_{\mathcal{A}_{j-\frac{1}{2}}\otimes\mathcal{A}_{j+\frac{1}{2}}}\right)^\dagger (\lambda^j)^{\mathcal{A}_{j+\frac{1}{2}}}_{\mathcal{A}_{j-\frac{1}{2}}\otimes\mathcal{A}_{j+\frac{1}{2}}} ,$$

---

[11]The symmetric condition follows from the haploid condition [84], which states that the identity defect appears with multiplicity one inside $\mathcal{A} = 1 \oplus \cdots$.

and the gauging map

$$G = \left(\prod_{j=1}^{L} P_j\right) \bigotimes_{j=1}^{L} u_{\mathcal{A}_{j+\frac{1}{2}}} = \ldots \qquad \ldots, \tag{100}$$

where $u_{\mathcal{A}_{j+\frac{1}{2}}}$ is the unit of the algebra. It follows from the associativity, Frobenius, and separability conditions that Gauss's law operators $P_j$ are mutually commuting projection operators. In the diagrammatic presentation above, we have multiplied Gauss's operators in the order $\prod_{j \text{ odd}} P_j \prod_{j \text{ even}} P_j$.

3. Determine the action of the gauging map $G$ on symmetric local operators and use it to identify the gauged Hamiltonian $H_{\text{gauged}}$ satisfying

$$G H = H_{\text{gauged}} G, \quad \text{and} \quad H_{\text{gauged}} P_j = P_j H_{\text{gauged}}. \tag{101}$$

At the level of local operator algebras, the gauging map can be identified as a homomorphism $G : \mathcal{A} \to \mathcal{A}_{\text{gauged}}$ between the symmetric local operator algebra before and after gauging. The local operator algebra of the gauged theory $\mathcal{A}_{\text{gauged}}$ is the subalgebra of $\mathcal{A} \otimes \mathcal{A}_{\text{gauge fields}}$ that commute with Gauss's operators $P_j$ subject to the equivalence relation $\mathcal{O}_1 \sim \mathcal{O}_2 \iff \forall j : \mathcal{O}_1 P_j = \mathcal{O}_2 P_j$. In special cases, $\mathcal{A}$ and $\mathcal{A}_{\text{gauged}}$ can be identified through a "gauge fixing" procedure such as the one in Section 3.2.5. As discussed in Section 3.1, the gauging procedure includes coupling to dynamical gauge fields, imposing Gauss's law constraint, and "gauge fixing". These steps are implemented by the gauging map in one shot by transforming the symmetric local operators directly.

Finally, let us check that (100) indeed satisfies (93). First, we find

$$G^\dagger G = \bigotimes_{j=1}^{L} u^\dagger_{\mathcal{A}_{j+\frac{1}{2}}} \left(\prod_{j=1}^{L} P_j\right) \bigotimes_{j=1}^{L} u_{\mathcal{A}_{j+\frac{1}{2}}} = \cdots \qquad \cdots \tag{102}$$

To simplify this expression, we need the following relation

$$\cdots = \cdots, \tag{103}$$

which follows from the symmetric condition. Using (102) and (103), we find

$$G^\dagger G = \cdots \qquad \cdots = A. \tag{104}$$

Alternatively, by multiplying Gauss's operators in a sequential order, we find

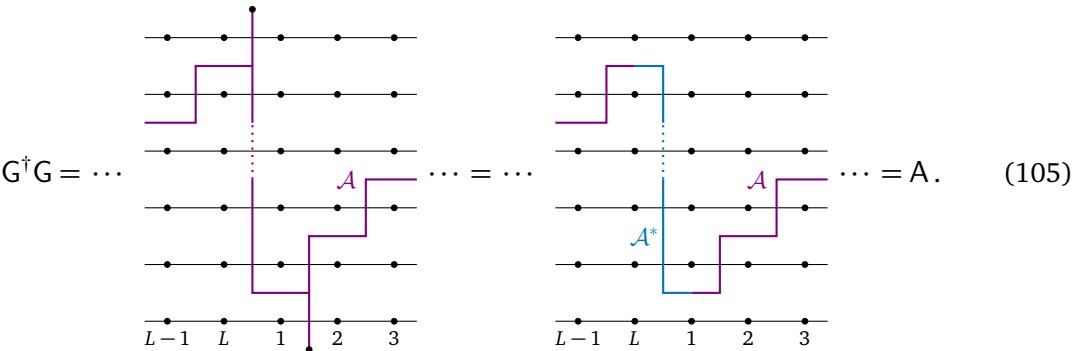

$$G^\dagger G = \cdots \qquad \cdots = \cdots \qquad \cdots = A. \tag{105}$$

## Acknowledgments

We thank Jeongwan Haah, Ryohei Kobayashi, Ho Tat Lam, Da-Chuan Lu, Sal Pace, Nikita Sopenko, Nat Tantivasadakarn, and Yifan Wang for helpful discussions.

**Funding information** SS gratefully acknowledges support from the Sivian Fund and the Paul Dirac Fund at the Institute for Advanced Study, the U.S. Department of Energy grant DE-SC0009988, and the Simons Collaboration on Ultra-Quantum Matter, which is a grant from the Simons Foundation (651444, NS). SHS is supported in part by NSF grant PHY-2210182. XY acknowledges the support from NSF under the grant number DMR-2424315.

## A  Gauging $\mathbb{Z}_2$ symmetry using algebra objects

In this section, we gauge the $\mathbb{Z}_2$ symmetry by first following the ordinary procedure in A.1 and then using the language of algebra objects in A.3. Both methods give the same Gauss's law and the same gauged Hamiltonian. The second method treats the background gauge fields as defects, as we review in A.2, and is generalizable to gauging fusion category symmetries.

### A.1  Review of the ordinary gauging procedure and the Kramers-Wannier operator

We start with the transverse-field Ising Hamiltonian at a generic coupling $h$

$$H = -\sum_{j=1}^{L} \left( h^{-1} Z_{j-1} Z_j + h X_j \right). \tag{A.1}$$

We want to gauge the $\mathbb{Z}_2$ symmetry generated by $\eta = \prod_{j=1}^{L} X_j$.

In the first step, we add an extra qubit on each link $(j-1, j)$, corresponding to the $\mathbb{Z}_2$ gauge field, on which the Pauli operators $\sigma^x_{j-\frac{1}{2}}$ and $\sigma^z_{j-\frac{1}{2}}$ act. The gauged Hamiltonian is

$$\tilde{H} = -\sum_{j=1}^{L} \left( h^{-1} Z_{j-1} \sigma^z_{j-\frac{1}{2}} Z_j + h X_j \right), \tag{A.2}$$

acting on an extended $2^{2L}$-dimensional Hilbert space. The extended system has a $\mathbb{Z}_2^L$ symmetry generated by the local operator

$$\mathcal{G}_j = \sigma^x_{j-\frac{1}{2}} X_j \sigma^x_{j+\frac{1}{2}}, \qquad j = 1, \ldots, L. \tag{A.3}$$

This unitary operator $\mathcal{G}_j$ commutes with every local term in the Hamiltonian $\tilde{H}$.

In the second step of the gauging, we project the extended Hilbert space onto states invariant under the $\mathbb{Z}_2^L$ symmetry by imposing Gauss's law

$$\mathcal{G}_j = 1, \quad j = 1, \dots, L, \tag{A.4}$$

so that we return to a Hilbert space of dimension $2^L$. In particular, $\eta = \prod_j \mathcal{G}_j = 1$ in the gauged Hilbert space.

In the final step, we introduce gauge invariant operators

$$\tilde{X}_{j-\frac{1}{2}} \equiv Z_{j-1} \sigma^z_{j-\frac{1}{2}} Z_j, \qquad \tilde{Z}_{j-\frac{1}{2}} \equiv \sigma^x_{j-\frac{1}{2}}, \tag{A.5}$$

and the effective Hamiltonian is

$$\tilde{H} = -\sum_{j=1}^{L} \left( h \, \tilde{Z}_{j-\frac{1}{2}} \tilde{Z}_{j+\frac{1}{2}} + h^{-1} \tilde{X}_{j-\frac{1}{2}} \right). \tag{A.6}$$

Up to the relabeling $\tilde{X}_{j+\frac{1}{2}} \to X_j$, $\tilde{Z}_{j+\frac{1}{2}} \to Z_j$, this is the same as the original Hamiltonian with $h$ replaced by $h^{-1}$.

To summarize, the gauging maps the $\mathbb{Z}_2$-even operators as $X_j \rightsquigarrow \tilde{Z}_{j-\frac{1}{2}} \tilde{Z}_{j+\frac{1}{2}}$ and $Z_{j-1} Z_j \rightsquigarrow \tilde{X}_{j-\frac{1}{2}}$. By relabeling $\tilde{X}_{j+\frac{1}{2}} \to X_j$, $\tilde{Z}_{j+\frac{1}{2}} \to Z_j$, this gauging map is implemented by the Kramers-Wannier operator KW in the following sense:

$$(\text{KW})X_j = Z_{j-1}Z_j(\text{KW}), \qquad (\text{KW})Z_{j-1}Z_j = X_{j-1}(\text{KW}). \tag{A.7}$$

However, this operator cannot be invertible on a closed periodic chain; otherwise $\eta = \prod_{j=1}^{L} X_j = \prod_{j=1}^{L} (\text{KW})^{-1} Z_{j-1} Z_j (\text{KW}) = (\text{KW})^{-1} \left( \prod_{j=1}^{L} Z_{j-1} Z_j \right) (\text{KW}) = 1$, which is a contradiction. In fact, the non-invertible Kramers-Wannier operator is an MPO whose tensor is given in (13). Any Hamiltonian $H$ that is invariant under gauging the $\mathbb{Z}_2$ symmetry enjoys an enhanced non-invertible Kramers-Wannier symmetry $\text{KW}' = U \, \text{KW}$ for some local unitary transformation $U$, i.e., $H(\text{KW}') = (\text{KW}')H$. The full enhanced algebra formed by KW is [6,7]

$$\eta^2 = 1, \quad \eta(\text{KW}) = (\text{KW})\eta = \text{KW}, \quad (\text{KW})^\dagger = T(\text{KW}) = (\text{KW})T,$$
$$(\text{KW})^2 = (1 + \eta)T^{-1}, \tag{A.8}$$

where $T$ is the lattice translation by one site.

## A.2 Gauging in the language of defects

Here, we gauge the $\mathbb{Z}_2$ symmetry in the language of defects following [56]. The gauging procedure involves two steps: we sum over the insertion of all defects on links, and then we impose Gauss's law via a local projector constructed from the fusion operators. We will see that this method gives the same gauged Hamiltonian and Gauss's law as those from the ordinary gauging procedure.

### A.2.1 $\mathbb{Z}_2$ defect

The insertion of a $\mathbb{Z}_2$ defect for the generator $\eta$ is described by modfiying the Hamiltonian locally at link $(0, 1)$:

$$H_\eta^{(0,1)} = -\sum_{j=2}^{L} \left( h^{-1} Z_{j-1} Z_j + h X_j \right) + h^{-1} Z_0 Z_1 - h X_1. \tag{A.9}$$

This defect is topological as it can be moved by the unitary movement operator $\lambda_\eta^j = X_j$ in the sense that $H_\eta^{(j,j+1)} = \lambda_\eta^j H_\eta^{(j-1,j)} (\lambda_\eta^j)^{-1}$. We represent this movement operator diagrammatically as

$$\lambda_\eta^j = X_j = \quad \text{[diagram]} \quad .$$

(A.10)

Given two defects labeled by $a, b = 1$ or $\eta$, we define a fusion operator $(\lambda^j)_{a\otimes b}^{ab}$ to fuse them into the product defect $ab$. It is a unitary operator satisfying the following defining equation

$$(\lambda^j)_{a\otimes b}^{ab} H_{a;b}^{(j-1,j);(j,j+1)} = H_{ab}^{(j,j+1)} (\lambda^j)_{a\otimes b}^{ab} ,$$

(A.11)

where $H_{a;b}^{(j-1,j);(j,j+1)}$ is the Hamiltonian with an $a$ defect on link $(j-1,j)$ and a $b$ defect on link $(j, j+1)$. This fusion operator can be diagrammatically represented as

$$(\lambda^j)_{a\otimes b}^{ab} = \quad \text{[diagram]} \quad , \qquad a, b \in \mathbb{Z}_2 .$$

(A.12)

As an example, $(\lambda^j)_{\eta\otimes\eta}^1 = X_j$ pair annihilates two adjacent $\eta$ defects into a trivial defect

$$(\lambda^j)_{\eta\otimes\eta}^1 H_{\eta;\eta}^{(j-1,j);(j,j+1)} ((\lambda^j)_{\eta\otimes\eta}^1)^{-1} = H .$$

(A.13)

This fusion operator can be diagrammatically represented as

$$(\lambda^j)_{\eta\otimes\eta}^1 = X_j = \quad \text{[diagram]} \quad .$$

(A.14)

### A.2.2  Coupling to $\mathbb{Z}_2$ gauge fields

Coupling to the dynamical gauge fields is equivalent to summing over the insertion of all possible defects. First, we extend the Hilbert space $\mathcal{H}$ by adding a qubit on each link:

$$\widetilde{\mathcal{H}} \equiv \bigoplus_{a_1,\ldots,a_L\in\mathbb{Z}_2} \mathcal{H}_{a_1;a_2;\ldots;a_L} \sim \bigoplus_{a_1,\ldots,a_L\in\mathbb{Z}_2} \quad \text{[diagram]} \quad .$$

(A.15)

These new qubits represent the degrees of freedom for the $\mathbb{Z}_2$ gauge fields. On this extended $2^{2L}$-dimensional Hilbert space, the Hamiltonian that includes the insertion of a $\mathbb{Z}_2$ defect at every link is

$$\tilde{H} = \sum_{a_1,a_2,\ldots,a_L\in\mathbb{Z}_2} H_{a_1;a_2;\cdots;a_L} \otimes |a_1, a_2, \ldots, a_L\rangle\langle a_1, a_2, \ldots, a_L|_{\text{links}}$$

$$= -\sum_{j=1}^L \left( h^{-1} Z_{j-1} \sigma_{j-\frac{1}{2}}^z Z_j + h X_j \right),$$

(A.16)

where the Pauli operators $\sigma_{j-\frac{1}{2}}^x, \sigma_{j-\frac{1}{2}}^z$ act on the qubit on the link $j-\frac{1}{2}$. This reproduces the gauged Hamiltonian in (A.2). (Here and below we sometimes denote a link $(j, j+1)$ as $j+\frac{1}{2}$ to shorten the expression.)

### A.2.3  Gauss's law from defects

Following [56], we impose Gauss's law via a unitary operator $\mathcal{G}_j(g)$ that generates a local $G$ symmetry. This local unitary operator is constructed from the extended fusion operator, which acts on the extended Hilbert space (A.15). We extend the fusion operator $(\lambda^j)^{ab}_{a\otimes b}$ to

$$(\tilde{\lambda}^j)^{ab}_{a\otimes b} \equiv (\lambda^j)^{ab}_{a\otimes b} \otimes \left(|1\rangle\langle a|_{(j-1,j)} \otimes |ab\rangle\langle b|_{(j,j+1)}\right) \bigotimes_{j'\neq j-1,j} \mathbb{1}_{(j',j'+1)}, \quad a,b \in \mathbb{Z}_2. \quad \text{(A.17)}$$

Crucially, it commutes with the extended Hamiltonian $\tilde{H}$.

We then define the local unitary operator $\mathcal{G}_j(g)$ (with $g = 1, \eta$) that creates a pair of defects around site $j$ as

$$\mathcal{G}_j(g) \equiv \sum_{a,b\in\mathbb{Z}_2} \left((\tilde{\lambda}^j)^{ab}_{ag^{-1}\otimes gb}\right)^\dagger (\tilde{\lambda}^j)^{ab}_{a\otimes b} = \sum_{a,b\in\mathbb{Z}_2} \quad \otimes |ag^{-1},gb\rangle\langle a,b|_{j-\frac{1}{2},j+\frac{1}{2}}. \quad \text{(A.18)}$$

This operator satisfies $\mathcal{G}_j(g_1)\mathcal{G}_j(g_2) = \mathcal{G}_j(g_1 g_2)$. For the nontrivial $\mathbb{Z}_2$ element $\eta$, this local unitary operator gives back the Gauss law operator (A.3) in the ordinary gauging procedure:

$$\mathcal{G}_j(\eta) = \quad + \quad + \quad + \quad \text{(A.19)}$$

$$= X_j \left(|00\rangle\langle 11|_{j-\frac{1}{2},j+\frac{1}{2}} + |11\rangle\langle 00|_{j-\frac{1}{2},j+\frac{1}{2}} + |10\rangle\langle 01|_{j-\frac{1}{2},j+\frac{1}{2}} + |01\rangle\langle 10|_{j-\frac{1}{2},j+\frac{1}{2}}\right)$$

$$= \sigma^x_{j-\frac{1}{2}} X_j \sigma^x_{j+\frac{1}{2}}.$$

(Here, we have added the dotted lines for the trivial defect for clarity.)

Finally, we define a projector by summing the Gauss law operator $\mathcal{G}_j(g)$ over the $\mathbb{Z}_2$ elements:

$$P_j = \frac{1}{2}\sum_{g\in\mathbb{Z}_2} \mathcal{G}_j(g) = \frac{1}{2}\left(1 + \sigma^x_{j-\frac{1}{2}} X_j \sigma^x_{j+\frac{1}{2}}\right). \quad \text{(A.20)}$$

This projects to the subspace $\sigma^x_{j-\frac{1}{2}} X_j \sigma^x_{j+\frac{1}{2}} = 1$ as in (A.4).

## A.3  Gauging in the language of algebra objects

Finally, we rewrite the gauging method of the previous section in the language of algebra objects. This method can be generalized to gauge a fusion category symmetry $\mathcal{C}$, in which the sum of defects is replaced by the algebra object $\mathcal{A}$.

### A.3.1  Lattice Frobenius algebra for gauging $\mathbb{Z}_2$

The sum over the insertion of defects and the projection operator $P_j$ can be conveniently packaged in the language of the algebra object. The algebra object is the categorical generalization

of the choice of a finite subgroup and the discrete torsion when gauging an invertible symmetry. More specifically, the algebra object $\mathcal{A}$ associated with gauging a finite subgroup $H \subseteq G$ is a formal sum of defects associated with $H$, i.e., $\mathcal{A} = \bigoplus_{g \in H} g$.

For gauging the $\mathbb{Z}_2$ symmetry, the algebra object is $\mathcal{A} = 1 \oplus \eta$. The defect Hamiltonian for $\mathcal{A}$ is

$$H_{\mathcal{A}}^{(0,1)} = H \otimes |0\rangle\langle 0|_\ell + H_\eta^{(0,1)} \otimes |1\rangle\langle 1|_\ell = -\sum_{j=2}^{L}\left(h^{-1}Z_{j-1}Z_j + hX_j\right) - h^{-1}Z_0 Z_\ell Z_1 - hX_1. \quad \text{(A.21)}$$

The defect Hilbert space includes an extra qubit labeled by $\ell$ localized on link $(0, 1)$. The state $|0\rangle_\ell$ (which has $Z_\ell = +1$) corresponds to the trivial defect $1$ of $\mathcal{A}$, while the state $|1\rangle_\ell$ (which has $Z_\ell = -1$) corresponds to the $\mathbb{Z}_2$ defect $\eta$ of $\mathcal{A}$. A defect, which is a sum of two constituents, is called non-simple.

Next, consider the Hamiltonian for two $\mathcal{A}$'s on the *same* link:

$$\begin{aligned}
H_{\mathcal{A}_1 \otimes \mathcal{A}_2}^{(j,j+1)} &= H_{1 \otimes 1}^{(j,j+1)} \otimes |00\rangle\langle 00|_{\ell_1 \ell_2} + H_{\eta \otimes 1}^{(j,j+1)} \otimes |10\rangle\langle 10|_{\ell_1 \ell_2} \\
&\quad + H_{1 \otimes \eta}^{(j,j+1)} \otimes |01\rangle\langle 01|_{\ell_1 \ell_2} + H_{\eta \otimes \eta}^{(j,j+1)} \otimes |11\rangle\langle 11|_{\ell_1 \ell_2} \\
&= -\sum_{j=2}^{L}\left(h^{-1}Z_{j-1}Z_j + hX_j\right) - h^{-1}Z_0 Z_{\ell_1} Z_{\ell_2} Z_1 - hX_1.
\end{aligned} \quad \text{(A.22)}$$

The insertion of two defects on the same link is unambiguous in this case because the symmetry is on-site. More generally, this is unambiguous when the movement operators of two defects commute with each other. In such cases, the defects Hamiltonian does not depend on the order in which the two defects have moved to the same link starting from a far separated configuration of the defects. We add a subscript $i = 1, 2$ for the algebra object to distinguish their associated qubits labeled by $\ell_1$ and $\ell_2$.

This pair of $\mathcal{A}$'s can be fused into a single copy of $\mathcal{A}$ by an onsite fusion operator $m_{\mathcal{A}_1 \otimes \mathcal{A}_2}^{\mathcal{A}_2}$ as follows

$$m_{\mathcal{A}_1 \otimes \mathcal{A}_2}^{\mathcal{A}_2} H_{\mathcal{A}_1 \otimes \mathcal{A}_2}^{(j,j+1)} = H_{\mathcal{A}_2}^{(j,j+1)} m_{\mathcal{A}_1 \otimes \mathcal{A}_2}^{\mathcal{A}_2}. \quad \text{(A.23)}$$

Explicitly, the on-site fusion operator is given by

$$m_{\mathcal{A}_1 \otimes \mathcal{A}_2}^{\mathcal{A}_2} = \frac{1}{\sqrt{2}}\langle 0|_{\ell_1} + \frac{1}{\sqrt{2}}X_{\ell_2}\langle 1|_{\ell_1} = \qquad . \quad \text{(A.24)}$$

More covariantly, if we label the output algebra object by $\mathcal{A}_3$, this operator becomes[12]

$$m_{\mathcal{A}_1 \otimes \mathcal{A}_2}^{\mathcal{A}_3} = \langle +|_{\ell_1}\langle +|_{\ell_2}|+\rangle_{\ell_3} + \langle -|_{\ell_1}\langle -|_{\ell_2}|-\rangle_{\ell_3}, \quad \text{(A.25)}$$

Similarly, the Hermitian conjugate of $m_{\mathcal{A}_1 \otimes \mathcal{A}_2}^{\mathcal{A}_3}$ is

$$(m_{\mathcal{A}_1 \otimes \mathcal{A}_2}^{\mathcal{A}_3})^\dagger = |+\rangle_{\ell_1}|+\rangle_{\ell_2}\langle +|_{\ell_3} + |-\rangle_{\ell_1}|-\rangle_{\ell_2}\langle -|_{\ell_3} = \qquad . \quad \text{(A.26)}$$

---

[12]This expression makes it clear that this onsite fusion operator $m_{\mathcal{A}_1 \otimes \mathcal{A}_2}^{\mathcal{A}_3}$ for gauging a $\mathbb{Z}_2$ symmetry is the $X$-spider in ZX calculus [87]. Most of the conditions below then directly follow from the general rules in ZX calculus.

The onsite fusion operator $m^{\mathcal{A}_2}_{\mathcal{A}_1 \mathcal{A}_2}$ and its Hermitian conjugate give the explicit lattice realization of the multiplication and co-multiplication for the algebra object. It is straightforward to check that they satisfy all the mathematical axioms. This includes the associativity (63), the Forbenius relation (65), the separability condition (66), and (71) that ensures the symmetric condition.

Two other important structures for an algebra object are the unit and the co-unit. For $\mathbb{Z}_2$, they are given by

$$u_{\mathcal{A}} = \sqrt{2}|0\rangle_\ell, \qquad (u_{\mathcal{A}})^\dagger = \sqrt{2}\langle 0|_\ell. \tag{A.27}$$

The unit and co-unit satisfy (68) and the Hermitian conjugate of this relation, where $\mathbb{1}_{\mathcal{A}_2}$ is the identity matrix on the qubit $\ell_2$ (tensor multiplied by the identity matrix on the physical qubits).

### A.3.2 Gauss law from the algebra object

We can readily extend the discussion of the movement operator to this algebra object. This non-simple defect is topological as it can be moved by a unitary movement operator $\lambda^j_{\mathcal{A}}$,

$$\lambda^j_{\mathcal{A}} = |0\rangle\langle 0|_\ell + X_j|1\rangle\langle 1|_\ell = \text{CNOT}_{\ell,j} = X_j^{\frac{1-Z_\ell}{2}}, \tag{A.28}$$

which moves the location of $\mathcal{A}$ by one site as

$$\lambda^j_{\mathcal{A}} H^{(j-1,j)}_{\mathcal{A}} (\lambda^j_{\mathcal{A}})^{-1} = H^{(j,j+1)}_{\mathcal{A}}. \tag{A.29}$$

The projector will be more directly written in terms of a fusion operator $(\lambda^j)^{\mathcal{A}_2}_{\mathcal{A}_1 \otimes \mathcal{A}_2}$ that also moves the algebra object by one site, which is defined by

$$(\lambda^j)^{\mathcal{A}_2}_{\mathcal{A}_1 \otimes \mathcal{A}_2} H^{(j-1,j);(j,j+1)}_{\mathcal{A}_1 \otimes \mathcal{A}_2} = H^{(j,j+1)}_{\mathcal{A}_2} (\lambda^j)^{\mathcal{A}_2}_{\mathcal{A}_1 \otimes \mathcal{A}_2}. \tag{A.30}$$

It can be diagrammatically represented as

$$(\lambda^j)^{\mathcal{A}_2}_{\mathcal{A}_1 \otimes \mathcal{A}_2} = \qquad , \tag{A.31}$$

which moves two defects to the same site and then performs the onsite fusion. Thus it can be decomposed into

$$(\lambda^j)^{\mathcal{A}_2}_{\mathcal{A}_1 \otimes \mathcal{A}_2} = m^{\mathcal{A}_2}_{\mathcal{A}_1 \otimes \mathcal{A}_2} \lambda^j_{\mathcal{A}_1}. \tag{A.32}$$

Using (A.28) and (A.24), this fusion operator is

$$(\lambda^j)^{\mathcal{A}_2}_{\mathcal{A}_1 \otimes \mathcal{A}_2} = \frac{1}{\sqrt{2}}\Big(\langle 0|_{\ell_1} + X_j X_{\ell_2}\langle 1|_{\ell_1}\Big). \tag{A.33}$$

Similarly, it satisfies the separability condition as its onsite counterpart:

$$(\lambda^j)^{\mathcal{A}_2}_{\mathcal{A}_1 \otimes \mathcal{A}_2}\Big((\lambda^j)^{\mathcal{A}_2}_{\mathcal{A}_1 \otimes \mathcal{A}_2}\Big)^\dagger = \frac{1}{2}\Big(\langle 0|_{\ell_1} + X_j X_{\ell_2}\langle 1|_{\ell_1}\Big)\Big(|0\rangle_{\ell_1} + X_j X_{\ell_2}|1\rangle_{\ell_1}\Big) = \mathbb{1}_{\mathcal{A}_2}, \tag{A.34}$$

which is diagrammatically represented as in (72).

We now define the projector for the Gauss law:

$$
P_j = \quad = \left(\left(\lambda^j\right)^{\mathcal{A}_2}_{\mathcal{A}_1 \otimes \mathcal{A}_2}\right)^\dagger \left(\left(\lambda^j\right)^{\mathcal{A}_2}_{\mathcal{A}_1 \otimes \mathcal{A}_2}\right). \tag{A.35}
$$

Indeed, it is a projector in the sense that $P_j^2 = P_j$, which follows from (A.34). Explicitly, the projector for gauging the $\mathbb{Z}_2$ symmetry is

$$
P_j = \frac{1}{2}\Big[|0\rangle\langle 0|_{\ell_1} + |1\rangle\langle 1|_{\ell_1} + (|0\rangle\langle 1|_{\ell_1} + |1\rangle\langle 0|_{\ell_1})X_j X_{\ell_2}\Big] = \frac{1 + X_{\ell_1} X_j X_{\ell_2}}{2}. \tag{A.36}
$$

Imposing $P_j = 1$ sets $\mathcal{G}_j = X_{\ell_1} X_j X_{\ell_2} = 1$. Relabeing $X_{\ell_1}, X_{\ell_2}$ as $\sigma^x_{j-\frac{1}{2}}, \sigma^x_{j+\frac{1}{2}}$, we recover Gauss's law $\mathcal{G}_j = \sigma^x_{j-\frac{1}{2}} X_j \sigma^x_{j+\frac{1}{2}} = 1$ from the ordinary gauging procedure in (A.4).

# B Cosine symmetry and Rep($D_8$) defects

In this appendix, we derive the cosine non-invertible symmetry of (6) by gauging the $\mathbb{Z}_2 \in O(2)$ charge conjugation symmetry of the XXZ chain. We derive an expression for the non-invertible defects as well as the operators. Finally, we identify the Rep($D_8$) symmetry as a subcategory of the cosine symmetry.

## B.1 Cosine symmetry from gauging XXZ

We start with the XXZ chain Hamiltonian with even $L$

$$
H_{\text{XXZ}} = \sum_{j=1}^{L}(X_j X_{j+1} + Y_j Y_{j+1} + \lambda Z_j Z_{j+1}), \tag{B.1}
$$

which has an $O(2) = \mathbb{Z}_2^X \ltimes U(1)_Z$ symmetry generated by

$$
X = \prod_{j=1}^{L} X_j, \quad \text{and} \quad U_\theta = \prod_{j=1}^{L} e^{i\theta \frac{Z_j}{2}}. \tag{B.2}
$$

Below, we find that the non-invertible cosine symmetry arises from this $O(2)$ symmetry upon gauging the $\mathbb{Z}_2^X$ symmetry.[13]

Gauging $\mathbb{Z}_2^X$ is implemented by Kramers-Wannier (KW) gauging map that acts on $\mathbb{Z}_2$ invariant operators as $\text{KW}: X_j \rightsquigarrow Z_{j-1}Z_j, Z_{j-1}Z_j \rightsquigarrow X_{j-1}$, from which we find the gauged Hamiltonian

$$
H_{\text{gauged}} = \sum_j (Z_{j-1}Z_{j+1} - Z_{j-1}X_j Z_{j+1} + \lambda X_j). \tag{B.3}
$$

This is related to the Hamiltonian (6) by conjugation with $\prod_j Z_j$ and setting $h_0 = -\lambda$ and $h_1 = 1$. The $U(1)_Z$ symmetry of the XXZ chain does not commute with the charge conjugation symmetry and naively is gone after gauging since the charge $\mathcal{Q} = \sum_j Z_j/2$ is not gauge

---

[13]The XXZ model at $\lambda = 0$, which is known as the XX model, has a rich family of non-invertible symmetries [79].

invariant. However, the "cosine" operator $e^{i\theta\mathcal{Q}} + e^{-i\theta\mathcal{Q}}$ is gauge invariant, and one expects it to remain as a (non-invertible) symmetry of the gauged theory. Note that since $\mathcal{Q}$ is not gauge invariant, we have to clarify what is meant by the expression $e^{i\theta\mathcal{Q}} + e^{-i\theta\mathcal{Q}}$ in the gauged theory. We will see that a more precise expression for the cosine operator is $\frac{1+\eta}{2}(e^{i\theta Q} + e^{-i\theta Q})$, where $\eta = \prod_j X_j$ is the dual $\mathbb{Z}_2$ symmetry upon gauging $\mathbb{Z}_2^X$ and $Q$ is a new charge operator.[14]

We start by adding a single dynamical $\mathbb{Z}_2^X$ gauge field, which is a qubit labeled by $\ell$, to the XXZ Hamiltonian on link $(L, 1)$. For simplicity, we set $\lambda = 0$. Adding a dynamical gauge field is the same as inserting a $1 \oplus X$ defect, which corresponds to the defect Hamiltonian

$$(H_{\text{XXZ}})_{1\oplus X}^{(L,1)} = \sum_{j=1}^{L-1}(X_j X_{j+1} + Y_j Y_{j+1}) + X_L X_1 + Z_\ell Y_L Y_1 \,. \tag{B.4}$$

Gauging is implemented by first doing the *unitary* transformation [7, Section 2.3.1]

$$\text{KW}_{1\oplus X} : \begin{array}{ll} X_j \mapsto Z_{j-1}Z_j & \text{(for } j \neq 1), \qquad Z_j \mapsto X_\ell X_j X_{j+1}\cdots X_L\,, \\ X_1 \mapsto Z_\ell Z_L Z_1\,, & \hspace{2.4cm} Z_\ell \mapsto X_1 X_2 \cdots X_L\,, \end{array} \tag{B.5}$$

which leads to the Hamiltonian

$$\widetilde{H}_{\text{gauged}} = \sum_{j=2}^{L-1}(Z_{j-1}Z_{j+1} - Z_{j-1}X_j Z_{j+1}) + Z_\ell Z_{L-1}Z_1(1-X_L) + Z_\ell Z_L Z_2(1-X_1) \,. \tag{B.6}$$

Next, we impose the Gauss law $Z_\ell = 1$. (Since we started with a single dynamical gauge field, there is only a single Gauss law.) Indeed, by setting $Z_\ell = 1$ in (B.6) we find (B.3) for $\lambda = 0$. Violation of Gauss's law corresponds to inserting a defect for the dual $\mathbb{Z}_2^\eta$ symmetry. Thus, the Hamiltonian (B.6) can be interpreted as the Hamiltonian of the gauged theory coupled to a dynamical $\mathbb{Z}_2^\eta$ gauge field, i.e., $\widetilde{H}_{\text{gauged}} = (H_{\text{gauged}})_{1\oplus\eta}^{(L,1)}$.

The idea is to apply (B.5) to the $U(1)_Z$ symmetry and find its image in the gauged theory. Note that $\mathcal{Q} = \frac{1}{2}\sum_j Z_j$ is not a symmetry of (B.4) in the presence of a dynamical $\mathbb{Z}_2$ gauge field and needs to be modified. The modified conserved charge is

$$\mathcal{Q}_{1\oplus X} = \frac{1+Z_\ell}{2}(Z_1 + Z_2 + \ldots + Z_L) \,, \tag{B.7}$$

which commutes with (B.4).[15] Under the unitary transformation (B.5), this conserved charge becomes

$$\begin{aligned} \widetilde{Q} &= \frac{1+\eta}{2}X_\ell(1 + X_1 + X_1 X_2 + \ldots + X_1 X_2 \cdots X_{L-1}) \\ &= \frac{X_\ell}{2}\sum_{j=1}^{2L} X_1 X_2 \cdots X_j \,, \end{aligned} \tag{B.8}$$

which commutes with (B.6). Note that $X_j$ obeys periodic boundary condition $X_{j+L} = X_j$.

Having found the expression for the conserved charge $\widetilde{Q} = \sum_{j=1}^{2L}\tilde{q}_j$, we conjugate the Hamiltonian (B.6) by a truncated symmetry operator,

$$e^{-i\theta(\tilde{q}_1 + \tilde{q}_2 + \ldots + \tilde{q}_j)}\,, \quad \text{for} \quad \tilde{q}_j = \frac{X_\ell}{2}X_1 X_2 \cdots X_j \,. \tag{B.9}$$

---

[14]The dual $\mathbb{Z}_2^\eta$ symmetry is generated by $\prod_{j=1}^{L} X_j$ as defined in (15). Even though it takes the same expression as the operator $X$, we use two different symbols to distinguish them as they are viewed as symmetries of two different Hamiltonians.

[15]Note that we can rewrite the modified charge as $\mathcal{Q}_{1\oplus X} = \frac{1}{2}\sum_{j=1}^{2L} Z_j$ if we use the boundary condition $Z_{j+L} = Z_\ell Z_j$.

The conjugation modifies the Hamiltonian around site 1 and around site $j$ and creates a pair of defects associated with the conserved charge at these two locations. This is the lattice analog of the generalized Noether procedure described in [54]. Doing the conjugation, and focusing around link $(L, 1)$, we find

$$\ldots + Z_{L-2}Z_L(1-X_{L-1}) + e^{i\theta X_\ell}Z_\ell Z_{L-1}Z_1(1-X_L) + Z_\ell Z_L Z_2(1-X_1) + Z_1 Z_3(1-X_2) + \ldots \quad (B.10)$$

Now that we have found the local expression for the defect around a site, we can insert a single defect on a periodic chain at link $(J, J+1)$ for the non-invertible cosine symmetry:

$$(H_{\text{gauged}})_\theta^{(J,J+1)} = \sum_{j\neq J} Z_{j-1}Z_{j+1}(1-X_j) + e^{-i\theta X_\ell}Z_\ell Z_{J-1}Z_{J+1}(1-X_J) + Z_\ell Z_J Z_{J+2}(1-X_{J+1}). \quad (B.11)$$

Note that this is a deformation of the $1 \oplus \eta$ defect at $\theta = 0$. Indeed, this defect is topological, and its movement operator is given by $\text{CNOT}_{\ell,j}\, e^{i\theta \frac{X_\ell}{2}} = (X_j)^{\frac{1-Z_\ell}{2}} e^{i\theta \frac{X_\ell}{2}}$.

Finally, to connect with the convention used in the main text, we conjugate the system by $\frac{1-iZ_\ell}{\sqrt{2}}\prod_j Z_j$ and find the defect Hamiltonian

$$H_\theta^{(J,J+1)} = \sum_{j\neq J} Z_{j-1}Z_{j+1}(1+X_j) + e^{-i\theta Y_\ell}Z_\ell Z_{J-1}Z_{J+1}(1+X_J) + Z_\ell Z_J Z_{J+2}(1+X_{J+1}), \quad (B.12)$$

with the movement operator

$$\lambda_\theta^j = (-X_j)^{\frac{1-Z_\ell}{2}} e^{i\theta \frac{Y_\ell}{2}} = \text{CNOT}_{\ell,j}Z_\ell\, e^{i\theta \frac{Y_\ell}{2}}. \quad (B.13)$$

We note that the Rep($D_8$) defect $\mathcal{D}$ in (25) is a special case of the cosine defect for $\theta = \frac{\pi}{2}$.

## B.2 MPO for the cosine symmetry

Since the movement operator only acts on a single physical site, it can be identified as an MPO tensor $\mathbb{U}_\theta^j = \lambda_\theta^j$ and leads to the following MPO presentation of the cosine non-invertible operator

$$\mathsf{L}_\theta = \text{Tr}_\ell\left(\mathbb{U}_\theta^L \cdots \mathbb{U}_\theta^2 \mathbb{U}_\theta^1\right). \quad (B.14)$$

We simplify the expression above to

$$\begin{aligned}
\mathsf{L}_\theta &= \text{Tr}_\ell\left((-X_L)^{\frac{1-Z_\ell}{2}}\cdots(-X_1)^{\frac{1-Z_\ell}{2}} e^{i\theta Y_\ell Q}\right) \\
&= \text{Tr}_\ell\left(\eta^{\frac{1-Z_\ell}{2}} e^{i\theta Y_\ell Q}\right) = \frac{1+\eta}{2}\left(e^{i\theta Q} + e^{-i\theta Q}\right),
\end{aligned} \quad (B.15)$$

where

$$Q = \frac{1}{2}(1 - X_1 + X_1 X_2 - X_1 X_2 X_3 + \ldots - X_1 X_2 \cdots X_{L-1}). \quad (B.16)$$

The expression (B.15) makes the following cosine-like fusion relations manifest

$$\mathsf{L}_\theta \mathsf{L}_{\theta'} = \mathsf{L}_{\theta+\theta'} + \mathsf{L}_{\theta-\theta'}, \qquad (\mathsf{L}_\theta)^\dagger = \mathsf{L}_{-\theta} = \mathsf{L}_\theta, \qquad \mathsf{L}_{\theta+2\pi} = \mathsf{L}_\theta. \quad (B.17)$$

## B.3 Action on local operators

The action of the non-invertible operator $\mathsf{L}_\theta$ on local operators can be computed by sequentially acting with the unitary operators $\mathbb{U}_\theta^j$ given in (18). This sequential action terminates after a finite number of steps if the local operators commute with $\mathsf{L}_{2\theta}$.[16]

---

[16]More generally, a non-invertible symmetry operator $\mathsf{S}$ has a 'nice' action on a local operator $O_j$ if $O_j$ commutes with $\mathsf{S}^\dagger\mathsf{S}$. By a nice action, we mean a commutation relation such as $\mathsf{S}O_j = O_j'\mathsf{S}$ where $O_j'$ is a local operator.

For instance, using the MPO presentation (18), the cosine operator $\mathsf{L}_{\pi/4}$ acts on some of the local operators as

$$
\begin{aligned}
\mathbb{U}_{\pi/4}^{j} : & \qquad X_j \mapsto X_j, \\
\mathbb{U}_{\pi/4}^{j+1}\mathbb{U}_{\pi/4}^{j}\mathbb{U}_{\pi/4}^{j-1} : & \qquad Z_{j-1}Z_{j+1}(1+X_j) \mapsto Z_{j-1}Z_{j+1}(1+X_j), \\
\mathbb{U}_{\pi/4}^{j+2}\mathbb{U}_{\pi/4}^{j+1}\mathbb{U}_{\pi/4}^{j}\mathbb{U}_{\pi/4}^{j-1}\mathbb{U}_{\pi/4}^{j-2} : & \quad Z_{j-2}Z_{j+2}(1+X_{j-1}X_{j+1}) \mapsto Z_{j-2}Z_{j+2}(X_j)^{\frac{1-X_{j-1}}{2}}(1+X_{j-1}X_{j+1}).
\end{aligned}
\tag{B.18}
$$

## C  More on the defect Hamiltonians and gauge fields

In Section 3, we gave a general algorithm on how to gauge the algebra object $\mathcal{A} = 1 \oplus \eta \oplus \mathcal{D}$ for any $\mathrm{Rep}(D_8)$ symmetric Hamiltonian and applied it to the $h_0$ and $h_1$ terms of the following Hamiltonian in (79) as an illustration:

$$
H = h_0 \sum_j X_j + h_1 \sum_j Z_{j-1}Z_{j+1}(1+X_j) + h_2 \sum_j Z_{j-2}Z_{j+2}(1+X_{j-1}X_{j+1}).
\tag{C.1}
$$

However, these two terms turn out to be invariant under gauging $\mathcal{A}$. In this appendix, we apply the same algorithm to the $h_2$ term and find that it transforms nontrivially under gauging $\mathcal{A}$. The Hamiltonian (C.1) is the simplest $\mathrm{Rep}(D_8)$ symmetric term that transforms under gauging.

### C.1  Defect Hamiltonian of $\mathcal{D}$

We first derive the Hamiltonian for the non-invertible defect $\mathcal{D}$ for the $h_2$ term following the algorithm in Section 2.3. As discussed there, on an infinite chain, the Hamiltonian with a defect inserted at link $(J, J+1)$ is obtained from the one without by $H_{\mathcal{D}}^{(J,J+1)} = U_{\mathcal{D}}^{\leq J} H (U_{\mathcal{D}}^{\leq J})^\dagger$, where $U_{\mathcal{D}}^{\leq J}$ is the defect creation homomorphism (22) and $\lambda_{\mathcal{D}}^j = \mathrm{CNOT}_{\ell,j}\mathsf{H}_\ell$ is the movement operator defined in (20).

Applying the transformation (24) to the $\mathrm{Rep}(D_8)$ symmetric local operators in (77), we find

$$
U_{\mathcal{D}}^{\leq J} : \quad
\begin{aligned}
Z_{j-1}Z_{j+1}(1+X_j) &\mapsto Z_{j-1}Z_{j+1}(1+X_j), \\
Z_{J-1}Z_{J+1}(1+X_J) &\mapsto X_\ell Z_{J-1}Z_{J+1}(1+X_J), \\
Z_J Z_{J+2}(1+X_{J+1}) &\mapsto Z_\ell Z_J Z_{J+2}(1+X_{J+1}),
\end{aligned}
\tag{C.2}
$$

for $j \neq J, J+1$, and

$$
U_{\mathcal{D}}^{\leq J} : \quad
\begin{aligned}
Z_{j-2}Z_{j+2}(1+X_{j-2}X_j) &\mapsto Z_{j-2}Z_{j+2}(1+X_{j-2}X_j), \\
Z_{J-3}Z_{J+1}(1+X_{J-2}X_J) &\mapsto X_\ell Z_{J-3}Z_{J+1}(1+X_{J-2}X_J), \\
Z_{J-2}Z_{J+2}(1+X_{J-1}X_{J+1}) &\mapsto Z_\ell Z_{J-2}Z_{J+2}(X_{J-1}+X_{J+1}), \\
Z_{J-1}Z_{J+3}(1+X_J X_{J+2}) &\mapsto X_\ell Z_{J-1}Z_{J+3}(X_J + X_{J+2}), \\
Z_J Z_{J+4}(1+X_{J+1}X_{J+3}) &\mapsto Z_\ell Z_J Z_{J+4}(1+X_{J+1}X_{J+3}),
\end{aligned}
\tag{C.3}
$$

for $j \neq J-1, J, J+1, J+2$.

Applying these transformations to the Hamiltonian (C.1), we find the defect Hamiltonian:

$$
\begin{aligned}
H_{\mathcal{D}}^{(J,J+1)} = {} & h_0 \sum_j X_j + h_1 \sum_{j \neq J, J+1} Z_{j-1}Z_{j+1}(1+X_j) + h_2 \sum_{j \neq J-1, J, J+1, J+2} Z_{j-2}Z_{j+2}(1+X_{j-1}X_{j+1}) \\
& + h_1 X_\ell Z_{J-1}Z_{J+1}(1+X_J) + h_1 Z_\ell Z_J Z_{J+2}(1+X_{J+1}) \\
& + h_2 \Big( X_\ell Z_{J-3}Z_{J+1}(1+X_{J-2}X_J) + Z_\ell Z_{J-2}Z_{J+2}(X_{J-1}+X_{J+1}) \\
& \qquad + X_\ell Z_{J-1}Z_{J+3}(X_J + X_{J+2}) + Z_\ell Z_J Z_{J+4}(1+X_{J+1}X_{J+3})\Big).
\end{aligned}
\tag{C.4}
$$

## C.2 Coupling to $\mathcal{A}$ gauge fields

Using (57), we find the Hamiltonian with a single $\mathcal{A} = 1 \oplus \eta \oplus \mathcal{D}$ defect at link $(J, J+1)$ including the $h_2$ term:

$$
\begin{aligned}
H_{\mathcal{A}}^{(J,J+1)} = {} & h_0 \sum_j X_j + h_1 \sum_{j \neq J, J+1} Z_{j-1} Z_{j+1}(1+X_j) + h_2 \sum_{j \neq J-1, J, J+1, J+2} Z_{j-2} Z_{j+2}(1+X_{j-1}X_{j+1}) \\
& + h_1 \Big( (-iY_\ell)^{\frac{1-Z_a}{2}} Z_\ell Z_{J-1} Z_{J+1}(1+X_J) + Z_\ell Z_J Z_{J+2}(1+X_{J+1}) \Big) \\
& + h_2 \Big( (-iY_\ell)^{\frac{1-Z_a}{2}} Z_\ell Z_{J-3} Z_{J+1}(1+X_{J-2}X_J) + (X_{J-1})^{\frac{1-Z_a}{2}} Z_\ell Z_{J-2} Z_{J+2}(1+X_{J-1}X_{J+1}) \\
& \quad + (-iY_\ell X_J)^{\frac{1-Z_a}{2}} Z_\ell Z_{J-1} Z_{J+3}(1+X_J X_{J+2}) + Z_\ell Z_J Z_{J+4}(1+X_{J+1}X_{J+3}) \Big).
\end{aligned}
\tag{C.5}
$$

Using the variables introduced in (75), this defect Hamiltonian is equal to

$$
\begin{aligned}
H_{\mathcal{A}}^{(J,J+1)} = {} & h_0 \sum_j X_j + h_1 \sum_{j \neq J, J+1} Z_{j-1} Z_{j+1}(1+X_j) + h_2 \sum_{j \neq J-1, J, J+1, J+2} Z_{j-2} Z_{j+2}(1+X_{j-1}X_{j+1}) \\
& + h_1 \sigma_{j+\frac{1}{2}}^x \Big( \frac{1+i\tau_j^x \sigma_{j+\frac{1}{2}}^z}{\sqrt{2}} Z_{J-1} Z_{J+1}(1+X_J) + \frac{1-i\sigma_{j+\frac{1}{2}}^z}{\sqrt{2}} Z_J Z_{J+2}(1+X_{J+1}) \Big) \\
& + h_2 \sigma_{j+\frac{1}{2}}^x \Big( \frac{1+i\tau_j^x \sigma_{j+\frac{1}{2}}^z}{\sqrt{2}} Z_{J-3} Z_{J+1}(1+X_{J-2}X_J) + \frac{1-i\sigma_{j+\frac{1}{2}}^z}{\sqrt{2}} (X_{J-1})^{\frac{1+\tau_j^x}{2}} Z_{J-2} Z_{J+2}(1+X_{J-1}X_{J+1}) \\
& + \frac{1+i\tau_j^x \sigma_{j+\frac{1}{2}}^z}{\sqrt{2}} (X_J)^{\frac{1+\tau_j^x}{2}} Z_{J-1} Z_{J+3}(1+X_J X_{J+2}) + \frac{1-i\sigma_{j+\frac{1}{2}}^z}{\sqrt{2}} Z_J Z_{J+4}(1+X_{J+1}X_{J+3}) \Big).
\end{aligned}
\tag{C.6}
$$

Repeating the same algorithm on every link, we find the Hamiltonian coupled to the $\mathcal{A}$ gauge fields:

$$
\begin{aligned}
\tilde{H} = {} & h_0 \sum_j X_j + h_1 \sum_j Z_{j-1} Z_{j+1}(1+X_j) Z_{\ell_{j-\frac{1}{2}}} \left( \frac{1+Z_{a_{j+\frac{1}{2}}}}{2} Z_{\ell_{j+\frac{1}{2}}} + \frac{1-Z_{a_{j+\frac{1}{2}}}}{2} X_{\ell_{j+\frac{1}{2}}} \right) \\
& + h_2 \sum_j Z_{j-2} Z_{j+2}(1+X_{j-1}X_{j+1})(Z_{\ell_{j-\frac{3}{2}}}) \left( \frac{1+Z_{a_{j-\frac{1}{2}}}}{2} Z_{\ell_{j-\frac{1}{2}}} + \frac{1-Z_{a_{j-\frac{1}{2}}}}{2} X_{\ell_{j-\frac{1}{2}}} X_{j-1} \right) \\
& \left( \frac{1+Z_{a_{j+\frac{1}{2}}}}{2} Z_{\ell_{j+\frac{1}{2}}} + \frac{1-Z_{a_{j+\frac{1}{2}}}}{2} Z_{\ell_{j+\frac{1}{2}}} X_{j-1} \right) \left( \frac{1+Z_{a_{j+\frac{3}{2}}}}{2} Z_{\ell_{j+\frac{3}{2}}} + \frac{1-Z_{a_{j+\frac{3}{2}}}}{2} X_{\ell_{j+\frac{3}{2}}} \right).
\end{aligned}
\tag{C.7}
$$

Finally, in terms of the variables in (75), it is

$$
\begin{aligned}
\tilde{H} = {} & h_0 \sum_j X_j + h_1 \sum_j Z_{j-1} Z_{j+1}(1+X_j) \left( \sigma_{j-\frac{1}{2}}^x \frac{1-i\sigma_{j-\frac{1}{2}}^z}{\sqrt{2}} \sigma_{j+\frac{1}{2}}^x \frac{1+i\tau_j^x \sigma_{j+\frac{1}{2}}^z}{\sqrt{2}} \right) \\
& + h_2 \sum_j Z_{j-2} Z_{j+2}(1+X_{j-1}X_{j+1}) \left( \sigma_{j-\frac{3}{2}}^x \frac{1-i\sigma_{j-\frac{3}{2}}^z}{\sqrt{2}} \sigma_{j-\frac{1}{2}}^x \frac{1+i\tau_{j-1}^x \sigma_{j-\frac{1}{2}}^z}{\sqrt{2}} \right. \\
& \left. \times \sigma_{j+\frac{1}{2}}^x \frac{1-i\sigma_{j+\frac{1}{2}}^z}{\sqrt{2}} \sigma_{j+\frac{3}{2}}^x \frac{1+i\tau_{j+1}^x \sigma_{j+\frac{3}{2}}^z}{\sqrt{2}} (\tau_{j-1}^x \tau_j^x)^{\frac{1-X_{j-1}}{2}} \right).
\end{aligned}
\tag{C.8}
$$

# D Lattice F-symbols

In this appendix, we compute the lattice F-symbols and verify (44), (45), and (46). We proceed by first computing the on-site F-symbols and then showing that they coincide with the lattice

F-symbols.

The lattice F-symbols and the on-site F-symbols, respectively, are defined in (43) and (48); we use the following relation to show that they are identical

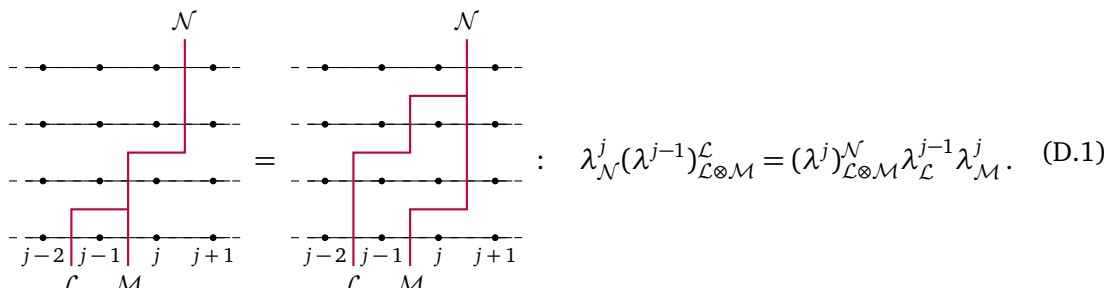

$$= \qquad : \qquad \lambda_{\mathcal{N}}^j (\lambda^{j-1})_{\mathcal{L}\otimes\mathcal{M}}^{\mathcal{L}} = (\lambda^j)_{\mathcal{L}\otimes\mathcal{M}}^{\mathcal{N}} \lambda_{\mathcal{L}}^{j-1} \lambda_{\mathcal{M}}^j. \tag{D.1}$$

Using (32), we write the lattice F-symbol relation (43) in terms of the on-site fusion operators as

$$(m^j)_{a\otimes e}^d \, \lambda_a^j \, \lambda_a^{j-1} (m^j)_{b\otimes c}^e \, \lambda_b^j = \sum_f (F_{abc}^d)_e^f \, (m^j)_{f\otimes c}^d \, \lambda_f^j (m^{j-1})_{a\otimes b}^f \lambda_a^{j-1}. \tag{D.2}$$

Multiplying this relation from the right by $(\lambda_a^{j-1})^{-1}\lambda_c^j$ we find

$$(m^j)_{a\otimes e}^d \, \lambda_a^j (m^j)_{b\otimes c}^e \, \lambda_b^j \lambda_c^j = \sum_f (F_{abc}^d)_e^f \, (m^j)_{f\otimes c}^d \, \lambda_f^j \lambda_c^j (m^{j-1})_{a\otimes b}^f,$$

$$\lambda_d^j (m^j)_{a\otimes e}^d (m^j)_{b\otimes c}^e = \lambda_d^j \sum_f (F_{abc}^d)_e^f \, (m^j)_{f\otimes c}^d (m^{j-1})_{a\otimes b}^f, \tag{D.3}$$

where in the second equality we have used (D.1). Multiplying the latter equality, from left, by $(\lambda_d^j)^{-1}$ we recover the one-site F-symobls relation (48).

To finish the argument, we need to show (D.1). As mentioned in Section 2.3, there are phase ambiguities in our definitions of the movement operators. Below we show that (D.1) is satisfied if we use the following phase convention for the movement operators:[17]

$$\lambda_{\mathcal{D}}^j = \mathrm{CNOT}_{\ell,j} \, \mathrm{H}_\ell, \qquad \lambda_g^j = (-1)^{g_e g_o} X_j^{j g_o + (j+1) g_e}. \tag{D.4}$$

The relation (D.1) is equivalent to

$$\lambda_{g+h}^j (m^{j-1})_{g\otimes h}^{g+h} = (m^j)_{g\otimes h}^{g+h} \lambda_g^j \lambda_h^j,$$

$$\lambda_{\mathcal{D}}^j (m^{j-1})_{g\otimes\mathcal{D}}^{\mathcal{D}} = (m^j)_{g\otimes\mathcal{D}}^{\mathcal{D}} \lambda_g^j \lambda_{\mathcal{D}}^j,$$

$$\lambda_{\mathcal{D}}^j (m^{j-1})_{\mathcal{D}\otimes g}^{\mathcal{D}} = (m^j)_{\mathcal{D}\otimes g}^{\mathcal{D}} \lambda_{\mathcal{D}}^j \lambda_g^j,$$

$$\lambda_g^j (m^{j-1})_{\mathcal{D}_1\otimes\mathcal{D}_2}^g = (m^j)_{\mathcal{D}_1\otimes\mathcal{D}_2}^g \lambda_{\mathcal{D}_1}^j \lambda_{\mathcal{D}_2}^j, \tag{D.5}$$

where $g = (g_e, g_o)$ parametrizes $\mathbb{Z}_2^e \times \mathbb{Z}_2^o$ defects. The first three equations are easy to verify using (42) and are compatible with (D.4). The last relation is equivalent to

$$(-1)^{g_e g_o} X_j^{j g_o + (j+1) g_e} \Big( \langle 00|_{\ell_1\ell_2} + \langle 11|_{\ell_1\ell_2} \Big) X_{\ell_1}^{j g_e + (j+1) g_o} Z_{\ell_1}^{j g_o + (j+1) g_e} =$$

$$\Big( \langle 00|_{\ell_1\ell_2} + \langle 11|_{\ell_1\ell_2} \Big) (X_{\ell_1})^{j g_o + (j+1) g_e} (Z_{\ell_1})^{j g_e + (j+1) g_o} \Big( \mathrm{CNOT}_{\ell_1,j} \, \mathrm{H}_{\ell_1} \Big) \Big( \mathrm{CNOT}_{\ell_2,j} \, \mathrm{H}_{\ell_2} \Big), \tag{D.6}$$

which is also straightforward to verify.

---

[17]Equation (D.1) is also satisfied if we had chosen $\lambda_{\mathcal{D}}^j$ to be $-\mathrm{CNOT}_{\ell,j} \, \mathrm{H}_\ell$.

### D.1 On-site F-symbols

Here, we compute the on-site F-symbols in (48) that involve the on-site fusion operators (42). For simplicity, here we take $j$ to be odd.

Let us start with

$$\vcenter{\hbox{\includegraphics{tree1}}} = \frac{1}{2}\sum_h \chi(g,h)\, \vcenter{\hbox{\includegraphics{tree2}}} . \tag{D.7}$$

We have labeled the defects as $\mathcal{D}_1, \mathcal{D}_2, \mathcal{D}_3, \mathcal{D}$ so that the qubit localized on $\mathcal{D}_j$ are denoted as $|0\rangle_{\ell_j}, |1\rangle_{\ell_j}$. More explicitly, the F-symbol equation above is equivalent to

$$m^{\mathcal{D}}_{\mathcal{D}_1 \otimes g}\, m^{g}_{\mathcal{D}_2 \otimes \mathcal{D}_3} = \frac{1}{2}\sum_h \chi(g,h)\, m^{\mathcal{D}}_{h \otimes \mathcal{D}_3}\, m^{h}_{\mathcal{D}_1 \otimes \mathcal{D}_2}, \tag{D.8}$$

where (see (42))

$$\begin{aligned}
m^{\mathcal{D}}_{\mathcal{D}_1 \otimes g} &= \left(|0\rangle_\ell \langle 0|_{\ell_1} + |1\rangle_\ell \langle 1|_{\ell_1}\right) X^{g_o}_{\ell_1} Z^{g_e}_{\ell_1}, \\
m^{\mathcal{D}}_{h \otimes \mathcal{D}_3} &= \left(|0\rangle_\ell \langle 0|_{\ell_3} + |1\rangle_\ell \langle 1|_{\ell_3}\right) Z^{h_e}_{\ell_3} X^{h_o}_{\ell_3}.
\end{aligned} \tag{D.9}$$

Here, we have parametrized $\mathbb{Z}^e_2 \times \mathbb{Z}^o_2$ elements by $g = (g_e, g_o)$ such that $\eta^e = (1,0)$, $\eta^o = (0,1)$, $\eta = (1,1)$, and $1 = (0,0)$. Using this parametrization, we have

$$m^{g}_{\mathcal{D}_1 \otimes \mathcal{D}_2} = \left(\langle 00|_{\ell_1,\ell_2} + \langle 11|_{\ell_1,\ell_2}\right) X^{g_o}_{\ell_1} Z^{g_e}_{\ell_1} = \left(\langle 00|_{\ell_1,\ell_2} + \langle 11|_{\ell_1,\ell_2}\right) Z^{g_e}_{\ell_2} X^{g_o}_{\ell_2}. \tag{D.10}$$

Equation (D.8) reduces to

$$\begin{aligned}
&\left(|0\rangle_\ell \langle 0|_{\ell_1} + |1\rangle_\ell \langle 1|_{\ell_1}\right) X^{g_o}_{\ell_1} Z^{g_e}_{\ell_1} \left(\langle 00|_{\ell_2,\ell_3} + \langle 11|_{\ell_2,\ell_3}\right) X^{g_o}_{\ell_2} Z^{g_e}_{\ell_2} \\
&= \frac{1}{2}\sum_{h_e,h_o} \chi(g,h)\left(|0\rangle_\ell \langle 0|_{\ell_3} + |1\rangle_\ell \langle 1|_{\ell_3}\right) Z^{h_e}_{\ell_3} X^{h_o}_{\ell_3} \left(\langle 00|_{\ell_1,\ell_2} + \langle 11|_{\ell_1,\ell_2}\right) Z^{h_e}_{\ell_2} X^{h_o}_{\ell_2}.
\end{aligned} \tag{D.11}$$

By replacing $\langle \bullet|_{\ell_3}$ with $|\bullet\rangle_{\ell_2}$ and identifying $|\bullet\rangle_\ell$ with $|\bullet\rangle_{\ell_1}$, we simplify the equation above into

$$X^{g_o}_{\ell_1} Z^{g_e}_{\ell_1} X^{g_o}_{\ell_2} Z^{g_e}_{\ell_2} = \frac{1}{2}\sum_{h_e,h_o} \chi(g,h) X^{h_o}_{\ell_2} Z^{h_e}_{\ell_2}\left(|00\rangle_{\ell_1,\ell_2} + |11\rangle_{\ell_1,\ell_2}\right)\left(\langle 00|_{\ell_1,\ell_2} + \langle 11|_{\ell_1,\ell_2}\right) Z^{h_e}_{\ell_2} X^{h_o}_{\ell_2}. \tag{D.12}$$

Using $\chi(g,h) = (-1)^{g_e h_o + g_o h_e}$, we further simplify the equation above into

$$\begin{aligned}
\mathbb{1}_{\ell_1,\ell_2} &= \frac{1}{2}\sum_{h_e,h_o} X^{h_o+g_o}_{\ell_2} Z^{h_e+g_e}_{\ell_2}\left(|00\rangle_{\ell_1,\ell_2} + |11\rangle_{\ell_1,\ell_2}\right)\left(\langle 00|_{\ell_1,\ell_2} + \langle 11|_{\ell_1,\ell_2}\right) Z^{h_e+g_e}_{\ell_2} X^{h_o+g_o}_{\ell_2} \\
&= |00\rangle\langle 00|_{\ell_1,\ell_2} + |11\rangle\langle 11|_{\ell_1,\ell_2} + |01\rangle\langle 01|_{\ell_1,\ell_2} + |10\rangle\langle 10|_{\ell_1,\ell_2}.
\end{aligned} \tag{D.13}$$

Thus, we have proven (D.7).

The next F-symbol relation is

$$\vcenter{\hbox{\includegraphics{tree3}}} = \chi(g,h) \vcenter{\hbox{\includegraphics{tree4}}} \quad : \quad m^{\mathcal{D}}_{g \otimes \mathcal{D}}\, m^{\mathcal{D}}_{\mathcal{D} \otimes h} = \chi(g,h)\, m^{\mathcal{D}}_{\mathcal{D} \otimes h}\, m^{\mathcal{D}}_{g \otimes \mathcal{D}}, \tag{D.14}$$

which using (42) and $\chi(g,h) = (-1)^{g_e h_o + g_o h_e}$, is equivalent to the following identity

$$Z_\ell^{g_e} X_\ell^{g_o} X_\ell^{h_o} Z_\ell^{h_e} = (-1)^{g_e h_o + g_o h_e} X_\ell^{h_o} Z_\ell^{h_e} Z_\ell^{g_e} X_\ell^{g_o}. \tag{D.15}$$

The next non-trivial F-symbols relation is

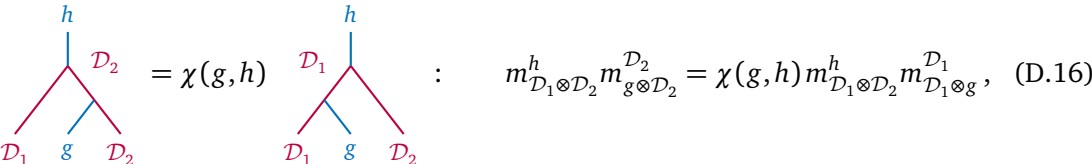

$$m_{\mathcal{D}_1 \otimes \mathcal{D}_2}^{h} m_{g \otimes \mathcal{D}_2}^{\mathcal{D}_2} = \chi(g,h) m_{\mathcal{D}_1 \otimes \mathcal{D}_2}^{h} m_{\mathcal{D}_1 \otimes g}^{\mathcal{D}_1}, \tag{D.16}$$

which follows from

$$\left(\langle 00|_{\ell_1,\ell_2} + \langle 11|_{\ell_1,\ell_2}\right) X_{\ell_1}^{h_o} Z_{\ell_1}^{h_e} Z_{\ell_2}^{g_e} X_{\ell_2}^{g_o} = (-1)^{g_e h_o + g_o h_e} \left(\langle 00|_{\ell_1,\ell_2} + \langle 11|_{\ell_1,\ell_2}\right) X_{\ell_1}^{h_o} Z_{\ell_1}^{h_e} X_{\ell_1}^{g_o} Z_{\ell_1}^{g_e}. \tag{D.17}$$

The rest of the F-symbols are

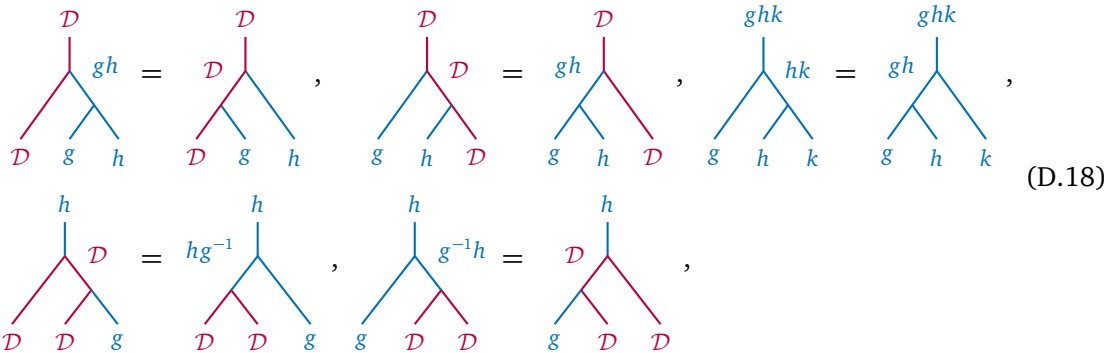

which in equations correspond to

$$\begin{aligned}
m_{\mathcal{D} \otimes gh}^{\mathcal{D}} m_{g \otimes h}^{gh} &= m_{\mathcal{D} \otimes h}^{\mathcal{D}} m_{\mathcal{D} \otimes g}^{\mathcal{D}}, \\
m_{g \otimes \mathcal{D}}^{\mathcal{D}} m_{h \otimes \mathcal{D}}^{\mathcal{D}} &= m_{gh \otimes \mathcal{D}}^{\mathcal{D}} m_{g \otimes h}^{gh}, \\
m_{g \otimes hk}^{ghk} m_{h \otimes k}^{hk} &= m_{gh \otimes k}^{ghk} m_{g \otimes h}^{gh}, \\
m_{\mathcal{D}_1 \otimes \mathcal{D}_2}^{h} m_{\mathcal{D}_2 \otimes g}^{\mathcal{D}_2} &= m_{hg^{-1} \otimes g}^{h} m_{\mathcal{D}_1 \otimes \mathcal{D}_2}^{hg^{-1}}, \\
m_{g \otimes g^{-1}h}^{h} m_{\mathcal{D}_1 \otimes \mathcal{D}_2}^{g^{-1}h} &= m_{\mathcal{D}_1 \otimes \mathcal{D}_2}^{h} m_{g \otimes \mathcal{D}_1}^{\mathcal{D}_1}.
\end{aligned} \tag{D.19}$$

These relations are, respectively, equivalent to the following identities

$$\begin{aligned}
& X_\ell^{g_o + h_o} Z_\ell^{g_e + h_e} (-1)^{g_o h_e} = X_\ell^{h_o} Z_\ell^{h_e} X_\ell^{g_o} Z_\ell^{g_e}, \\
& Z_\ell^{g_e} X_\ell^{g_o} Z_\ell^{h_e} X_\ell^{h_o} = Z_\ell^{g_e + h_e} X_\ell^{g_o + h_o} (-1)^{g_o h_e}, \\
& (-1)^{g_o(h_e + k_e) + h_o k_e} = (-1)^{(g_o + h_o)k_e + g_o h_e}, \\
& \left(\langle 00|_{\ell_1,\ell_2} + \langle 11|_{\ell_1,\ell_2}\right) X_{\ell_1}^{h_o} Z_{\ell_1}^{h_e} X_{\ell_2}^{g_o} Z_{\ell_2}^{g_e} = (-1)^{(h_o + g_o)g_e} \left(\langle 00|_{\ell_1,\ell_2} + \langle 11|_{\ell_1,\ell_2}\right) X_{\ell_1}^{h_o + g_o} Z_{\ell_1}^{h_e + g_e}, \\
& (-1)^{g_o(g_e + h_e)} \left(\langle 00|_{\ell_1,\ell_2} + \langle 11|_{\ell_1,\ell_2}\right) X_{\ell_1}^{g_o + h_o} Z_{\ell_1}^{g_e + h_e} = \left(\langle 00|_{\ell_1,\ell_2} + \langle 11|_{\ell_1,\ell_2}\right) X_{\ell_1}^{h_o} Z_{\ell_1}^{h_e} Z_{\ell_1}^{g_e} X_{\ell_1}^{g_o}.
\end{aligned} \tag{D.20}$$

Thus, we have verified all the F-symbol relations in (44), (45), and (46).

# E  More on the algebra objects

In this appendix, we discuss some of the calculations inlvoling the algebra objects.

## E.1 On-site fusion operator $m^{\mathcal{A}}_{\mathcal{A}\otimes\mathcal{A}}$

We first give a simplified form of the on-site fusion operator (61). Using (42), this expression is simplified to

$$
\begin{aligned}
2m^{\mathcal{A}_2}_{\mathcal{A}_1\otimes\mathcal{A}_2} = &\Big(\langle 0|_{\ell_1}+X_{\ell_2}Z_{\ell_2}\langle 1|_{\ell_1}\Big)|0\rangle_{a_2}\langle 00|_{a_1 a_2}+\Big(\langle 0|_{\ell_1}-X_{\ell_2}Z_{\ell_2}\langle 1|_{\ell_1}\Big)|1\rangle_{a_2}\langle 01|_{a_1 a_2}\\
&+\Big(\langle 0|_{\ell_1}+X_{\ell_2}Z_{\ell_2}\langle 1|_{\ell_1}\Big)|1\rangle_{a_2}\langle 10|_{a_1 a_2}+\Big(\langle 0|_{\ell_1}-X_{\ell_2}Z_{\ell_2}\langle 1|_{\ell_1}\Big)|0\rangle_{a_2}\langle 11|_{a_1 a_2}\,,
\end{aligned}
\tag{E.1}
$$

and leads to the final simplified form

$$
m^{\mathcal{A}_2}_{\mathcal{A}_1\otimes\mathcal{A}_2}=\frac{1}{2}\Big(\langle 0|_{a_1}+X_{a_2}\langle 1|_{a_1}\Big)\Big(\langle 0|_{\ell_1}+Z_{a_2}X_{\ell_2}Z_{\ell_2}\langle 1|_{\ell_1}\Big).
\tag{E.2}
$$

Multiplying $m^{\mathcal{A}_2}_{\mathcal{A}_1\otimes\mathcal{A}_2}$ with $\Big(|0\rangle_{a_1}\langle 0|_{a_2}+|1\rangle_{a_1}\langle 1|_{a_2}\Big)\Big(|0\rangle_{\ell_1}\langle 0|_{\ell_2}+|1\rangle_{\ell_1}\langle 1|_{\ell_2}\Big)$ from the left, we find

$$
m^{\mathcal{A}_1}_{\mathcal{A}_1\otimes\mathcal{A}_2}=\frac{1}{2}\Big(\langle 0|_{a_2}+X_{a_1}Z_{\ell_1}\langle 1|_{a_2}\Big)\Big(\langle 0|_{\ell_2}+Z_{a_2}X_{\ell_1}Z_{\ell_1}\langle 1|_{\ell_2}\Big).
\tag{E.3}
$$

## E.2 The on-site Frobenius algebra axioms

Now we verify that $(\mathcal{A}, m, u)$ satisfies the axioms of Frobenius algebras. We begin with the associativity condition (63), which is

$$
m^{\mathcal{A}_3}_{\mathcal{A}_1\otimes\mathcal{A}_3}m^{\mathcal{A}_3}_{\mathcal{A}_2\otimes\mathcal{A}_3}=m^{\mathcal{A}_3}_{\mathcal{A}_2\otimes\mathcal{A}_3}m^{\mathcal{A}_2}_{\mathcal{A}_1\otimes\mathcal{A}_2}\,.
\tag{E.4}
$$

Using (62), it is equivalent to showing that

$$
\text{LHS}=\Big(\langle 0|_{a_1}+X_{a_3}\langle 1|_{a_1}\Big)\Big(\langle 0|_{\ell_1}+Z_{a_3}X_{\ell_3}Z_{\ell_3}\langle 1|_{\ell_1}\Big)\Big(\langle 0|_{a_2}+X_{a_3}\langle 1|_{a_2}\Big)\Big(\langle 0|_{\ell_2}+Z_{a_3}X_{\ell_3}Z_{\ell_3}\langle 1|_{\ell_2}\Big),
\tag{E.5}
$$

is equal to

$$
\text{RHS}=\Big(\langle 0|_{a_2}+X_{a_3}\langle 1|_{a_2}\Big)\Big(\langle 0|_{\ell_2}+Z_{a_3}X_{\ell_3}Z_{\ell_3}\langle 1|_{\ell_2}\Big)\Big(\langle 0|_{a_1}+X_{a_2}\langle 1|_{a_1}\Big)\Big(\langle 0|_{\ell_1}+Z_{a_2}X_{\ell_2}Z_{\ell_2}\langle 1|_{\ell_1}\Big).
\tag{E.6}
$$

We begin by commuting the two middle terms in (E.5) and rewriting it as

$$
\text{LHS}=\Big(\langle 0|_{a_1}+X_{a_3}\langle 1|_{a_1}\Big)\Big(\langle 0|_{a_2}+X_{a_3}\langle 1|_{a_2}\Big)\Big(\langle 0|_{\ell_1}+Z_{a_2}Z_{a_3}X_{\ell_3}Z_{\ell_3}\langle 1|_{\ell_1}\Big)\Big(\langle 0|_{\ell_2}+Z_{a_3}X_{\ell_3}Z_{\ell_3}\langle 1|_{\ell_2}\Big).
\tag{E.7}
$$

Next, by exchanging the two (commuting) middle terms in (E.6) and noting that $(\langle 0|_{a_2}+X_{a_3}\langle 1|_{a_2})X_{a_2}=(\langle 0|_{a_2}+X_{a_3}\langle 1|_{a_2})X_{a_3}$, we rewrite it as

$$
\text{RHS}=\Big(\langle 0|_{a_1}+X_{a_3}\langle 1|_{a_1}\Big)\Big(\langle 0|_{a_2}+X_{a_3}\langle 1|_{a_2}\Big)\Big(\langle 0|_{\ell_2}+Z_{a_3}X_{\ell_3}Z_{\ell_3}\langle 1|_{\ell_2}\Big)\Big(\langle 0|_{\ell_1}+Z_{a_2}X_{\ell_2}Z_{\ell_2}\langle 1|_{\ell_1}\Big).
\tag{E.8}
$$

The last two terms can be rewritten as

$$
\begin{aligned}
&\Big(\langle 0|_{\ell_2}+Z_{a_3}X_{\ell_3}Z_{\ell_3}\langle 1|_{\ell_2}\Big)\Big(\langle 0|_{\ell_1}+Z_{a_2}X_{\ell_2}Z_{\ell_2}\langle 1|_{\ell_1}\Big)\\
&=\Big(\langle 00|_{\ell_1\ell_2}+Z_{a_3}X_{\ell_3}Z_{\ell_3}\langle 01|_{\ell_1\ell_2}-Z_{a_2}\langle 11|_{\ell_1\ell_2}+Z_{a_2}Z_{a_3}X_{\ell_3}Z_{\ell_3}\langle 10|_{\ell_1\ell_2}\Big)\\
&=\Big(\langle 0|_{\ell_1}+Z_{a_2}Z_{a_3}X_{\ell_3}Z_{\ell_3}\langle 1|_{\ell_1}\Big)\Big(\langle 0|_{\ell_2}+Z_{a_3}X_{\ell_3}Z_{\ell_3}\langle 1|_{\ell_2}\Big).
\end{aligned}
\tag{E.9}
$$

Using this relation, we verify that, indeed (E.7) is equal to (E.8).

Now that we have verified the associativity condition (E.4), we use it to show the Frobenius conditions (65). One of the Frobenius conditions can be rewritten as

$$\left(m^{\mathcal{A}_3}_{\mathcal{A}_1\otimes\mathcal{A}_3}\right)^\dagger m^{\mathcal{A}_3}_{\mathcal{A}_2\otimes\mathcal{A}_3} = m^{\mathcal{A}_3}_{\mathcal{A}_2\otimes\mathcal{A}_3}\left(m^{\mathcal{A}_2}_{\mathcal{A}_1\otimes\mathcal{A}_2}\right)^\dagger. \tag{E.10}$$

To show this equation, we note that the on-site fusion operator (62) satisfies the relation

$$\left(m^{\mathcal{A}_2}_{\mathcal{A}_1\otimes\mathcal{A}_2}\right)^\dagger = Z_{\ell_1}\left(m^{\mathcal{A}_2}_{\mathcal{A}_1\otimes\mathcal{A}_2}\right)^{\mathrm{T}_1}, \tag{E.11}$$

where $\mathrm{T}_1$ is the transpose operation on qubits $a_1$ and $\ell_1$ that acts as

$$\left(\alpha|0\rangle_{a_1} + \beta|1\rangle_{a_1}\right)^{\mathrm{T}_1} = \alpha\langle 0|_{a_1} + \beta\langle 1|_{a_1}, \quad \text{and} \quad \left(\alpha|0\rangle_{\ell_1} + \beta|1\rangle_{\ell_1}\right)^{\mathrm{T}_1} = \alpha\langle 0|_{\ell_1} + \beta\langle 1|_{\ell_1}. \tag{E.12}$$

One can check that the Frobenius condition (E.10) follows from acting with $\mathrm{T}_1$ on both sides of (E.4) and multiplying it from the left with $Z_{\ell_1}$. The other Frobenius condition

$$\left(m^{\mathcal{A}_3}_{\mathcal{A}_2\otimes\mathcal{A}_3}\right)^\dagger m^{\mathcal{A}_3}_{\mathcal{A}_1\otimes\mathcal{A}_3} = m^{\mathcal{A}_2}_{\mathcal{A}_1\otimes\mathcal{A}_2}\left(m^{\mathcal{A}_3}_{\mathcal{A}_2\otimes\mathcal{A}_3}\right)^\dagger, \tag{E.13}$$

follows from the Hermitian conjugate of (E.10) and $\left(m^{\mathcal{A}_2}_{\mathcal{A}_2\otimes\mathcal{A}_3}\right)^\dagger = Z_{\ell_3}\left(m^{\mathcal{A}_2}_{\mathcal{A}_2\otimes\mathcal{A}_3}\right)^{\mathrm{T}_1}$.

## E.3 The lattice Frobenius algebra axioms

Having shown that the *on-site* fusion operator $m^{\mathcal{A}_2}_{\mathcal{A}_1\otimes\mathcal{A}_2}$ satisfies the Frobenius algebra axioms, here we show the axioms for the fusion operator $(\lambda^j)^{\mathcal{A}_2}_{\mathcal{A}_1\otimes\mathcal{A}_2}$ defined in (60). As we will see, it suffices to show the special relation:

$$\lambda^j_{\mathcal{A}_2}(\lambda^{j-1})^{\mathcal{A}_2}_{\mathcal{A}_1\otimes\mathcal{A}_2} = (\lambda^j)^{\mathcal{A}_2}_{\mathcal{A}_1\otimes\mathcal{A}_2}\lambda^{j-1}_{\mathcal{A}_1}\lambda^j_{\mathcal{A}_2}, \tag{E.14}$$

which is equivalent to

$$\lambda^j_{\mathcal{A}_2}m^{\mathcal{A}_2}_{\mathcal{A}_1\otimes\mathcal{A}_2} = m^{\mathcal{A}_2}_{\mathcal{A}_1\otimes\mathcal{A}_2}\lambda^j_{\mathcal{A}_1}\lambda^j_{\mathcal{A}_2}. \tag{E.15}$$

This readily follows from (D.5) by using $\lambda^j_{\mathcal{A}} = \frac{1+Z_a}{2}\left(\frac{1+Z_\ell}{2} + \frac{1-Z_\ell}{2}\lambda^j_\eta\right) + \frac{1-Z_a}{2}\lambda^j_{\mathcal{D}}$. We can take the Hermitian conjugate of the above to obtain

$$(m^{\mathcal{A}_2}_{\mathcal{A}_2\otimes\mathcal{A}_1})^\dagger\lambda^j_{\mathcal{A}_2} = \lambda^j_{\mathcal{A}_1}\lambda^j_{\mathcal{A}_2}(m^{\mathcal{A}_2}_{\mathcal{A}_2\otimes\mathcal{A}_1})^\dagger. \tag{E.16}$$

Next, we demonstrate that the algebra axioms for $(\lambda^j)^{\mathcal{A}_2}_{\mathcal{A}_1\otimes\mathcal{A}_2}$ follows from those of $m^{\mathcal{A}_2}_{\mathcal{A}_1\otimes\mathcal{A}_2}$.

**Associativity relation:** The associativity axiom for the fusion operator $(\lambda^j)_{\mathcal{A}_1 \otimes \mathcal{A}_2}^{\mathcal{A}_2}$ is

$$\dots = \dots : (\lambda^j)_{\mathcal{A}_2 \otimes \mathcal{A}_3}^{\mathcal{A}_3} (\lambda^{j-1})_{\mathcal{A}_1 \otimes \mathcal{A}_2}^{\mathcal{A}_2} = (\lambda^j)_{\mathcal{A}_1 \otimes \mathcal{A}_3}^{\mathcal{A}_3} \lambda_{\mathcal{A}_1}^{j-1} (\lambda^j)_{\mathcal{A}_2 \otimes \mathcal{A}_3}^{\mathcal{A}_3}, \quad \text{(E.17)}$$

which using (60) is equivalent to

$$m_{\mathcal{A}_2 \otimes \mathcal{A}_3}^{\mathcal{A}_3} \lambda_{\mathcal{A}_2}^{j} m_{\mathcal{A}_1 \otimes \mathcal{A}_2}^{\mathcal{A}_2} \lambda_{\mathcal{A}_1}^{j-1} = m_{\mathcal{A}_1 \otimes \mathcal{A}_3}^{\mathcal{A}_3} \lambda_{\mathcal{A}_1}^{j} \lambda_{\mathcal{A}_1}^{j-1} m_{\mathcal{A}_2 \otimes \mathcal{A}_3}^{\mathcal{A}_3} \lambda_{\mathcal{A}_2}^{j}. \quad \text{(E.18)}$$

Using (E.15) and the fact that $\lambda_{\mathcal{A}_1}^{j} \lambda_{\mathcal{A}_1}^{j-1}$ commutes with $m_{\mathcal{A}_2 \otimes \mathcal{A}_3}^{\mathcal{A}_3}$, the above relation is simplified into $m_{\mathcal{A}_2 \otimes \mathcal{A}_3}^{\mathcal{A}_3} m_{\mathcal{A}_1 \otimes \mathcal{A}_2}^{\mathcal{A}_2} = m_{\mathcal{A}_1 \otimes \mathcal{A}_3}^{\mathcal{A}_3} m_{\mathcal{A}_2 \otimes \mathcal{A}_3}^{\mathcal{A}_3}$ which is the associativity relation (E.4) of the on-site fusion operator.

**Frobenius conditions:** The first Frobenius condition is:

$$\dots = \dots : \left((\lambda^j)_{\mathcal{A}_1 \otimes \mathcal{A}_3}^{\mathcal{A}_3}\right)^\dagger (\lambda^j)_{\mathcal{A}_2 \otimes \mathcal{A}_3}^{\mathcal{A}_3} = \lambda_{\mathcal{A}_1}^{j-1} (\lambda^j)_{\mathcal{A}_2 \otimes \mathcal{A}_3}^{\mathcal{A}_3} \left((\lambda^{j-1})_{\mathcal{A}_1 \otimes \mathcal{A}_2}^{\mathcal{A}_2}\right)^\dagger, \quad \text{(E.19)}$$

which using (60) is equivalent to

$$(\lambda_{\mathcal{A}_1}^{j})^\dagger (m_{\mathcal{A}_1 \otimes \mathcal{A}_3}^{\mathcal{A}_3})^\dagger m_{\mathcal{A}_2 \otimes \mathcal{A}_3}^{\mathcal{A}_3} \lambda_{\mathcal{A}_2}^{j} = \lambda_{\mathcal{A}_1}^{j-1} m_{\mathcal{A}_2 \otimes \mathcal{A}_3}^{\mathcal{A}_3} \lambda_{\mathcal{A}_2}^{j} (\lambda_{\mathcal{A}_1}^{j-1})^\dagger (m_{\mathcal{A}_1 \otimes \mathcal{A}_2}^{\mathcal{A}_2})^\dagger. \quad \text{(E.20)}$$

Using (E.16) and the fact that $\lambda_{\mathcal{A}_1}^{j}$ commutes with $m_{\mathcal{A}_2 \otimes \mathcal{A}_3}^{\mathcal{A}_3}$, it can be simplified to the on-site Frobenius condition in (E.10): $\left(m_{\mathcal{A}_1 \otimes \mathcal{A}_3}^{\mathcal{A}_3}\right)^\dagger m_{\mathcal{A}_2 \otimes \mathcal{A}_3}^{\mathcal{A}_3} = m_{\mathcal{A}_2 \otimes \mathcal{A}_3}^{\mathcal{A}_3} \left(m_{\mathcal{A}_1 \otimes \mathcal{A}_2}^{\mathcal{A}_2}\right)^\dagger$.

The second Frobenius condition

$$\dots = \dots : \quad \text{(E.21)}$$

$$\left((\lambda^j)_{\mathcal{A}_2 \otimes \mathcal{A}_3}^{\mathcal{A}_3}\right)^\dagger (\lambda^j)_{\mathcal{A}_1 \otimes \mathcal{A}_3}^{\mathcal{A}_3} = (\lambda^{j-1})_{\mathcal{A}_1 \otimes \mathcal{A}_2}^{\mathcal{A}_2} \left((\lambda^j)_{\mathcal{A}_2 \otimes \mathcal{A}_3}^{\mathcal{A}_3}\right)^\dagger (\lambda_{\mathcal{A}_1}^{j-1})^\dagger,$$

is the Hermitian conjugate of (E.19).

**Seperability condition:**

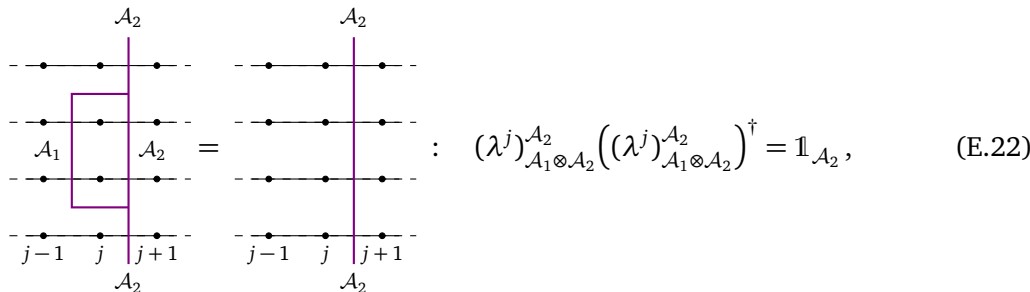

$$(\lambda^j)^{\mathcal{A}_2}_{\mathcal{A}_1 \otimes \mathcal{A}_2} \left( (\lambda^j)^{\mathcal{A}_2}_{\mathcal{A}_1 \otimes \mathcal{A}_2} \right)^\dagger = \mathbb{1}_{\mathcal{A}_2}, \tag{E.22}$$

which is equivalent to the on-site separability condition (66) using $\lambda^j_{\mathcal{A}_1} (\lambda^j_{\mathcal{A}_1})^\dagger = \mathbb{1}$.

**The unit and co-unit:** The unit axioms on the lattice are:

$$(\lambda^j)^{\mathcal{A}_2}_{\mathcal{A}_1 \otimes \mathcal{A}_2} u_{\mathcal{A}_1} = \mathbb{1}_{\mathcal{A}_2}, \tag{E.23}$$

and

$$(\lambda^j)^{\mathcal{A}_1}_{\mathcal{A}_1 \otimes \mathcal{A}_2} u_{\mathcal{A}_2} = \lambda^j_{\mathcal{A}_1}. \tag{E.24}$$

We take the Hermitian conjugate to obtain the co-unit axioms. Using the fact that $\lambda^j_{\mathcal{A}_1} u_{\mathcal{A}_1} = u_{\mathcal{A}_1}$ (which itself follows from (E.15) and the uniqueness of the on-site unit), the unit and co-unit axioms are reduced to their on-site versions (68): $m^{\mathcal{A}_2}_{\mathcal{A}_1 \otimes \mathcal{A}_2} u_{\mathcal{A}_1} = m^{\mathcal{A}_2}_{\mathcal{A}_2 \otimes \mathcal{A}_1} u_{\mathcal{A}_1} = \mathbb{1}_{\mathcal{A}_2}$.

**Symmetric condition:** Finally, the symmetric condition (98) follows from its on-site version (69) and using (E.15).

## E.4   A matrix algebra

Here, we identify an ordinary matrix algebra $\text{Mat}(2, \mathbb{C})$ induced by the on-site fusion operator $m^{\mathcal{A}_2}_{\mathcal{A}_1 \otimes \mathcal{A}_2}$ on the four-dimensional vector space $\mathbb{C}^2_a \otimes \mathbb{C}^2_\ell$. This also provides alternative verifications of the special associativity condition in (E.15) and the on-site associativity (E.4).

To identify this algebra, we consider the following basis for $\mathbb{C}^2_a \otimes \mathbb{C}^2_\ell$

$$1 \equiv 2|0\rangle_a |0\rangle_\ell, \qquad \chi \equiv 2|1\rangle_a |0\rangle_\ell, \qquad \psi \equiv 2|1\rangle_a |1\rangle_\ell, \qquad \chi\psi \equiv 2|0\rangle_a |1\rangle_\ell, \tag{E.25}$$

and use the notation

$$m^{\mathcal{A}_2}_{\mathcal{A}_1 \otimes \mathcal{A}_2} |\mathcal{X}\rangle_1 \otimes |\mathcal{Y}\rangle_2 = |\mathcal{Z}\rangle_2 \quad \Rightarrow \quad \mathcal{X} \cdot \mathcal{Y} = \mathcal{Z}, \quad \text{for } \mathcal{X}, \mathcal{Y}, \mathcal{Z} \in \langle 1, \chi, \psi, \chi\psi \rangle. \tag{E.26}$$

Using the above notation and recalling that $m_{\mathcal{A}_1 \otimes \mathcal{A}_2}^{\mathcal{A}_2} = \frac{1}{2} \big( \langle 0|_{a_1} + X_{a_2} \langle 1|_{a_1} \big) \big( \langle 0|_{\ell_1} + Z_{a_2} X_{\ell_2} Z_{\ell_2} \langle 1|_{\ell_1} \big)$, we find

$$
\begin{aligned}
m_{\mathcal{A}_1 \otimes \mathcal{A}_2}^{\mathcal{A}_2} |1\rangle_1 &= \mathbb{1}_{\mathcal{A}_2} & : & \quad 1 \cdot 1 = 1, & 1 \cdot \chi = \chi, & \quad 1 \cdot \psi = \psi, & \quad 1 \cdot \chi\psi = \chi\psi, \\
m_{\mathcal{A}_1 \otimes \mathcal{A}_2}^{\mathcal{A}_2} |\chi\rangle_1 &= X_{a_2} & : & \quad \chi \cdot 1 = \chi, & \chi \cdot \chi = 1, & \quad \chi \cdot \psi = \chi\psi, & \quad \chi \cdot \chi\psi = \psi, \\
m_{\mathcal{A}_1 \otimes \mathcal{A}_2}^{\mathcal{A}_2} |\psi\rangle_1 &= -Y_{a_2} Y_{\ell_2} & : & \quad \psi \cdot 1 = \psi, & \psi \cdot \chi = -\chi\psi, & \quad \psi \cdot \psi = 1, & \quad \psi \cdot \chi\psi = -\chi, \\
m_{\mathcal{A}_1 \otimes \mathcal{A}_2}^{\mathcal{A}_2} |\chi\psi\rangle_1 &= Z_{a_2} X_{\ell_2} Z_{\ell_2} : & & \quad \chi\psi \cdot 1 = \chi\psi, & \chi\psi \cdot \chi = -\psi, & \quad \chi\psi \cdot \psi = \chi, & \quad \chi\psi \cdot \chi\psi = -1.
\end{aligned}
\tag{E.27}
$$

We see that $1, \chi, \psi, \chi\psi$ in fact generate the rank-2 Clifford algebra, which is the same as $\mathrm{Mat}(2, \mathbb{C})$. Since the latter is associative, this verifies the on-site associativity relation (E.4).

Finally, to verify (E.15), we compute the action of the movement operator $\lambda_{\mathcal{A}}^j$ on (E.25). Using $\lambda_{\mathcal{A}}^j = \big(-X_j\big)^{\frac{1-Z_\ell}{2}} (Z_\ell \, H_\ell)^{\frac{1-Z_a}{2}}$ in (59), we find

$$
\begin{aligned}
\lambda^j(1) &\equiv \lambda_{\mathcal{A}}^j \big( 2|0\rangle_a |0\rangle_\ell \big) = 1, \\
\lambda^j(\chi) &\equiv \lambda_{\mathcal{A}}^j \big( 2|1\rangle_a |0\rangle_\ell \big) = (-X_j)^{\frac{1-Z_\ell}{2}} \big( 2|1\rangle_a |-\rangle_\ell \big) = \frac{1}{\sqrt{2}} \big( \chi + X_j \psi \big), \\
\lambda^j(\psi) &\equiv \lambda_{\mathcal{A}}^j \big( 2|1\rangle_a |1\rangle_\ell \big) = (-X_j)^{\frac{1-Z_\ell}{2}} \big( 2|1\rangle_a |+\rangle_\ell \big) = \frac{1}{\sqrt{2}} \big( \chi - X_j \psi \big), \\
\lambda^j(\chi\psi) &\equiv \lambda_{\mathcal{A}}^j \big( 2|0\rangle_a |1\rangle_\ell \big) = (-X_j) \chi\psi.
\end{aligned}
\tag{E.28}
$$

Using the notation above, (E.15) is equivalent to

$$
\lambda(\mathcal{X} \cdot \mathcal{Y}) = \lambda(\mathcal{X}) \cdot \lambda(\mathcal{Y}), \quad \text{for } \mathcal{X}, \mathcal{Y} \in \langle 1, \chi, \psi, \chi\psi \rangle.
\tag{E.29}
$$

For $\mathcal{Y} = 1$ or $\mathcal{X} = 1$,

$$
\lambda(\mathcal{X} \cdot 1) = \lambda(\mathcal{X}) \cdot \lambda(1), \quad \text{and} \quad \lambda(1 \cdot \mathcal{Y}) = \lambda(1) \cdot \lambda(\mathcal{Y}),
\tag{E.30}
$$

are satisfied as $\lambda(1) = 1$. It remains to show (E.29) for the generators of the Clifford algebra:

$$
\begin{aligned}
\lambda(\chi)^2 &= \frac{1}{2} \big( \chi + X_j \psi \big)^2 = 1 = \lambda(\chi^2), \\
\lambda(\psi)^2 &= \frac{1}{2} \big( \chi - X_j \psi \big)^2 = 1 = \lambda(\psi^2), \\
\lambda(\chi) \cdot \lambda(\psi) &= \frac{1}{2} \big( \chi + X_j \psi \big) \big( \chi - X_j \psi \big) = -X_j \, \chi\psi = \lambda(\chi \cdot \psi),
\end{aligned}
\tag{E.31}
$$

thus the special relation (E.15) is satisfied.

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
