# Peer review of "Gauging non-invertible symmetries on the lattice"

_SciPost Physics, doi:SciPost Phys. 19, 063 (2025)_

## Round 1 · Referee Report · Anonymous (Referee 2) · 2025-6-23

Strengths

  1. Concrete realization of abstract algebraic structures
  2. General gauging framework
  3. Explicit, fully worked-out example

Report

This paper presents a general method for gauging finite non-invertible symmetries in 1+1D lattice systems. The authors provide a concrete and detailed construction for implementing and gauging the Rep(D₈) fusion category symmetry in spin chains, starting from the well-known Kennedy–Tasaki duality. They develop a systematic framework, including the introduction of gauge qubits and the explicit realization of the fusion algebra and F-symbols on the lattice. The paper presents, to my knowledge, the first systematic and explicit procedure for gauging finite non-invertible symmetries in microscopic lattice Hamiltonians. Thus I would thus recommend this paper for publication.

I only have a small question: the paper provides a clear and detailed derivation of gauging Rep(D₈) symmetries on the lattice in Section 3, but is there any physical intuition behind some of the key results, such as the form of the Gauss law or the choice of minimal coupling to the gauge field?

Recommendation

Publish (surpasses expectations and criteria for this Journal; among top 10%)

  • validity: top
  • significance: top
  • originality: top
  • clarity: top
  • formatting: perfect
  • grammar: perfect

Author:  Sahand Seifnashri  on 2025-07-19  [id 5652]

(in reply to Report 2 on 2025-06-23)

We thank the referee for their positive feedback. Our gauging procedure generalizes the gauging of invertible symmetries, for which the physical intuition is more transparent, as reviewed in Appendix A.

---

## Round 1 · Referee Report · Anonymous (Referee 1) · 2025-6-23

Strengths

  1. Detailed description of the gauging procedure of $\mathrm{Rep}(D_8)$ symmetry realized in 1+1d spin chains of qubits

  2. A new prescription for gauging general finite non-invertible symmetries in 1+1d lattice systems

  3. Detailed computations and helpful background information are provided in the appendices, which make the paper self-contained.

  4. The manuscript is written clearly.

Report

This paper discusses the gauging of finite non-invertible symmetries in 1+1d lattice systems defined on a tensor product Hilbert space. The authors proposed a prescription for gauging finite non-invertible symmetries on the lattice, based on the framework developed by one of the authors in a previous paper. The gauging procedure proposed in this paper can be thought of as a lattice analogue of the gauging in continuum quantum field theories. The authors demonstrated their gauging procedure in great detail in the case of a non-maximal gauging of $\mathrm{Rep}(D_8)$ symmetry realized in simple spin chains of qubits. They also revealed that the corresponding gauging map is implemented by one of the symmetry operators of the non-invertible cosine symmetry. Although a particular focus is on the gauging of the $\mathrm{Rep}(D_8)$ symmetry, the gauging prescription described in the paper should also be applicable to more general finite non-invertible symmetries, as articulated in Section 4.

This paper provides new insights into the gauging of finite non-invertible symmetries. In particular, it clarifies the notion of gauge fields for non-invertible symmetries on the lattice and their relation to topological defects. The definition of the gauging in this paper is complementary to the existing definition of the gauging in the anyon chain model, whose state space typically does not admit a tensor product decomposition. The conceptual and technical developments in this paper would be a valuable addition to the existing literature. I would thus recommend this paper for publication.

Requested changes

It might be good to address the following points.

  1. p.27, around (3.33): could you clarify why the local terms in the Hamiltonian $\tilde{H}$ commute with the projection $P_j$ by construction?

  2. p.30, Section 4: the first paragraph says that the gauging procedure in this paper can be applied to any finite non-invertible symmetry. Is it implicitly assumed here that the finite non-invertible symmetry under consideration is realized on a tensor product Hilbert space without mixing with the lattice translation?

  3. p.30, the last sentence: could you elaborate on why "the gauging map has an ambiguity by composing it with a local unitary transformation from the left"?

Please see also the following typos.

  1. p.8, Figure 1: "For $\lambda = - h_0/h_1 \in [-1, 1)$" would be a typo of "For $\lambda = - h_0/h_1 \in (-1, 1]$"

  2. p.12, above (2.23): "$H_{\eta^o}^{(2, 3)} = H_{\eta^o}^{(1, 2)}$, and $H_{\eta^e}^{(0, 1)} = H_{\eta^e}^{(1, 2)}$" would be a typo of "$H_{\eta^e}^{(2, 3)} = H_{\eta^o}^{(1, 2)}$, and $H_{\eta^o}^{(0, 1)} = H_{\eta^e}^{(1, 2)}$"

  3. p.13, footnote 7: "$\mathcal{H}_{\mathcal{L}; \mathcal{M}}^{(j-1, j); (j, j+2)}$" on the first and second lines would be a typo of "$\mathcal{H}_{\mathcal{L}; \mathcal{M}}^{(j-1, j); (j, j+1)}$"

  4. p.14, below (2.27): "a $\mathcal{D}$ defects" would be a typo of "a $\mathcal{D}$ defect"

  5. p.18, the first line of Section 3: ref [82] is the same as ref [56].

  6. p.28, below (3.37): "gauged invariant" would be a typo of "gauge invariant"

  7. p.33, above (4.12): "gauging fixing" would be a typo of "gauge fixing"

8.p.37, above (A.12): "This fusion operator be" would be a typo of "This fusion operator can be"

  1. p.47, footnote 16: "if $O_j$ is commutes with" would be a typo of "if $O_j$ commutes with"

  2. p.51, the first line of Section D.1: "Here, compute" would be a typo of "Here, we compute"

  3. p.55, (E.13): "$(m_{\mathcal{A}_2 \otimes \mathcal{A}_3}^{\mathcal{A}_2})^{\dagger}$" would be a typo of "$(m_{\mathcal{A}_2 \otimes \mathcal{A}_3}^{\mathcal{A}_3})^{\dagger}$"

Recommendation

Publish (surpasses expectations and criteria for this Journal; among top 10%)

  • validity: -
  • significance: -
  • originality: -
  • clarity: -
  • formatting: -
  • grammar: -

Author:  Sahand Seifnashri  on 2025-07-19  [id 5651]

(in reply to Report 1 on 2025-06-23)

We thank the referee for their thoughtful comments and valuable feedback.

1. This point follows from our procedure for coupling to the gauge field. While it may not be immediately evident, the result emerges from a diagrammatic representation of defects and $P_j$, as discussed in Section 4 and Appendix A.

2. A finite non-invertible symmetry cannot mix with lattice translation, as such mixing would generate an infinite number of simple objects. Therefore, no additional assumptions are required.

3. This refers to the invariance of the left-hand side of equation (4.3) under the transformation $\mathsf{G} \mapsto U \mathsf{G}$.

We also thank the referee for identifying the typographical errors. All have been corrected except for item 2, which is not a typo and is correct as written.

---

## Round 1 · Referee Report · Anonymous (Referee 3) · 2025-6-29

Strengths

  1. Giving a general framework for gauging noninvertible symmetry on lattice.

  2. Illustrating this framework with the Rep$(D_8)$ example.

Weaknesses

  1. The calculation for the Rep$(D_8)$ example is too heavy and reader might be lost.

Report

The authors present a general framework to gauge noninvertible symmetry in (1+1)d spin chain and illustrate this method using the example of non-maximal gauging of Rep$(D_8)$ fusion category. It definitely opens a new research direction and deserves publication. However, I have some questions and minor suggestions in the requested changes.

Requested changes

Suggestions: 1. The calculations for Rep$(D_8)$ is too heavy and the reader might be lost in the details. It is better to integrate Appendix A to the main text to help readers to grasp the essence of gauging (non)invertible symmetries on the lattice in the language of algebraic objects. I think even reviewing gauging invertible symmetry in the language of algebraic objects is useful.

  1. Could author comment on the connections between lattice and continuum? How do the two gauge fields become in the continuum? Can we turn on these background gauge fields in the partition functions?

Questions: 1. Is gauging a noninvertible symmetry a duality transformation?

  1. For ordinary symmetries, we can impose the Gauss law energetically. Can we do the same trick here for gauging noninvertible symmetries?

  2. The authors claim gauging in Rep$(D_8)$ is the simplest example. But since you have realized lattice models with cosine symmetry, I guess non-maximal gauging in Rep$(S_3)$ should be simpler?

  3. It might be different realization of noninvertible symmetry on the lattice. For example, the noninvertible symmetry of Rep($D_8$) can be realized as gauging $\mathbb Z_2\times \mathbb Z_2$ (S) or KT (TST). Does it matter in the non-maximal gauging?

Recommendation

Ask for minor revision

  • validity: top
  • significance: top
  • originality: top
  • clarity: high
  • formatting: excellent
  • grammar: perfect

Author:  Sahand Seifnashri  on 2025-07-19  [id 5653]

(in reply to Report 3 on 2025-06-29)

We thank the referee for their thoughtful comments and feedback.

1 - This is an excellent question. Unfortunately, we do not currently have an answer, and we believe it remains an open question worth exploring in future work.

  1. Absolutely — one can impose Gauss’s law energetically, and in this respect, there is no difference from the standard case of gauging an invertible symmetry.

  2. Indeed, Rep(S₃) is simpler. The advantage of using Rep(D₈) is that the cosine symmetry provides an alternative route for computing the gauging map, which we have used to independently verify our result. That said, applying our method to the non-maximal gauging of Rep(S₃), which is comparatively simple, would be worthwhile.

  3. Indeed, it does not matter, as the choices are related by a local unitary transformation, i.e., a local change of basis.

---

## Editorial Decision

published